Submission

# An introduction to Markovian open quantum systems

**Shovan Dutta**[1⋆]

**1** Raman Research Institute, Bengaluru 560080, India

⋆ shovan.dutta@rri.res.in

## Abstract

This is a concise, pedagogical introduction to the dynamic field of open quantum systems governed by Markovian master equations. We focus on the mathematical and physical origins of the widely used Lindblad equation, its unraveling in terms of stochastic pure-state trajectories and the corresponding continuous measurement protocols, the structure of steady states with emphasis on the role of symmetry and conservation laws, and a sampling of the novel physical phenomena that arise from nonunitary dynamics (dissipation and measurements). This is far from a comprehensive summary of the field. Rather, the objective is to provide a conceptual foundation and physically illuminating examples that are useful to graduate students and researchers entering this subject. There are exercise problems and references for further reading throughout the notes.

# 1  Introduction

Quantum systems coupled to an environment or subjected to continuous measurements or classical noise can be described by mixed states that evolve under a Markovian (time-local) master equation in appropriate limits. The study of such open quantum systems has its roots in quantum optics [1]. In recent decades it has also become increasingly relevant for many-body systems due to remarkable advances in trapping atoms and photons in controllable setups [2]. A central theme of this rapidly developing frontier is that dissipation can actually be a

resource [3–7], i.e., one can engineer dissipation or measurements to control the evolution of a quantum system [8], leading to nonequilibrium phases or stable manifolds that are useful for quantum information processing [6, 9]. More generally, nonunitary evolution produces new kinds of dynamical phenomena that are not found in Hamiltonian systems.

The aim of these lecture notes is to provide a conceptual framework and instructive case studies to develop intuition about such physical systems in a concise and readable format. The notes are based on a two-month course for first-year PhD students. Hence, we assume only a basic knowledge of undergraduate quantum mechanics and linear algebra, limiting technical details. For more in-depth discussions, we refer to one of the classic textbooks [1, 8, 10–12] or recent reviews focused on the experimental [2, 7] and/or theoretical [2, 7, 13–15] techniques. We also leave out several aspects of open quantum systems that are topics of current research, as mentioned briefly in the conclusion.

After a brief refresher on density matrices, we discuss how Markovianity constrains the form of the master equation and how its eigenvalues and eigenvectors govern the dynamics. Next, we present different physical scenarios where such dynamics is realized, including noisy Hamiltonians and continuous measurement protocols that lead to stochastic quantum trajectories. We continue on to the structure of steady states, discussing dark states, decoherence-free subspaces, as well as mixed steady states. We highlight the role of symmetries, how they differ from closed systems, and how they can be used to control the late-time dynamics. We conclude with some examples of novel out-of-equilibrium phenomena, including dissipative state preparation, freezing or steering a quantum state by frequent measurements (quantum Zeno effect), and nonequilibrium phase transitions at the ensemble and trajectory levels.

## 2   Refresher on density matrices

A density matrix describes our incomplete knowledge of a quantum state. It arises from ignoring or not having access to some degrees of freedom or mesurement record in a closed system. In such cases, the best description of the state becomes statistical: One can at most say that the (sub)system is likely to be in state $|\psi_1\rangle$ with probability $p_1$, in state $|\psi_2\rangle$ with probability $p_2$, etc. The expectation value of any observable $\hat{O}$, averaged over such an ensemble of (pure) states, is given by

$$\langle \hat{O} \rangle = \sum_i p_i \langle \psi_i | \hat{O} | \psi_i \rangle = \text{Tr}\Big( \hat{O} \sum_i p_i |\psi_i\rangle \langle \psi_i| \Big), \tag{1}$$

which can be calculated from the density operator (or density matrix in some basis)

$$\hat{\rho} = \sum_i p_i |\psi_i\rangle \langle \psi_i| . \tag{2}$$

As $p_i \in [0, 1]$ and $\sum_i p_i = 1$, this is a positive semidefinite, Hermitian operator with $\text{Tr}(\hat{\rho}) = 1$. The positive semidefiniteness means that $\langle \psi | \hat{\rho} | \psi \rangle \geq 0$ for any state $|\psi\rangle$, or equivalently, that its eigenvalues are all nonnegative. The purity of $\hat{\rho}$ is measured by $\text{Tr}(\hat{\rho}^2) \in [0, 1]$. A state is pure, i.e., $\hat{\rho} = |\psi\rangle \langle \psi|$ for some state $|\psi\rangle$, if and only if $\text{Tr}(\hat{\rho}^2) = 1$. Otherwise, it is called a mixed state.

> Q1. Which one or more of the following are valid density matrices?
>
> $$\rho_1 = \frac{1}{2}\begin{pmatrix} 1 & 2 \\ 2 & 1 \end{pmatrix}, \quad \rho_2 = \frac{1}{2}\begin{pmatrix} 1 & -1 \\ 1 & 1 \end{pmatrix}, \quad \rho_3 = \frac{1}{2}\begin{pmatrix} 1 & -i/2 \\ i/2 & 1 \end{pmatrix}.$$

Note that, for a given density matrix, the decomposition in terms of pure states is generally not unique. While it can always be written as Eq. (2) in its eigenbasis, it may also have the same form in terms of other, non-orthogonal states. For example, the state $\hat{\rho} = \frac{3}{4}|0\rangle\langle 0| + \frac{1}{4}|1\rangle\langle 1|$ is identical to $\hat{\rho} = \frac{1}{2}|\phi_+\rangle\langle\phi_+| + \frac{1}{2}|\phi_-\rangle\langle\phi_-|$, where $|\phi_\pm\rangle := \frac{\sqrt{3}}{2}|0\rangle \pm \frac{1}{2}|1\rangle$. Thus, different ensembles of pure states can have the same density matrix [16].

> Q2. What observables (if any) can be measured to differentiate the following ensembles of qubits?
> 1) each qubit in the pure state $\frac{1}{\sqrt{2}}(|0\rangle + |1\rangle)$,
> 2) equal statistical mixture of $|0\rangle$ and $|1\rangle$,
> 3) equal statistical mixture of $\frac{1}{\sqrt{2}}(|0\rangle + |1\rangle)$ and $\frac{1}{\sqrt{2}}(|0\rangle - |1\rangle)$.

A very useful concept is that of a reduced density matrix. Suppose a system composed of parts $A$ and $B$ has the density matrix $\hat{\rho}$. If we perform measurements only on subsystem $A$, the expectation of any such operator $\hat{O}_A$ can be written as

$$\langle \hat{O}_A \rangle = \text{Tr}\big[(\hat{O}_A \otimes \hat{1}_B)\hat{\rho}\big] = \text{Tr}_A\text{Tr}_B\big[(\hat{O}_A \otimes \hat{1}_B)\hat{\rho}\big] = \text{Tr}_A\big[\hat{O}_A(\text{Tr}_B\hat{\rho})\big] = \text{Tr}_A\big[\hat{O}_A\hat{\rho}_A\big], \qquad (3)$$

where $\hat{\rho}_A := \text{Tr}_B(\hat{\rho})$ is called the reduced density matrix of $A$. One can readily check that the partial trace preserves Hermiticity, trace, and positivity. Even if $\hat{\rho}$ is a pure state, $\hat{\rho}_A$ is mixed if there is any entanglement between $A$ and $B$. For example, consider a Bell state $\hat{\rho} = |\psi\rangle\langle\psi|$ where $|\psi\rangle = \frac{1}{\sqrt{2}}(|0_A 1_B\rangle + |1_A 0_B\rangle)$. This gives $\hat{\rho}_A = \frac{1}{2}(|0\rangle\langle 0| + |1\rangle\langle 1|)$, which is a completely mixed state. The loss of purity represents the loss of information by "tracing out" or ignoring the state of $B$. This information loss, measured by the von Neumann entropy of $\hat{\rho}_A$ (or $\hat{\rho}_B$), gives the entanglement entropy of $A$ and $B$, $S := -\text{Tr}(\hat{\rho}_A \log \hat{\rho}_A) = -\sum_i p_i \log p_i$, where $p_i$ are the eigenvalues of $\hat{\rho}_A$. As $\sum_i p_i = 1$, $S$ is maximized when all eigenvalues are equal. This is true for the Bell state as all information is shared between $A$ and $B$. Note this interpretation breaks down if $\hat{\rho}$ itself is mixed. An extreme example is $\hat{\rho} \propto \hat{1}$, for which $\hat{\rho}_A \propto \hat{1}_A$ and $S$ is maximum even though $A$ and $B$ are not entangled. (There is no easy quantifier of entanglement for general mixed states; commonly used measures include negativity [17], concurrence for two qubits [18], or the more general but hard to compute entanglement of formation [19].)

> Q3. Show that $\text{Tr}(\hat{\rho}_A \log \hat{\rho}_A) = \text{Tr}(\hat{\rho}_B \log \hat{\rho}_B)$ holds if $\hat{\rho}$ is pure, but is not necessarily true if $\hat{\rho}$ is mixed.

# 3  Markovian master equation

## 3.1  Transformations of the density matrix

We are interested in the time evolution of the density matrix, which is nothing but a dynamical map. Any valid transformation or map $\Phi$ of a density matrix must preserve its defining properties: (i) Hermiticity, (ii) unit trace, and (iii) positive semidefiniteness (i.e., nonnegative eignevalues). Furthermore, one is often interested in the dynamics of a subsystem embedded in an environment, where the full density matrix $\hat{\rho}$ evolves under $\Phi \otimes I$. Then the positivity has to be satisfied for $\hat{\rho}$, not just for the subsystem. This stronger condition is called complete positivity. A simple example of a transformation that is positive but not completely positive is partial transposition: Consider a state of two qubits $\hat{\rho} = \sum_{i,j,i',j'=0,1} \rho_{(i,j),(i',j')} |i\rangle\langle i'| \otimes |j\rangle\langle j'|$, where $(i,j)$ labels the rows and $(i',j')$ labels the columns. Transposing $i$ with $i'$ changes

$$\rho_1 = \frac{1}{2} \begin{bmatrix} 1 & 0 & 0 & 1 \\ 0 & 0 & 0 & 0 \\ 0 & 0 & 0 & 0 \\ 1 & 0 & 0 & 1 \end{bmatrix} \quad \text{to} \quad \rho_2 = \frac{1}{2} \begin{bmatrix} 1 & 0 & 0 & 0 \\ 0 & 0 & 1 & 0 \\ 0 & 1 & 0 & 0 \\ 0 & 0 & 0 & 1 \end{bmatrix}. \tag{4}$$

Although the reduced density matrices for both $\rho_1$ and $\rho_2$ are proportional to the identity, $\rho_2$ is no longer positive semidefinite (it has an eignevalue of $-1/2$). As we will see below, complete positivity, Hermiticity and trace preservation strongly constrain the form of the allowed transformations.

---
Q4.  Show that a unitary transformation $\Phi$ satisfies all of the above conditions.

---

Let us focus on linear maps. Both unitary evolution and the partial trace operation preserve linearity, so this is a natural assumption. Later we will consider continuously monitored systems, where the evolution conditioned on the measurement record will be nonlinear.

## 3.2  Choi-Kraus theorem

Let $\mathcal{B}(\mathcal{H})$ be the set of all bounded linear operators on a Hilbert space $\mathcal{H}$. A linear map $\Phi$ on $\mathcal{B}(\mathcal{H})$ is completely positive (CP) if and only if

$$\Phi(\hat{\rho}) = \sum_\mu \hat{K}_\mu \hat{\rho} \hat{K}_\mu^\dagger \tag{5}$$

for all $\hat{\rho} \in \mathcal{B}(\mathcal{H})$. The operators $\hat{K}_\mu \in \mathcal{B}(\mathcal{H})$ — which are not necessarily unitary — are called Kraus operators. This theorem was derived independently by Kraus [20] and Choi [21].

*Proof.* First, let's check that if $\Phi$ has the above form, then it is CP, i.e., $(\Phi \otimes I)(\hat{\rho}) \geq 0 \ \ \forall \hat{\rho} \geq 0$. We can write $\hat{\rho}$ in its diagonal basis, $\hat{\rho} = \sum_n p_n |p_n\rangle\langle p_n|$ where $p_n \geq 0$. For any state $|\psi\rangle$,

$$\langle\psi|(\Phi \otimes I)(\hat{\rho})|\psi\rangle = \sum_{\mu,n} p_n \langle\psi|\hat{K}_\mu \otimes \hat{1}|p_n\rangle\langle p_n|\hat{K}_\mu^\dagger \otimes \hat{1}|\psi\rangle = \sum_{\mu,n} p_n |\langle\psi|\hat{K}_\mu \otimes \hat{1}|p_n\rangle|^2 \geq 0 \,. \tag{6}$$

Second, we have to show that if $\Phi$ is CP, it has the Kraus form. Let us act $\Phi \otimes I$ on the maximally entangled state $|\Omega\rangle = \sum_i |i\rangle \otimes |i\rangle$ in a doubled Hilbert space $\mathcal{H} \otimes \mathcal{H}$, where $\{|i\rangle\}$ is an orthonormal basis in $\mathcal{H}$. This defines the Choi matrix

$$C_\Phi := (\Phi \otimes I)(|\Omega\rangle\langle\Omega|) = \sum_{i,j} \Phi(|i\rangle\langle j|) \otimes |i\rangle\langle j| \,. \tag{7}$$

As $C_\Phi \geq 0$, we can write $C_\Phi = \sum_\mu |\psi_\mu\rangle\langle\psi_\mu|$ for some unnormalized orthogonal eigenvectors $|\psi_\mu\rangle$. Each of these vectors can be expressed as $|\psi_\mu\rangle = \sum_{i,j} K_{i,j}^\mu |i\rangle \otimes |j\rangle = \sum_j \hat{K}_\mu |j\rangle \otimes |j\rangle$, where $\hat{K}_\mu := \sum_{i,j} K_{i,j}^\mu |i\rangle\langle j|$. Thus,

$$C_\Phi = \sum_{i,j} \sum_\mu \hat{K}_\mu |i\rangle\langle j| \hat{K}_\mu^\dagger \otimes |i\rangle\langle j| \,. \tag{8}$$

Comparing Eqs. (7) and (8) gives $\Phi(|i\rangle\langle j|) = \sum_\mu \hat{K}_\mu |i\rangle\langle j| \hat{K}_\mu^\dagger \;\; \forall i,j$. Hence, $\Phi(\hat{\rho}) = \sum_\mu \hat{K}_\mu \hat{\rho} \hat{K}_\mu^\dagger$ $\forall \hat{\rho} \in \mathcal{B}(\mathcal{H})$. $\blacksquare$

### 3.2.1 Trace and Hermiticity preservation

From Eq. (5),

$$\mathrm{Tr}\,\Phi(\hat{\rho}) = \sum_\mu \mathrm{Tr}\big(\hat{K}_\mu \hat{\rho} \hat{K}_\mu^\dagger\big) = \sum_\mu \mathrm{Tr}\big(\hat{\rho} \hat{K}_\mu^\dagger \hat{K}_\mu\big) = \mathrm{Tr}\left(\hat{\rho} \sum_\mu \hat{K}_\mu^\dagger \hat{K}_\mu\right) \,. \tag{9}$$

Hence, for CP maps, trace preservation is equivalent to the condition $\sum_\mu \hat{K}_\mu^\dagger \hat{K}_\mu = \hat{\mathbb{1}}$. Such maps are called completely positive and trace preserving (CPTP). It is also clear from Eq. (5) that CP maps preserve Hermiticity, i.e., $\Phi(\hat{\rho}^\dagger) = \Phi(\hat{\rho})^\dagger$ if $\hat{\rho}^\dagger = \hat{\rho}$.

### 3.2.2 Nonuniqueness of Kraus operators

The decomposition in Eq. (5) is not unique. One can define a new set of Kraus operators $\hat{K}_\mu' = \sum_\nu U_{\mu,\nu} \hat{K}_\nu$ via a unitary matrix $U$ which would give the same mapping:

$$\sum_\mu \hat{K}_\mu' \hat{\rho} \hat{K}_\mu'^\dagger = \sum_{\nu,\nu'} \left(\sum_\mu U_{\mu,\nu} U_{\mu,\nu'}^*\right) \hat{K}_\nu \hat{\rho} \hat{K}_{\nu'}^\dagger = \sum_{\nu,\nu'} (U^\dagger U)_{\nu,\nu'}^* \hat{K}_\nu \hat{\rho} \hat{K}_{\nu'}^\dagger = \sum_\mu \hat{K}_\mu \hat{\rho} \hat{K}_\mu^\dagger \,. \tag{10}$$

The unitary mixing also preserves the trace condition, as $\sum_\mu \hat{K}_\mu'^\dagger \hat{K}_\mu' = \sum_\mu \hat{K}_\mu^\dagger \hat{K}_\mu$.

## 3.3 Lindblad master equation

We can use the Choi-Kraus theorem to derive the most general form of a Markovian master equation for a continuous time evolution of a density matrix $\hat{\rho}(t)$. Markovianity requires that $\hat{\rho}(t + \mathrm{d}t)$ depends only on $\hat{\rho}(t)$ and not on past values. Hence, the infinitesimal change over time $\mathrm{d}t$ must have the CPTP form

$$\hat{\rho}(t + \mathrm{d}t) = \sum_\mu \hat{K}_\mu \hat{\rho}(t) \hat{K}_\mu^\dagger \,, \tag{11}$$

where $\sum_\mu \hat{K}_\mu^\dagger \hat{K}_\mu = 1$. However, the right-hand side of Eq. (11) should be $\hat{\rho}(t)$ plus something proportional to $\mathrm{d}t$. This is achieved by setting all but one Kraus operators proportional to $\sqrt{\mathrm{d}t}$,

i.e., $\hat{K}_{\mu\neq 0} = \hat{L}_\mu \sqrt{\mathrm{d}t}$. The remaining Kraus operator $\hat{K}_0$ must satisfy $\hat{K}_0^\dagger \hat{K}_0 = \hat{1} - \sum_{\mu\neq 0} \hat{K}_\mu^\dagger \hat{K}_\mu$ up to $O(\mathrm{d}t)$, which gives the form

$$\hat{K}_0 = \hat{1} - \left( i\hat{H} + \frac{1}{2} \sum_{\mu\neq 0} \hat{L}_\mu^\dagger \hat{L}_\mu \right) \mathrm{d}t \,, \tag{12}$$

where $\hat{H}$ is an arbitrary Hermitian operator. Using these expressions in Eq. (11) and rearranging terms yield

$$\frac{\mathrm{d}\hat{\rho}}{\mathrm{d}t} = -i[\hat{H}, \hat{\rho}] + \sum_{\mu\neq 0} \left( \hat{L}_\mu \hat{\rho} \hat{L}_\mu^\dagger - \frac{1}{2} \hat{L}_\mu^\dagger \hat{L}_\mu \hat{\rho} - \frac{1}{2} \hat{\rho} \hat{L}_\mu^\dagger \hat{L}_\mu \right) \,. \tag{13}$$

Thus, $\hat{H}$ plays the role of the system Hamiltonian and $\hat{L}_\mu$ encode various dissipation (i.e. nonunitary) processes. Equation (13) was first derived independently by Gorini, Kossakowski, Sudarshan [22] and Lindblad [23], and is thus called the Gorini-Kossakowski-Sudarshan-Lindblad (GKS-L) master equation or just the Lindblad master equation for short. The operators $\hat{L}_\mu$ are called Lindblad (or jump) operators. As the summation in Eq. (13) is over all $\hat{L}_\mu$, we will omit the $\mu \neq 0$ label and simply write $\sum_\mu$ from now on (with $\mu = 1, 2, 3, \dots$).

Note the CPTP condition does not require the operators $\hat{H}$ and $\hat{L}_\mu$ to be time-independent. However, this is often assumed for simplicity or based on physical reasoning, e.g., a lossy optical cavity that is not changing with time may be reasonably described by a Lindblad operator $\hat{L}_1 = \sqrt{\kappa}\,\hat{a}$, where $\hat{a}$ is the photon annihilation operator and $\kappa$ is the loss rate. The square root appears because Eq. (13) is quadratic in $\hat{L}_\mu$.

### 3.3.1 Decay and recycling terms

A different way to express the master equation is to write $\hat{K}_0 := 1 - i\hat{H}_{\mathrm{eff}}\mathrm{d}t$, where

$$\hat{H}_{\mathrm{eff}} = \hat{H} - \frac{i}{2} \sum_\mu \hat{L}_\mu^\dagger \hat{L}_\mu \,. \tag{14}$$

Then Eq. (13) becomes

$$\frac{\mathrm{d}\hat{\rho}}{\mathrm{d}t} = -i\left( \hat{H}_{\mathrm{eff}}\hat{\rho} - \hat{\rho}\hat{H}_{\mathrm{eff}}^\dagger \right) + \sum_\mu \hat{L}_\mu \hat{\rho} \hat{L}_\mu^\dagger \,. \tag{15}$$

The first term describes the evolution under a non-Hermitian effective Hamiltonian, which causes $\mathrm{Tr}\,\hat{\rho}(t)$ to decay over time. This loss of probability is exactly compensated or "recycled" by the second term, which ensures that $\frac{\mathrm{d}}{\mathrm{d}t}\mathrm{Tr}(\hat{\rho}) = 0$. As an illustration, consider the case of spontaneous emission described by the Lindblad operator $\hat{L} = |g\rangle\langle e|$, where $|g\rangle$ and $|e\rangle$ denote the ground and excited states of an atom, respectively. Then $\hat{H}_{\mathrm{eff}} = \hat{H} - (i/2)|e\rangle\langle e|$ causes the excited-state population to decay, whereas the recycling term $\hat{L}\hat{\rho}\hat{L}^\dagger = |g\rangle\langle e|\hat{\rho}|e\rangle\langle g|$ pumps the lost population into the ground state. These two components of the dynamics will acquire physical meaning when we discuss quantum trajectories in Sec. 4.2.

As Eq. (15) is linear in $\hat{H}_{\mathrm{eff}}$, it suffices to have a single Kraus operator $\hat{K}_0$ that is not proportional to $\sqrt{\mathrm{d}t}$. This is not the case, however, for $\hat{K}_{\mu\neq 0}$ as the recycling term is quadratic in $\hat{L}_\mu$. Crucially, it means the dynamics generated by two Lindblad operators $\hat{L}_1$ and $\hat{L}_2$ is different from that generated by a single Lindblad operator $\hat{L}_1 + \hat{L}_2$.

### 3.3.2 Non-diagonal form

Suppose we write $\hat{L}_\mu = \sum_\nu A_{\nu,\mu} \hat{M}_\nu$, where $\hat{M}_\nu$ are a new set of linearly independent operators. Substituting this expansion into Eq. (13) gives

$$\frac{d\hat{\rho}}{dt} = -i[\hat{H}, \hat{\rho}] + \sum_{\nu,\nu'} (AA^\dagger)_{\nu,\nu'} \left( \hat{M}_\nu \hat{\rho} \hat{M}_{\nu'}^\dagger - \frac{1}{2} \hat{M}_{\nu'}^\dagger \hat{M}_\nu \hat{\rho} - \frac{1}{2} \hat{\rho} \hat{M}_{\nu'}^\dagger \hat{M}_\nu \right) . \tag{16}$$

The "Kossakowski matrix" $C = AA^\dagger$ is generally not diagonal. However, it is Hermitian and positive semidefinite. One can always reduce such a nondiagonal Lindblad equation to a diagonal form by writing $C = UDU^\dagger$ where $D$ is a diagonal matrix with eigenvalues of $C$ and $U$ is a unitary matrix composed of its eigenvectors. Then the equation becomes diagonal in terms of the Lindblad operators $\hat{L}_\mu' := \sum_\nu U_{\nu,\mu} \sqrt{D_{\mu,\mu}} \hat{M}_\nu$. (Note that $\hat{L}_\mu$ and $\hat{L}_\mu'$ are not necessarily the same set, but are related by a unitary transformation.)

---

Q5. Find a diagonal form for $C = \{\{1, \epsilon\}, \{\epsilon, 1\}\}$ and $\hat{M}_\nu \in \{\hat{\sigma}^+, \hat{\sigma}^-\}$ for a qubit, where $\epsilon \in [0, 1]$. Is the diagonal form unique for $\epsilon = 0$ and for $\epsilon = 1$?

---

### 3.3.3 Invariance under rescaling

To see how the dynamics is altered by a rescaling of the jump operators, we write Eq. (13) as

$$\frac{d\hat{\rho}}{dt} = -i[\hat{H}, \hat{\rho}] + \sum_\mu \mathcal{D}[\hat{L}_\mu]\hat{\rho} , \tag{17}$$

$$\text{with} \quad \mathcal{D}[\hat{L}]\hat{\rho} := \hat{L}\hat{\rho}\hat{L}^\dagger - \frac{1}{2}\hat{L}^\dagger\hat{L}\hat{\rho} - \frac{1}{2}\hat{\rho}\hat{L}^\dagger\hat{L} . \tag{18}$$

Here, $\mathcal{D}[\hat{L}]$ is called the dissipator associated with $\hat{L}$. Clearly, $\mathcal{D}[a\hat{L}] = |a|^2 \mathcal{D}[\hat{L}]$. Thus, the Lindblad equation is also written as

$$\frac{d\hat{\rho}}{dt} = -i[\hat{H}, \hat{\rho}] + \sum_\mu \gamma_\mu \mathcal{D}[\hat{L}_\mu]\hat{\rho} \tag{19}$$

with dissipation rates $\gamma_\mu \geq 0$, which corresponds to the rescaling $\hat{L}_\mu \to \sqrt{\gamma_\mu}\hat{L}_\mu$. The equation is also manifestly invariant under a constant shift of the Hamiltonian $\hat{H}$. For $\hat{L} \to \hat{L} + a\hat{1}$,

$$\mathcal{D}[\hat{L} + a\hat{1}]\hat{\rho} = \mathcal{D}[\hat{L}]\hat{\rho} + \frac{1}{2}[a^*\hat{L} - a\hat{L}^\dagger, \hat{\rho}] . \tag{20}$$

The additional term on the right vanishes if $a^*\hat{L}$ is Hermitian. Else it can be absorbed into $\hat{H}$. Thus, the master equation is invariant under the joint transformation

$$\hat{L}_\mu \to \hat{L}_\mu + a_\mu\hat{1} , \quad \hat{H} \to \hat{H} + \frac{1}{2i}\sum_\mu \left( a_\mu^*\hat{L}_\mu - a_\mu\hat{L}_\mu^\dagger \right) . \tag{21}$$

### 3.3.4 Evolution of expectation values

From Eq. (13), the expectation of a time-independent operator $\hat{O}$ evolves as

$$\text{Tr}\left( \hat{O}\frac{d\hat{\rho}}{dt} \right) = -i\,\text{Tr}(\hat{O}\hat{H}\hat{\rho} - \hat{O}\hat{\rho}\hat{H}) + \sum_\mu \left[ \text{Tr}(\hat{O}\hat{L}_\mu\hat{\rho}\hat{L}_\mu^\dagger) - \frac{1}{2}\text{Tr}(\hat{O}\hat{L}_\mu^\dagger\hat{L}_\mu\hat{\rho}) - \frac{1}{2}\text{Tr}(\hat{O}\hat{\rho}\hat{L}_\mu^\dagger\hat{L}_\mu) \right]. \tag{22}$$

Using the cyclic property of trace, we get

$$\frac{\mathrm{d}}{\mathrm{d}t}\langle\hat{O}\rangle = \left\langle \mathrm{i}[\hat{H},\hat{O}] + \sum_{\mu}\left(\hat{L}_{\mu}^{\dagger}\hat{O}\hat{L}_{\mu} - \frac{1}{2}\hat{L}_{\mu}^{\dagger}\hat{L}_{\mu}\hat{O} - \frac{1}{2}\hat{L}_{\mu}^{\dagger}\hat{L}_{\mu}\hat{O}\right)\right\rangle \tag{23}$$

$$= \mathrm{i}\langle[\hat{H},\hat{O}]\rangle + \frac{1}{2}\sum_{\mu}\langle\hat{L}_{\mu}^{\dagger}[\hat{O},\hat{L}_{\mu}] + [\hat{L}_{\mu}^{\dagger},\hat{O}]\hat{L}_{\mu}\rangle. \tag{24}$$

In particular, if $\hat{O}$ is Hermitian, the right-hand side simplifies to

$$\frac{\mathrm{d}}{\mathrm{d}t}\langle\hat{O}\rangle = \mathrm{Im}\langle[\hat{O},\hat{H}]\rangle + \mathrm{Re}\sum_{\mu}\langle\hat{L}_{\mu}^{\dagger}[\hat{O},\hat{L}_{\mu}]\rangle. \tag{25}$$

It is tempting to think of Eq. (23) as giving the Heisenberg equation of motion for the operators themselves, not just their expectation values. However, this is misleading: One can check that $\hat{O} \propto \hat{1}$ would be a steady state of this equation, and as we will discuss later, generically the steady state is unique. This would mean all operators evolve toward $\hat{1}$ and commute at long times, violating canonical commutation relations. In fact, the correct operator evolution includes additional, stochastic terms that represent quantum fluctuations of the environment, leading to the formalism of quantum Langevin equations (Chap. 3 in [8,10]). This will become somewhat apparent when we discuss continuously monitored systems in Sec. 4.2.2 and 4.2.4, however we will largely limit our discussion to the Schrödinger picture.

### 3.3.5 Examples

Let us consider two types of Lindblad dynamics for a qubit with states $|0\rangle$ and $|1\rangle$. For simplicity we take $\hat{H} = 0$ and study a purely dissipative evolution with a single Lindblad operator.

First, take $\hat{L} = \sqrt{\gamma}\hat{\sigma}^{-} := \sqrt{\gamma}|0\rangle\langle 1|$, modeling spontaneous emission. The density matrix of a qubit can be written as

$$\hat{\rho} = \frac{1}{2}\begin{bmatrix} 1 + \langle\hat{\sigma}^{z}\rangle & \langle\hat{\sigma}^{x}\rangle - \mathrm{i}\langle\hat{\sigma}^{y}\rangle \\ \langle\hat{\sigma}^{x}\rangle + \mathrm{i}\langle\hat{\sigma}^{y}\rangle & 1 - \langle\hat{\sigma}^{z}\rangle \end{bmatrix}, \tag{26}$$

where $\hat{\sigma}^{x}, \hat{\sigma}^{y}, \hat{\sigma}^{z}$ are the Pauli spin operators. Using $[\hat{\sigma}^{z}, \sigma^{\pm}] = \pm 2\hat{\sigma}^{\pm}$ and $2\hat{\sigma}^{\pm}\hat{\sigma}^{\mp} = 1 \pm \hat{\sigma}^{z}$ in Eq. (24), we find

$$\frac{\mathrm{d}\langle\hat{\sigma}^{z}\rangle}{\mathrm{d}t} = -\gamma(1 + \langle\hat{\sigma}^{z}\rangle) \quad \text{and} \quad \frac{\mathrm{d}\langle\hat{\sigma}^{-}\rangle}{\mathrm{d}t} = -\frac{\gamma}{2}\langle\hat{\sigma}^{-}\rangle. \tag{27}$$

Thus, $\langle\hat{\sigma}^{z}\rangle$ approaches $-1$ as $e^{-\gamma t}$ and $\langle\hat{\sigma}^{x}\rangle$, $\langle\hat{\sigma}^{y}\rangle$ vanish as $e^{-\gamma t/2}$, so the qubit reaches the (pure) steady state $|0\rangle$.

Second, take $\hat{L} = \sqrt{\gamma}\hat{\sigma}^{z}$. From Eq. (24),

$$\frac{\mathrm{d}\langle\hat{\sigma}^{z}\rangle}{\mathrm{d}t} = 0 \quad \text{and} \quad \frac{\mathrm{d}\langle\hat{\sigma}^{-}\rangle}{\mathrm{d}t} = -2\gamma\langle\hat{\sigma}^{-}\rangle. \tag{28}$$

Here, the relative populations do not change, but the coherence vanishes as $e^{-2\gamma t}$. As coherence describes a definite phase relation between the states, this process is called dephasing. The steady state is purely diagonal and generically mixed. However, it is not unique as the initial populations are conserved. Such degeneracies are quite rare and arises here from the fact that $[\hat{\sigma}^{z}, \hat{L}] = [\hat{\sigma}^{z}, \hat{H}] = 0$ [see Eq. (25)]. We will discuss such conservation laws in Sec. 5.3.

These two examples also illustrate that Hermitian and non-Hermitian Lindblad operators produce different kinds of dynamics ("dephasing" versus "loss").

Q6. Show that the purity $\text{Tr}(\hat{\rho}^2)$ cannot increase with time if all of the Lindblad operators are Hermitian. Show this is not necessarily the case for a non-Hermitian Lindblad operator.

### 3.3.6 Lindbladian superoperator

The right-hand side of Eq. (17) can be thought of as a linear operator $\mathcal{L}$ acting on $\hat{\rho}$,

$$\frac{\mathrm{d}\hat{\rho}}{\mathrm{d}t} = \mathcal{L}\hat{\rho} \quad \text{where} \quad \mathcal{L} := -\mathrm{i}[\hat{H}, \cdot\,] + \sum_{\mu} \mathcal{D}[\hat{L}_{\mu}]. \tag{29}$$

As $\mathcal{L}$ acts on the space of operators, it is called a superoperator. To make use of this concept, it is helpful to write the density matrix $\rho_{i,j}$ as a vector $|\hat{\rho}\rangle\rangle$ by flattening the indices $(i,j)$ into a single index $k = (i-1)D + j$, where $D$ is the Hilbert space dimension,

$$|\hat{\rho}\rangle\rangle = [\rho_{1,1} \ \rho_{1,2} \ \cdots \ \rho_{1,D} \ \rho_{2,1} \ \cdots \ \rho_{2,D} \ \cdots \ \rho_{D,D}]^T. \tag{30}$$

We denote this correspondence by $k \equiv (i,j)$. Then $\mathcal{L}$ can be written as a $D^2 \times D^2$ matrix acting on $|\hat{\rho}\rangle\rangle$ by noting that

$$(\mathbf{A}\rho\mathbf{B})_{i,j} = \sum_{i',j'} \mathbf{A}_{i,i'} \rho_{i',j'} \mathbf{B}^T_{j,j'} = \sum_{k'} (\mathbf{A} \otimes \mathbf{B}^T)_{k,k'} \rho_{k'}, \tag{31}$$

where $k \equiv (i,j)$ and $k' \equiv (i',j')$, and $\mathbf{A} \otimes \mathbf{B}$ is the Kronecker product of two matrices $\mathbf{A}$ and $\mathbf{B}$. Thus, from Eq. (15), the superoperator matrix can be constructed as

$$\mathcal{L} = \mathrm{i}\big(\mathbf{I} \otimes \mathbf{H}^*_{\text{eff}} - \mathbf{H}_{\text{eff}} \otimes \mathbf{I}\big) + \sum_{\mu} \mathbf{L}_{\mu} \otimes \mathbf{L}^*_{\mu}. \tag{32}$$

The time-evolved state is then $|\hat{\rho}(t)\rangle\rangle = e^{\mathcal{L}t}|\rho(0)\rangle\rangle$, which is the analog of $|\psi(t)\rangle = e^{-\mathrm{i}\mathbf{H}t}|\psi(0)\rangle$ for pure states. Therefore, the dynamics is governed by the eigenvalues and eigenvectors of the generator $\mathcal{L}$.

Note, however, that $\mathcal{L}$ is generally not Hermitian. In fact, from Eq. (32),

$$\mathcal{L}^{\dagger} = \mathrm{i}\big(\mathbf{H}^{\dagger}_{\text{eff}} \otimes \mathbf{I} - \mathbf{I} \otimes \mathbf{H}^T_{\text{eff}}\big) + \sum_{\mu} \mathbf{L}^{\dagger}_{\mu} \otimes \mathbf{L}^T_{\mu}. \tag{33}$$

Using $\hat{H}_{\text{eff}} = \hat{H} - (\mathrm{i}/2)\sum_{\mu} \hat{L}^{\dagger}_{\mu}\hat{L}_{\mu}$ and the correspondence in Eq. (31), one can write

$$\mathcal{L}^{\dagger}\hat{O} = \mathrm{i}[\hat{H}, \hat{O}] + \sum_{\mu}\left(\hat{L}^{\dagger}_{\mu}\hat{O}\hat{L}_{\mu} - \frac{1}{2}\hat{L}^{\dagger}_{\mu}\hat{L}_{\mu}\hat{O} - \frac{1}{2}\hat{O}\hat{L}^{\dagger}_{\mu}\hat{L}_{\mu}\right). \tag{34}$$

Comparing Eq. (13) and (34) shows that $\mathcal{L} = \mathcal{L}^{\dagger}$ requires $\hat{H} = 0$ and $\hat{L}_{\mu} = \hat{L}^{\dagger}_{\mu} \ \forall \mu$, which is rather special. Hence, the eigenvalues of $\mathcal{L}$ are generally complex. Note also that Eq. (34) is just the right-hand side of Eq. (23), which means

$$\frac{\mathrm{d}\langle\hat{O}\rangle}{\mathrm{d}t} = \langle\mathcal{L}^{\dagger}\hat{O}\rangle. \tag{35}$$

Q7.   Consider the operator evolution $\mathrm{d}\hat{O}/\mathrm{d}t = \mathcal{L}^{\dagger}\hat{O}$ which correctly predicts the expectation values [Eq. (35)]. If $\mathcal{L}^{\dagger}$ has a unique steady state, show that all operators evolve toward $\hat{1}$ under $\mathcal{L}^{\dagger}$. How can they have different expectation values in steady state?

It is instructive to see how Eq. (35) comes about from more basic algebraic structures. First, the standard inner product of two vectorised operators is given by

$$\langle\langle\hat{A}|\hat{B}\rangle\rangle = \sum_{i,j} A^*_{i,j} B_{i,j} = \mathrm{Tr}(\hat{A}^{\dagger}\hat{B}) , \tag{36}$$

which is also called the Hilbert-Schmidt inner product of $\hat{A}$ and $\hat{B}$. Second, from Eq. (13), $(\mathcal{L}\hat{\rho})^{\dagger} = \mathcal{L}\hat{\rho}^{\dagger}$. Using these two identities, one finds

$$\mathrm{Tr}[(\mathcal{L}\hat{\rho})\hat{O}] = \langle\langle\mathcal{L}\hat{\rho}^{\dagger}|\hat{O}\rangle\rangle = \langle\langle\hat{O}|\mathcal{L}|\hat{\rho}^{\dagger}\rangle\rangle^* = \langle\langle\hat{\rho}^{\dagger}|\mathcal{L}^{\dagger}|\hat{O}\rangle\rangle = \mathrm{Tr}[\hat{\rho}(\mathcal{L}^{\dagger}\hat{O})] , \tag{37}$$

which is the same as Eq. (35) if $\hat{O}$ is time independent. A word of caution about the abuse of notation here: The adjoint that converts $|\cdot\rangle\rangle$ to $\langle\langle\cdot|$ and acts on superoperators is defined on the space of vectorised operators and is different from the operator adjoint, i.e., $|\hat{\rho}\rangle\rangle^{\dagger} = \langle\langle\hat{\rho}|$, not $\langle\langle\hat{\rho}^{\dagger}|$. Similarly, $|\mathcal{L}\hat{\rho}\rangle\rangle^{\dagger} = \langle\langle\hat{\rho}\mathcal{L}^{\dagger}|$ but $(\mathcal{L}\hat{\rho})^{\dagger} \neq \hat{\rho}\mathcal{L}^{\dagger}$ or $\hat{\rho}^{\dagger}\mathcal{L}^{\dagger}$.

Q8.   Show that for $\hat{H} = 0$ and $\hat{L}^{\dagger}_{\mu} = \hat{L}_{\mu} \; \forall\mu$, all eigenvalues of $\mathcal{L}$ are real and nonpositive.

### 3.3.7   Spectrum

As $\mathcal{L}$ is generically non-Hermitian, it has a right eigenvector and a left eigenvector for each eigenvalue $\lambda$ (this is because the characteristic equations of a matrix and its transpose are the same [24]),

$$\mathcal{L}|\hat{r}_{\lambda}\rangle\rangle = \lambda|\hat{r}_{\lambda}\rangle\rangle \tag{38}$$

$$\text{and} \quad \langle\langle\hat{l}_{\lambda}|\mathcal{L} = \lambda\langle\langle\hat{l}_{\lambda}| . \tag{39}$$

For denegerate eigenvalues, the eigenvectors will have another label, e.g., $\mathcal{L}|\hat{r}_{\lambda,\alpha}\rangle\rangle = \lambda|\hat{r}_{\lambda,\alpha}\rangle\rangle$. There are several points to note here. (1) The adjoint of Eq. (39) is $\mathcal{L}^{\dagger}|\hat{l}_{\lambda}\rangle\rangle = \lambda^*|\hat{l}_{\lambda}\rangle\rangle$. Thus, the left eigenvectors of $\mathcal{L}$ are right eigenvectors of $\mathcal{L}^{\dagger}$, which are generally distinct from $|\hat{r}_{\lambda}\rangle\rangle$. (2) As $(\mathcal{L}\hat{\rho})^{\dagger} = \mathcal{L}\hat{\rho}^{\dagger}$, the operator adjoint of Eq. (38) gives $\mathcal{L}|\hat{r}^{\dagger}_{\lambda}\rangle\rangle = \lambda^*|\hat{r}^{\dagger}_{\lambda}\rangle\rangle$, so the eigenvalues are either real (in which case $\hat{r}^{\dagger}_{\lambda} = \hat{r}_{\lambda}$ if $\lambda$ is nondegenerate) or come in complex conjugate pairs. (3) Together, (1) and (2) imply that $\mathcal{L}$ and $\mathcal{L}^{\dagger}$ share the same spectrum. (4) From Eq. (34), $\mathcal{L}^{\dagger}\hat{1} = 0$, i.e., $\hat{1}$ is a steady state of $\mathcal{L}^{\dagger}$, so $\mathcal{L}$ must also have a steady state. (5) If $\hat{L}^{\dagger}_{\mu} = \hat{L}_{\mu} \; \forall\mu$, then from Eq. (13), $\hat{1}$ is also a steady state of $\mathcal{L}$. Thus, dephasing-type dynamics generically leads to an infinite-temperature steady state (irrespective of $\hat{H}$). For non-Hermitian Lindblad operators, the steady state of $\mathcal{L}$ is nonuniversal.

Let us now come to the orthogonality and completeness properties. From Eqs. (38) and (39), $\langle\langle\hat{l}_{\lambda}|\mathcal{L}|\hat{r}_{\lambda'}\rangle\rangle = \lambda\langle\langle\hat{l}_{\lambda}|\hat{r}_{\lambda'}\rangle\rangle = \lambda'\langle\langle\hat{l}_{\lambda}|\hat{r}_{\lambda'}\rangle\rangle \implies$

$$(\lambda - \lambda')\langle\langle\hat{l}_{\lambda}|\hat{r}_{\lambda'}\rangle\rangle = 0 , \tag{40}$$

or $\langle\langle\hat{l}_\lambda|\hat{r}_{\lambda'}\rangle\rangle = 0$ unless $\lambda = \lambda'$, i.e., the left and right eigenvectors are bi-orthogonal, which holds more generally for non-Hermitian matrices. We can normalize the states such that

$$\langle\langle\hat{l}_\lambda|\hat{r}_{\lambda'}\rangle\rangle = \delta_{\lambda,\lambda'} \, , \tag{41}$$

which can be generalized in the presence of degeneracies. Even with this normalization, however, the scale of $\hat{r}_\lambda$ and $\hat{l}_\lambda$ are not set individually. Taking $\hat{l}_0 = \hat{1}$ gives $\langle\langle\hat{1}|\hat{r}_\lambda\rangle\rangle = \text{Tr}(\hat{r}_\lambda) = \delta_{\lambda,0}$, which implies that all non-steady right eigenvectors are traceless and do not represent physical states on their own, although they can be added to $\hat{r}_0$ to describe distinct physical states. This is an important distinction from Hamiltonian eigenstates. Note the left eigenvectors are generally not traceless, except when $\hat{1}$ is also a right eigenstate (as in a dephasing dynamics). Similar features arise in classical rate equations governing a Markov process or a Fokker-Planck equation governing a probability distribution. In fact, the Lindblad equation can be formulated as an equation of motion for a quasiprobability distribution in phase space using the Wigner-Moyal correspondence [25–27], which is particularly useful for analyzing classical limits.

For diagonalizable, finite-dimensional Lindbladians, the right (as well as the left) eigenvectors form a complete basis. Thus, any operator $\hat{\rho}$ can be expanded as $|\hat{\rho}\rangle\rangle = \sum_\lambda c_\lambda|\hat{r}_\lambda\rangle\rangle$, where $c_\lambda = \langle\langle\hat{l}_\lambda|\hat{\rho}\rangle\rangle$ from Eq. (41). This is equivalent to the completeness relation

$$\sum_\lambda |\hat{r}_\lambda\rangle\rangle\langle\langle\hat{l}_\lambda| = \mathbb{1} \, . \tag{42}$$

Using this relation, one can write $e^{\mathcal{L}t} = \sum_\lambda e^{\lambda t}|\hat{r}_\lambda\rangle\rangle\langle\langle\hat{l}_\lambda|$, which gives the time-evolving state

$$|\hat{\rho}(t)\rangle\rangle = |\hat{r}_0\rangle\rangle + \sum_{\lambda\neq 0} c_\lambda e^{\lambda t}|\hat{r}_\lambda\rangle\rangle \, , \tag{43}$$

with $c_\lambda = \langle\langle\hat{l}_\lambda|\hat{\rho}(0)\rangle\rangle$. Here we have assumed a unique steady state and taken $\hat{l}_0 = \hat{1}$, such that $c_0 = \text{Tr}(\hat{\rho}) = 1$. If the dynamics is well behaved at long times—as is true in most physical situations—all nonzero eigenvalues in the summation must have nonpositive real parts. This is hard to show in general, but one can derive sufficient conditions on the Lindblad operators for which it holds [28]. The eigenvalue with the least negative (but nonzero) real part corresponds to the slowest decaying mode. This asymptotic decay rate is also called the gap and controls the late-time relaxation to steady state.

Figure 1 shows a sketch and simulations of the spectrum for simple systems. The wedge shape in Fig. 1(b) is characteristic of a classical fixed point, whereas the parabolic shapes in Fig. 1(c) indicate damped oscillations [29, 30].

---

Q9. Find the right and left eigenvectors of $\mathcal{D}[\hat{\sigma}^-]$ in terms of the Pauli operators. Check the biorthogonality and completeness relations. Do the same for $\mathcal{D}[\hat{\sigma}^z]$.

---

Q10. Compute the spectrum for a spin-$S$ with $\hat{H} = \hat{S}^x$ and $\hat{L} = \sqrt{\kappa/S}\,\hat{S}^-$ for $\kappa = 0.5$ and $\kappa = 2$ with $S = 10$. Plot the gap as a function of $1/S$ for both values of $\kappa$. What do you observe? For a reference on spin-$S$ operators see [31].

---

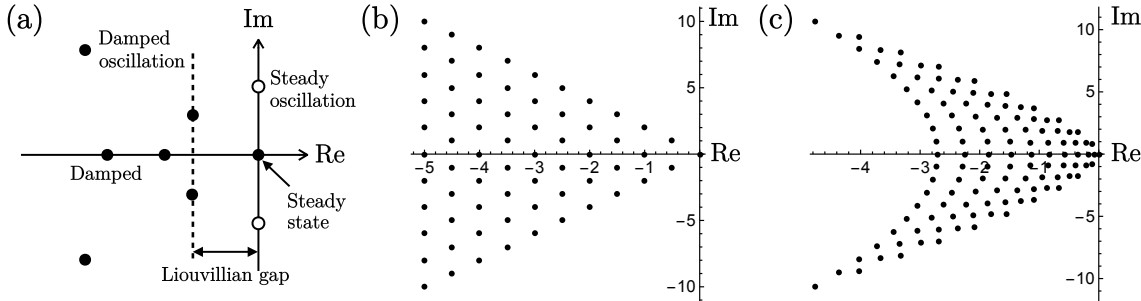

Figure 1: (a) Sketch showing possible eigenvalues of a Lindbladian. The steady oscillations are atypical. (b) Spectrum for a damped harmonic oscillator with $\hat{H} = \hat{a}^\dagger \hat{a}$ and $\hat{L} = \hat{a}$. (c) Spectrum for a spin-$S$ with $\hat{H} = \hat{S}^x$ and $\hat{L} = \sqrt{0.5/S}\,\hat{S}^-$ for $S = 5$.

# 4 Physical realizations

In the previous section, we derived the Lindblad master equation [Eq. (13)] by requiring the time evolution to be linear, CPTP, and Markovian. Let us now discuss several physical settings in which such dynamics is realized.

## 4.1 Noisy Hamiltonian

Consider a system evolving under a noisy Hamiltonian $\hat{H} = \hat{H}_0 + \eta(t)\hat{V}$, where $\eta(t)$ is a real-valued classical noise (meaning its a number, not an operator) and $\hat{H}_0$, $\hat{V}$ are Hermitian. Each realization of the noise generates a wave function $|\psi_\eta(t)\rangle$. The expectation value of any observable $\hat{O}$, averaged over this ensemble, is given by

$$\langle \hat{O} \rangle = \frac{1}{\mathcal{N}} \sum_\eta \langle \psi_\eta | \hat{O} | \psi_\eta \rangle = \mathrm{Tr}\left( \hat{O} \frac{1}{\mathcal{N}} \sum_\eta |\psi_\eta\rangle\langle\psi_\eta| \right) := \mathrm{Tr}(\hat{O}\hat{\rho}) \,, \tag{44}$$

where $\mathcal{N}$ is the number of realizations and $\hat{\rho}$ is the noise-averaged density matrix. Hence, it suffices to solve for $\hat{\rho}$ to track $\langle \hat{O} \rangle$ or, more generally, the ensemble average of any linear function of the density matrix $\hat{\rho}_\eta = |\psi_\eta\rangle\langle\psi_\eta|$. The equation of motion for $\hat{\rho}_\eta$ is

$$\frac{d\hat{\rho}_\eta}{dt} = -i[\hat{H}_0, \hat{\rho}_\eta(t)] - i[\hat{V}, \eta(t)\hat{\rho}_\eta(t)] \,. \tag{45}$$

Thus, we can write

$$\frac{d\hat{\rho}}{dt} = -i[\hat{H}_0, \hat{\rho}] - i\left[\hat{V}, \overline{\eta(t)\hat{\rho}_\eta(t)}\right] \,, \tag{46}$$

where the overline denotes noise averaging. Formally integrating Eq. (45) gives

$$\overline{\eta(t)\hat{\rho}_\eta(t)} = -i \int_0^t dt' \left( \left[\hat{H}_0, \overline{\eta(t)\hat{\rho}_\eta(t')}\right] + \left[\hat{V}, \overline{\eta(t)\eta(t')\hat{\rho}_\eta(t')}\right] \right) \,. \tag{47}$$

For a white noise, $\overline{\eta(t)\eta(t')} = \gamma\delta(t - t')$ where $\gamma \geq 0$ is the noise strength. Such a noise acts as an independent random variable at each instant. Thus, $\eta(t)$ is uncorrelated with $\rho_\eta(t')$ for $t > t'$, and the only nonvanishing term in Eq. (47) is $\overline{\eta(t)\eta(t')\hat{\rho}_\eta(t')} = \gamma\delta(t - t')\hat{\rho}(t)$. As the

delta function resides at the boundary of the integration domain, one has to consider it as a limiting form of a localized distribution, which gives $\int_0^t \mathrm{d}t' \delta(t - t') = 1/2$, yielding

$$\overline{\eta(t)\hat{\rho}_\eta(t)} = -\mathrm{i}(\gamma/2)[\hat{V}, \hat{\rho}(t)] \,. \tag{48}$$

Using this result in Eq. (46) gives

$$\frac{\mathrm{d}\hat{\rho}}{\mathrm{d}t} = -\mathrm{i}[\hat{H}_0, \hat{\rho}] - \frac{\gamma}{2}[\hat{V}, [\hat{V}, \hat{\rho}]] = -\mathrm{i}[\hat{H}_0, \hat{\rho}] + \gamma \mathcal{D}[\hat{V}]\hat{\rho} \,, \tag{49}$$

where we have used $\mathcal{D}[\hat{L}]\hat{\rho} = -\frac{1}{2}[\hat{L}, [\hat{L}, \hat{\rho}]]$ if $\hat{L}^\dagger = \hat{L}$ [see Eq. (18)]. Thus, the noise-averaged density matrix follows a Lindblad equation with Hamiltonian $\hat{H}_0$ and a Hermitian Lindblad operator $\hat{V}$. From Eq. (49), $\hat{\rho} \propto \hat{1}$ is a steady state, as is true generally for Hermitian Lindblad operators. Physically, the white noise pumps energy into the system, heating it to infinite temperature in the absence of a damping mechanism. If there are conserved quantities, one gets heating within each symmetry sector. In some cases such restricted infinite-temperature states can contain a large amount of entanglement or long-range correlations [32, 33].

> Q11.   Consider two spin-1/2s with the Hamiltonian $\hat{H} = \hat{S}_1 \cdot \hat{S}_2 + \eta(t)(\hat{S}_1^z + \hat{S}_2^z)$ where $\overline{\eta(t)\eta(t')} = \delta(t - t')$. Find the noise-averaged steady states.

By incorporating multiple independent white noises, $\hat{H} = \hat{H}_0 + \sum_j \eta_j(t)\hat{V}_j$, one can realize multiple Lindblad operators. While this approach is limited to Hermitian Lindblad operators, it is possible to implement equal pump and loss by using $\mathcal{D}[\hat{\sigma}^x] + \mathcal{D}[\hat{\sigma}^y] = 2\mathcal{D}[\sigma^+] + 2\mathcal{D}[\hat{\sigma}^-]$, as in [34]. In many-body systems this approach can also produce nonlocal Lindblad operators that cause dephasing in an entangled basis [35].

> Q12.   Consider a discretized version of Eq. (45), where the evolution in the $n$-th time step of duration $\Delta t$ occurs under $\hat{H} = \hat{H}_0 + \sum_j \hat{V}_j \eta_{j,n}/\sqrt{\Delta t}$, where $\eta$ is a discrete random variable with $\overline{\eta_{j,n}} = 0$ and $\overline{\eta_{i,m}\eta_{j,n}} = \delta_{i,j}\delta_{m,n}$. By finding the resulting change in the density matrix up to first order in $\Delta t$, show that the noise-averaged dynamics is governed by the Lindbladian $\mathcal{L} = -\mathrm{i}[\hat{H}_0, \cdot\,] + \sum_j \mathcal{D}[\hat{V}_j]$ for $\Delta t \to 0$. Note that the variable $\eta$ does not have to be Gaussian; it can be simply drawn from the set $\{-1, +1\}$, which gives a convenient way to engineer dephasing-type Lindblad dynamics.

For colored noises the dynamics is generally non-Markovian. To see this, consider a Gaussian colored noise $\eta(t)$. The noise averaging in Eq. (46) can be performed using Novikov's theorem [36], which gives

$$\overline{\eta(t)\hat{\rho}_\eta(t)} = \overline{\eta(t)}\,\overline{\hat{\rho}_\eta(t)} + \int_0^t \mathrm{d}t' \,\overline{\eta(t)\eta(t')}\,\overline{\frac{\delta\hat{\rho}_\eta(t')}{\delta\eta(t')}} \,, \tag{50}$$

where the functional derivative in the last term is defined in terms of the change $\delta\hat{\rho}_\eta(t)$ due to a change in $\delta\eta(t)$,

$$\delta\hat{\rho}_\eta(t) := \int_0^t \mathrm{d}t' \delta\eta(t') \frac{\delta\hat{\rho}_\eta(t')}{\delta\eta(t')} \,. \tag{51}$$

From Eq. (45) this derivative is simply $-\mathrm{i}[\hat{V}, \hat{\rho}_\eta]$. Thus, for a zero-mean stationary noise with $\overline{\eta(t)\eta(t')} = C(t - t')$,

$$\overline{\eta(t)\hat{\rho}_\eta(t)} = -\mathrm{i} \int_0^t \mathrm{d}t' C(t - t')[\hat{V}, \hat{\rho}(t')] . \tag{52}$$

Substituting this into Eq. (46), we find

$$\frac{\mathrm{d}\hat{\rho}}{\mathrm{d}t} = -\mathrm{i}[\hat{H}_0, \hat{\rho}] - \left[\hat{V}, \int_0^t \mathrm{d}t' C(t - t')[\hat{V}, \hat{\rho}(t')]\right] . \tag{53}$$

For a general correlation function $C$ this is non-Markovian. A special case is that of a Ornstein-Uhlenbeck noise (damped Brownian motion) for which $C(t - t') = \frac{\gamma}{2\tau} e^{-|t - t'|/\tau}$, where $\tau$ is the correlation time and $\gamma$ is the noise strength. In this case, Eq. (53) reduces to a pair of coupled differential equations:

$$\frac{\mathrm{d}\hat{\rho}}{\mathrm{d}t} = -\mathrm{i}[\hat{H}_0, \hat{\rho}] - [\hat{V}, \hat{\zeta}] , \tag{54}$$

$$\frac{\mathrm{d}\hat{\zeta}}{\mathrm{d}t} = -\frac{\hat{\zeta}}{\tau} + \frac{\gamma}{2\tau}[\hat{V}, \hat{\rho}] , \tag{55}$$

i.e., the dynamics of $(\hat{\rho}, \hat{\zeta})$ is Markovian. Here, $\tau$ sets the relaxation time for $\hat{\zeta}$. In the limit $\tau \to 0$, the noise becomes white and $\hat{\zeta}$ can be replaced by its instantaneous steady state, $\zeta = \frac{\gamma}{2}[\hat{V}, \hat{\rho}]$, which recovers Eq. (49). For an overview of other types of noises see [37].

## 4.2 Continuous measurement

In quantum optics setups it is common to continuously monitor a quantum system (e.g., detect photons coming out of a lossy cavity). As measurements are probabilistic, this process leads to an ensemble of *quantum trajectories* with jumps corresponding to the detection events [38]. Averaging over this ensemble can lead to a Lindblad master equation in appropriate limits. Then the trajectories constitute an *unraveling* of the master equation. One can also add feedbacks conditional on the measurement outcomes to steer the state of the system [8, 39–43].

### 4.2.1 Heuristic derivation

Let us consider the case of spontaneous emission by an atom with ground state $|g\rangle$ and excited state $|e\rangle$, where the latter has a lifetime of $1/\gamma$, beyond which it decays to $|g\rangle$ by emitting a photon of frequency equal to the gap between $|g\rangle$ and $|e\rangle$. Suppose we set up a photodetector that will detect any photon emitted by the atom. If the atom is initially in a superposition state $\alpha_g|g\rangle + \alpha_e|e\rangle$, it should eventually decay to the ground state. How many photons would we detect on average? Clearly, this count $N$ should depend on $\alpha_e$, going to zero for $\alpha_e = 0$ and one for $\alpha_e = 1$. In fact, from energy conservation we expect $N = |\alpha_e|^2 < 1$. Thus, there will be instances where no photon is detected, yet the atom relaxes to $|g\rangle$. How does this occur?

To make this question more concrete, let us define the projector onto the excited state, $\hat{P}_e := |e\rangle\langle e|$, whose expectation gives the excited-state population. The probability of detecting a photon over an infinitesimal interval $\mathrm{d}t$ is $p_1 = \gamma \mathrm{d}t \langle \hat{P}_e \rangle \ll 1$. If no photon is detected, how

does the state of the atom change? To answer this question, we should think of the combined state of the atom and the electromagnetic field, which is initially $(\alpha_g|g\rangle + \alpha_e|e\rangle) \otimes |0\rangle$, where $|0\rangle$ means zero photon. During time evolution, $|e\rangle \otimes |0\rangle$ evolves to a superposition of $|e\rangle \otimes |0\rangle$ and $|g\rangle \otimes |1\rangle$, where the latter gives the probability of photodetection. A measurement outcome of $|0\rangle$ removes this part of the wave function, which reduces the excited-state population by $p_1|\alpha_g|^2$ (upon normalization). Thus, the atom moves closer to $|g\rangle$ without emitting a photon! It is as if the atom has evolved under the non-Hermitian Hamiltonian $\hat{H}_{\text{eff}} = \hat{H}_{\text{atom}} - i(\gamma/2)\hat{P}_e$ [44]. On the other hand, if a photon is detected, the state afterward is $|g\rangle$.

---

Q13.   Consider the time-evolved state $(\alpha_g|g\rangle + \alpha'_e|e\rangle) \otimes |0\rangle + \beta|g\rangle \otimes |1\rangle$ with $|\beta|^2 = p_1$. Show that an outcome of $|0\rangle$ reduces the norm by $p_1$ and the excited-state population by $p_1|\alpha_g|^2$. Show the same atomic state is obtained by evolving under $\hat{H}_{\text{eff}}$.

---

The change in the ensemble-averaged density matrix $\hat{\rho}$ is given by a weighted sum of these two possibilities. If no photon is detected (called a null measurement), the normalized density matrix is

$$\hat{\rho}_0 = \frac{\hat{\rho} - i(\hat{H}_{\text{eff}}\hat{\rho} - \hat{\rho}\hat{H}_{\text{eff}}^\dagger)dt}{1 - p_1} . \tag{56}$$

If a photon is detected, the density matrix is $\hat{\rho}_1 = |g\rangle\langle g|$, which can also be written as

$$\hat{\rho}_1 = \frac{\hat{\sigma}^-\hat{\rho}\hat{\sigma}^+}{\text{Tr}(\hat{\sigma}^-\hat{\rho}\hat{\sigma}^+)} = \frac{\hat{\sigma}^-\hat{\rho}\hat{\sigma}^+}{\langle\hat{P}_e\rangle} , \tag{57}$$

where $\hat{\sigma}^- := |g\rangle\langle e|$, $\hat{\sigma}^+ := |e\rangle\langle g|$, and thus $\hat{\sigma}^+\hat{\sigma}^- = \hat{P}_e$. Given that the two cases occur with probabilities $1 - p_1$ and $p_1$, respectively, one finds $\hat{\rho}(t + dt) = (1 - p_1)\hat{\rho}_0 + p_1\hat{\rho}_1$, or

$$\frac{d\hat{\rho}}{dt} = -i(\hat{H}_{\text{eff}}\hat{\rho} - \hat{\rho}\hat{H}_{\text{eff}}^\dagger) + \gamma\hat{\sigma}^-\hat{\rho}\hat{\sigma}^+ . \tag{58}$$

This is just Eq. (15) with a Lindblad operator $\hat{L} = \sqrt{\gamma}\hat{\sigma}^-$ and the Hamiltonian $\hat{H} = \hat{H}_{\text{atom}}$. The decay and recycling terms correspond to null measurements and detection events, respectively.

One can easily incorporate feedback into this framework. If we apply a unitary operator $\hat{U}$ on the atom whenever a photon is detected, $\hat{\rho}_1$ in Eq. (57) changes to $\hat{U}\hat{\sigma}^-\hat{\rho}\hat{\sigma}^+\hat{U}^\dagger/\langle\hat{P}_e\rangle$. Thus, one gets a Lindblad dynamics with $\hat{L} = \sqrt{\gamma}\hat{U}\hat{\sigma}^-$. (Note that $\hat{H}_{\text{eff}} = \hat{H} - \frac{i}{2}\hat{L}^\dagger\hat{L}$ still holds.) For a more general feedback $\Omega$, Eq. (58) becomes $d\hat{\rho}/dt = -i(\hat{H}_{\text{eff}}\hat{\rho} - \hat{\rho}\hat{H}_{\text{eff}}^\dagger) + \Omega(\gamma\hat{\sigma}^-\hat{\rho}\hat{\sigma}^+)$ [45].

The premise behind this derivation is that the excited state decays irreversibly to the ground state at a rate $\gamma$. This is not possible if the atom is coupled to one or a finite number of photon modes, as then the probability of exciting a photon in an interval $dt$ is $O(dt^2)$ (see Q14). Thus, an implicit assumption is that the atom is coupled to a continuum of photon modes. In fact, for an exponential decay one requires the coupling to be uniform over a bandwidth that is large compared to $\gamma$ [46] (see Q15). We have also assumed that the field is in state $|0\rangle$ at the start of each interval $dt$, i.e., the field relaxes to its original state much faster than the system dynamics. These are called Born-Markov approximations [15]. For a more rigorous, step-by-step derivation see [40].

Q14. Consider a two-level atom coupled to a single, resonant photon mode $\hat{a}$, described by the Hamiltonian $\hat{H} = \frac{\Delta}{2}\hat{\sigma}^z + \Delta\hat{a}^\dagger\hat{a} + \frac{\Omega}{2}(\hat{\sigma}^-\hat{a}^\dagger + \hat{\sigma}^+\hat{a})$. Suppose the initial state is $(\sin\theta|g\rangle + \cos\theta|e\rangle) \otimes |0\rangle$.

> 1) What is the expectation of the photon number as a function of time?
>
> 2) How does the state evolve if we continuously measure the photon number?

Q15. Consider a two-level atom with gap $\Delta$ coupled uniformly to a continuous band of photon modes. The rotating-frame Hamiltonian is $\hat{H}(t) = \sqrt{\gamma}\int d\omega\hat{\sigma}^-\hat{b}^\dagger(\omega)e^{i(\omega-\Delta)t} + $h.c., where $[\hat{b}(\omega), \hat{b}^\dagger(\omega')] = \delta(\omega-\omega')$ and h.c. denotes Hermitian conjugate. The initial state is $|e\rangle \otimes |0\rangle$.

> 1) Show that the excited-state population will decay exponentially with time provided $\gamma \ll \Delta$.
>
> 2) If the coupling is nonuniform, i.e., $\hat{H}(t) = \int d\omega\sqrt{\gamma(\omega)}\hat{\sigma}^-\hat{b}^\dagger(\omega)e^{i(\omega-\Delta)t} + $h.c., argue that the exponential decay is recovered if $\gamma(\omega)$ does not vary significantly in the frequency range $\Delta \pm \gamma(\Delta)$.

### 4.2.2 Quantum jump trajectories

If we record the measurement outcomes, each run of the experiment gives a quantum trajectory $\hat{\rho}_c(t)$ conditioned on the measurement record. In the above example, we count the number of photons $N$. In each interval $dt$, the change $dN(t)$ is either 0 or 1, with $\overline{dN} = \gamma\langle\hat{P}_e\rangle dt = \langle\hat{L}^\dagger\hat{L}\rangle_c dt$. (In the above example, the maximum photon number is 1, but this is generally not true. For example, a coupling between $|g\rangle$ and $|e\rangle$ in $\hat{H}_{\text{atom}}$ can bring the atom back to the excited state.) If $dN = 0$, the post-measurement state of the system is [Eq. (56)]

$$\hat{\rho}_0 = \frac{\hat{\rho}_c - i(\hat{H}_{\text{eff}}\hat{\rho}_c - \hat{\rho}_c\hat{H}_{\text{eff}}^\dagger)dt}{1 - \langle\hat{L}^\dagger\hat{L}\rangle_c dt} = \hat{\rho}_c - i(\hat{H}_{\text{eff}}\hat{\rho}_c - \hat{\rho}_c\hat{H}_{\text{eff}}^\dagger)dt + \hat{\rho}_c\langle\hat{L}^\dagger\hat{L}\rangle_c dt, \tag{59}$$

up to $O(dt)$. Using $\langle\hat{L}^\dagger\hat{L}\rangle_c = \text{Tr}(\hat{L}^\dagger\hat{L}\hat{\rho}_c) = \text{Tr}[i(\hat{H}_{\text{eff}} - \hat{H}_{\text{eff}}^\dagger)\hat{\rho}_c]$, we can rewrite the above as

$$\hat{\rho}_0 = \hat{\rho}_c + \mathcal{H}[-i\hat{H}_{\text{eff}}]\hat{\rho}_c dt, \quad \text{where} \quad \mathcal{H}[\hat{A}]\hat{\rho} := \hat{A}\hat{\rho} + \hat{\rho}\hat{A}^\dagger - \hat{\rho}\,\text{Tr}[(\hat{A} + \hat{A}^\dagger)\hat{\rho}]. \tag{60}$$

On the other hand, if $dN = 1$, the state becomes [Eq. (57)]

$$\hat{\rho}_1 = \frac{\hat{L}\hat{\rho}_c\hat{L}^\dagger}{\text{Tr}(\hat{L}\hat{\rho}_c\hat{L}^\dagger)} = \frac{\hat{L}\hat{\rho}_c\hat{L}^\dagger}{\langle\hat{L}^\dagger\hat{L}\rangle_c}. \tag{61}$$

Putting together, the conditional state at time $t + dt$ is given by

$$\hat{\rho}_c(t + dt) = \hat{\rho}_0(1 - dN) + \hat{\rho}_1 dN \tag{62}$$

$$= \hat{\rho}_c + \mathcal{G}[\hat{L}]\hat{\rho}_c dN + \mathcal{H}[-i\hat{H}_{\text{eff}}]\hat{\rho}_c dt, \quad \text{with} \quad \mathcal{G}[\hat{L}]\hat{\rho} := \frac{\hat{L}\hat{\rho}\hat{L}^\dagger}{\text{Tr}(\hat{L}\hat{\rho}\hat{L}^\dagger)} - \hat{\rho}. \tag{63}$$

Hence, the differential change is

$$d\hat{\rho}_c = \mathcal{G}[\hat{L}]\hat{\rho}_c dN + \mathcal{H}[-i\hat{H}_{\text{eff}}]\hat{\rho}_c dt. \tag{64}$$

The conditional state follows this stochastic master equation (SME), undergoing jumps whenever $dN(t) = 1$. From the expressions for $\mathcal{G}$ and $\mathcal{H}$, we see that the SME is nonlinear in $\hat{\rho}_c$, where the nonlinearity comes from normalizing $\hat{\rho}_0$ and $\hat{\rho}_1$. The unconditional (or ensemble-averaged) dynamics is obtained by replacing $dN$ by its average value, $\overline{dN} = \langle \hat{L}^\dagger \hat{L} \rangle_c dt$, which removes this nonlinearity and recovers the Lindblad equation,

$$\frac{d\hat{\rho}}{dt} = -i\big(\hat{H}_{\text{eff}}\hat{\rho} - \hat{\rho}\hat{H}_{\text{eff}}^\dagger\big) + \hat{L}\hat{\rho}\hat{L}^\dagger \, . \tag{65}$$

The ensemble of stochastic trajectories is thus said to constitute an unraveling of the Lindblad equation. Such an unraveling is not unique, however, as the following examples show.

Experimental measurements are imperfect. Suppose, in the above example, the photodetector records only a fraction $\eta$ of the emitted photons; the rest are lost (absorbed somewhere else, buried in noise, etc.). Then a null measurement, $dN = 0$, can imply one of two things: (i) no photon was emitted, which occurs with probability $1 - \langle \hat{L}^\dagger \hat{L} \rangle_c dt$, and (ii) an emitted photon was lost, which occurs with probability $p_{\text{lost}} = (1-\eta)\langle \hat{L}^\dagger \hat{L} \rangle_c dt$. Hence, the conditional state at time $t + dt$ should be modified as

$$\hat{\rho}_c(t + dt) = dN\hat{\rho}_1 + (1 - dN)\big[p_{\text{lost}}\hat{\rho}_1 + (1 - p_{\text{lost}})\hat{\rho}_0\big] \, . \tag{66}$$

Substituting the expressions for $\hat{\rho}_0$ and $\hat{\rho}_1$ from Eqs. (59) and (61) gives

$$d\hat{\rho}_c = \mathcal{G}[\hat{L}]\hat{\rho}_c dN + \mathcal{H}\Big[-i\hat{H} - \frac{\eta}{2}\hat{L}^\dagger\hat{L}\Big]\hat{\rho}_c dt + (1-\eta)\mathcal{D}[\hat{L}]\hat{\rho}_c dt \, , \tag{67}$$

where $\mathcal{D}$ is the Lindblad dissipator in Eq. (18). Effectively, a fraction $1 - \eta$ of the dynamics is unconditional, and follows the Lindblad master equation. The conditional part follows Eq. (64) with $\hat{L} \to \sqrt{\eta}\hat{L}$. In the limit $\eta = 0$, there is no measurement record ($dN = 0$) and Eq. (67) is just the Lindblad equation. If there are multiple independent loss processes $\hat{L}_\mu$ that are being continuously monitored with efficiencies $\eta_\mu$, Eq. (67) generalizes to

$$d\hat{\rho}_c = \sum_\mu \mathcal{G}[\hat{L}_\mu]\hat{\rho}_c dN_\mu + \mathcal{H}\Big[-i\hat{H} - \sum_\mu \frac{\eta_\mu}{2}\hat{L}_\mu^\dagger\hat{L}_\mu\Big]\hat{\rho}_c dt + \sum_\mu (1-\eta_\mu)\mathcal{D}[\hat{L}_\mu]\hat{\rho}_c dt \, . \tag{68}$$

In all cases, the ensemble-averaged dynamics is governed by the Lindblad equation (17).

---

Q16. Starting from Eq. (66), derive Eq. (67). Show that taking the ensemble average of Eq. (67) reproduces the Lindblad equation [Eq. (65)] for any value of $\eta \in [0, 1]$.

---

### 4.2.3 Monte Carlo wave functions

If one starts from a pure state and has a perfect measurement record ($\eta = 1$), the conditional state remains pure. This gives a very useful unraveling of the Lindblad equation in terms of conditional wave functions $|\psi_c(t)\rangle$. For $dN = 0$, the state changes to [see Eq. (59)]

$$|\psi_0\rangle = \frac{(1 - i\hat{H}_{\text{eff}}dt)|\psi_c\rangle}{\sqrt{1 - \langle \hat{L}^\dagger\hat{L} \rangle_c dt}} = |\psi_c\rangle - \Big(i\hat{H}_{\text{eff}} - \frac{1}{2}\langle \hat{L}^\dagger\hat{L} \rangle_c\Big)|\psi_c\rangle dt \, . \tag{69}$$

For $dN = 1$, the state jumps to [Eq. (61)]

$$|\psi_1\rangle = \frac{\hat{L}|\psi_c\rangle}{\sqrt{\langle\hat{L}^\dagger\hat{L}\rangle_c}} \; . \tag{70}$$

Hence, the conditional evolution is given by

$$|\psi_c(t+dt)\rangle = (1-dN)|\psi_0\rangle + dN|\psi_1\rangle \, , \tag{71}$$

$$\text{or} \quad d|\psi_c\rangle = \left(\frac{\hat{L}}{\sqrt{\langle\hat{L}^\dagger\hat{L}\rangle_c}} - 1\right)|\psi_c\rangle dN - \left(i\hat{H}_{\text{eff}} - \frac{1}{2}\langle\hat{L}^\dagger\hat{L}\rangle_c\right)|\psi_c\rangle dt \, , \tag{72}$$

where, as before, the probability of a jump is $\overline{dN} = \langle\hat{L}^\dagger\hat{L}\rangle_c dt$. This is called a stochastic Schrödinger equation (SSE), and the resulting pure-state trajectories are called Monte-Carlo wave functions [47]. Averaging any observable over this ensemble gives the same expectation as the Lindblad equation (65). For multiple dissipation processes, Eq. (72) generalizes to

$$d|\psi_c\rangle = \sum_\mu\left(\frac{\hat{L}_\mu}{\sqrt{\langle\hat{L}_\mu^\dagger\hat{L}_\mu\rangle_c}} - 1\right)|\psi_c\rangle dN_\mu - \left(i\hat{H}_{\text{eff}} - \frac{1}{2}\sum_\mu\langle\hat{L}_\mu^\dagger\hat{L}_\mu\rangle_c\right)|\psi_c\rangle dt \, , \tag{73}$$

with $\overline{dN_\mu} = \langle\hat{L}_\mu^\dagger\hat{L}_\mu\rangle_c dt$ and $\hat{H}_{\text{eff}} = \hat{H} - \frac{i}{2}\sum_\mu\hat{L}_\mu^\dagger\hat{L}_\mu$ as in Eq. (14).

This formalism was first proposed as a numerical technique for evolving the Lindblad master equation for large systems. (See [15] for a review.) This is because evolving a state vector of length $D$ is much less demanding (both time and memory wise) than evolving a density matrix of size $D \times D$. Generally the compute time scales as $D^2$ for the former and $D^4$ for the latter. Thus, averaging over Monte Carlo wave functions is useful whenever the number of trajectories required for reliable statistics is much less than $D^2$, which is often the case. The trajectories can also be generated in parallel on separate machines.

Integrating Eq. (73) over discrete time steps $\delta t$ has the downside that the jumps are not instantaneous but occur over a duration $\delta t$, which imposes an artificial lower bound on the waiting time $\tau$ between jumps. Instead, note that $|\psi_c\rangle$ simply evolves under $\hat{H}_{\text{eff}}$ (and renormalized) except when a jump occurs, so if we can properly sample these jump times, we should be able to generate the ensemble of trajectories more efficiently. The probability of a jump between $t$ and $t+dt$ is $f(t)dt$ where $f(t) = \sum_\mu\langle\hat{L}_\mu^\dagger\hat{L}_\mu\rangle_c$. Hence, the probability that no jump occurs up to time $T$ should satisfy $P(\tau \geq t+dt) = P(\tau \geq t)\big(1-f(t)dt\big)$, which gives

$$P(\tau \geq T) = e^{-\int_0^T f(t)dt} \; . \tag{74}$$

We make two observations. First, the above expression is just the norm of a state $|\phi_c(T)\rangle$ that evolves under $\hat{H}_{\text{eff}}$, i.e., $d|\phi_c\rangle/dt = -i\hat{H}_{\text{eff}}|\phi_c\rangle$. This is because

$$\frac{d}{dt}\langle\phi_c|\phi_c\rangle = i\langle\phi_c|\hat{H}_{\text{eff}}^\dagger - \hat{H}_{\text{eff}}|\phi_c\rangle = -\sum_\mu\langle\phi_c|\hat{L}_\mu^\dagger\hat{L}_\mu|\phi_c\rangle = -f(t)\langle\phi_c|\phi_c\rangle \, , \tag{75}$$

which gives

$$\langle\phi_c(T)|\phi_c(T)\rangle = e^{-\int_0^T f(t)dt} \, , \tag{76}$$

starting from unit norm at $t = 0$. The state $|\phi_c\rangle$ is an unnormalized version of the physical state $|\psi_c\rangle$. Second, the norm of $|\phi_c\rangle$ should be equally distributed between 0 and 1 at the jump times. To see this, let us calculate the probability that $\langle\phi_c(\tau)|\phi_c(\tau)\rangle \leq x$, where $\tau$ is the waiting time before the next jump and $x \in [0,1]$:

$$P\big(\langle\phi_c(\tau)|\phi_c(\tau)\rangle \leq x\big) = P\left(e^{-\int_0^\tau f(t)\mathrm{d}t} \leq x\right) = P\left(\tau \geq T \,\bigg|\, \int_0^T f(t)\mathrm{d}t = -\ln x\right) = e^{-\ln x} = x \,, \tag{77}$$

which implies a uniform distribution in $[0,1]$. Thus, we implement the jumps as follows: Draw a random number $r \in [0,1]$, then evolve $|\phi_c\rangle$ until its norm crosses $r$. This is when a jump occurs. If there are multiple dissipation processes, the state changes to $\hat{L}_\mu|\phi_c\rangle/\langle\phi_c|\hat{L}_\mu^\dagger\hat{L}_\mu|\phi_c\rangle^{1/2}$ with probability $p_\mu \propto \langle\phi_c|\hat{L}_\mu^\dagger\hat{L}_\mu|\phi_c\rangle$ (with $\sum_\mu p_\mu = 1$). Then the same process is repeated. This gives an alternative and numerically efficient formulation of the trajectories in Eq. (73).

Figures 2(a) and 2(b) show a trajectory for a spin-1/2 with drive and spontaneous emission. The drive rotates the spin about the $x$ axis, while the loss projects to spin-↓. As shown in Fig. 2(c), averaging over many such trajectories reproduces the expectation values obtained from the Lindblad equation. Note that the jumps continue to happen in a given trajectory after the ensemble average has reached steady state. The counting statistics of these jumps contain information about the nature of the steady state [48, 49].

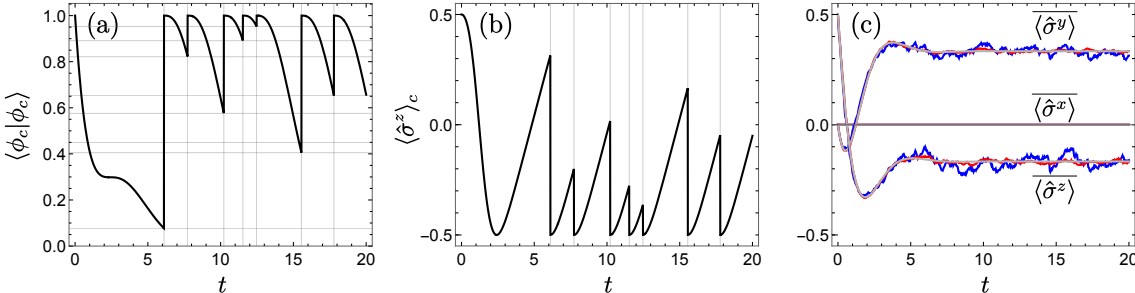

Figure 2: (a) Norm of $|\phi_c\rangle$ and (b) $z$-magnetization for a Monte-Carlo trajectory of a spin-1/2 with Hamiltonian $\hat{H} = \hat{S}^x$ and jump operator $\hat{L} = \hat{S}^-$. (c) Average over 100 (blue) and 1000 (red) trajectories compared to the Lindblad dynamics (gray).

---

Q17.  Consider a spin-1 (three-level system) initially pointing along $+x$ with $\hat{H} = 0$.

1) Generate 1000 trajectories for the dissipation $\hat{L} = \hat{S}^-$ up to $t = 10$ and check that the average evolution of $\langle\hat{S}^x\rangle$, $\langle\hat{S}^y\rangle$, and $\langle\hat{S}^z\rangle$ agree closely with the Lindblad equation.

2) Make a histogram of the number of jumps. Can you explain the relative frequencies based on the initial state?

3) Does a jump always decrease $\langle\hat{S}^z\rangle_c$? Why or why not?

---

Q18.  Consider a driven two-level atom undergoing spontaneous emission with $\hat{H} = \hat{S}^x$ and $\hat{L} = \sqrt{2}\,\hat{S}^-$.

1) Find a closed-form expression for the waiting-time distribution in steady state. (Hint: you can use Eqs. (74) and (76) with a suitable initial state). Check that the average

emission rate $\langle \hat{L}^\dagger \hat{L} \rangle$ in steady state equals the inverse of the mean waiting time.

2) Make a histogram of the number of emitted photons up to $t = 100$ from many trajectories. Check this is sub-Poissonian (i.e. variance / mean < 1), which is also called nonclassical light [50, 51]. Can you explain the mean and variance of the photon count from the waiting-time distribution? You may find these notes helpful: [52].

### 4.2.4 Quantum state diffusion

In Sec. 3.3.3 we saw that the Lindblad master equation is invariant under $\hat{L} \rightarrow \hat{L} + a$ and $\hat{H} \rightarrow \hat{H} - \frac{i}{2}(a^* \hat{L} - a \hat{L}^\dagger)$. Such a transformation yields a different type of unraveling where the conditional state undergoes a Brownian motion [1, Chap. 9]. For a physical scenario, suppose $\hat{L}$ describes photon loss from a cavity. The transformation is realized in a homodyne detection where, instead of detecting the photons directly, one first interferes the cavity output with a laser of amplitude $a$ (called a local oscillator) and measures the displaced field $\hat{L} + a$ [8, Chap. 4]. The laser has the same frequency as the cavity and a strong coherent field which is replaced by its classical value $a$. (We will later take the limit $a \rightarrow \infty$.) Let us also assume that the two fields are in phase so that $a$ is real. The average photon count in an interval $dt$ is $\overline{dN} = \langle (\hat{L}^\dagger + a)(\hat{L} + a) \rangle_c dt = (a^2 + 2a\langle \hat{x} \rangle_c) dt$, where $\hat{x} := (\hat{L} + \hat{L}^\dagger)/2$. We have dropped the $\langle \hat{L}^\dagger \hat{L} \rangle_c$ term as its contribution will vanish in the limit $a \rightarrow \infty$. Thus, the photon count measures a quadrature $\hat{x}$ of the cavity field. Additionally, the fluctuation in $dN$ is dominated by the coherent laser field, for which the variance is equal to the mean, i.e., $\sigma^2 = a^2 dt$. Thus, one can replace $dN$ by the Gaussian process (Poissonian with large mean) [53]

$$dN = (a^2 + 2a\langle \hat{x} \rangle_c) dt + a \, dW , \tag{78}$$

where $dW$ is a Wiener increment satisfying $\overline{dW} = 0$ and $dW^2 = dt$ [54]. Substituting the transformed operators into Eq. (72) gives the conditional evolution

$$d|\psi_c\rangle = \left( \frac{a + \hat{L}}{\sqrt{\langle (a + \hat{L}^\dagger)(a + \hat{L}) \rangle_c}} - 1 \right) |\psi_c\rangle dN - \left( i\hat{H} + \frac{1}{2}\hat{L}^\dagger\hat{L} - \frac{1}{2}\langle \hat{L}^\dagger\hat{L} \rangle_c + a\hat{L} - a\langle \hat{x} \rangle_c \right) |\psi_c\rangle dt . \tag{79}$$

Using Eq. (78) and taking the limit $a \rightarrow \infty$, we find

$$d|\psi_c\rangle = (\hat{L} - \langle \hat{x} \rangle_c)|\psi_c\rangle dW + \left( -i\hat{H}_{\text{eff}} + \hat{L}\langle \hat{x} \rangle_c - \frac{1}{2}\langle \hat{x} \rangle_c^2 \right) |\psi_c\rangle dt , \tag{80}$$

where $\hat{H}_{\text{eff}} := \hat{H} - \frac{i}{2}\hat{L}^\dagger\hat{L}$, as before. This is a nonlinear Langevin equation where the stochastic term produces diffusion, as in a Brownian motion. The stochastic term is interpreted in the Itô sense [54, 55], as $\hat{L}$ is applied to the state before photoemission. The unnormalized state $|\phi_c\rangle$ is governed by a simpler equation,

$$d|\phi_c\rangle = \hat{L}|\phi_c\rangle dW + (2\hat{L}\langle \hat{x} \rangle_c - i\hat{H}_{\text{eff}})|\phi_c\rangle dt , \tag{81}$$

$$\text{or} \quad \frac{d|\phi_c\rangle}{dt} = \left[ -i\hat{H}_{\text{eff}} + \hat{L}J_{\text{hom}}(t) \right] |\phi_c\rangle , \tag{82}$$

$$\text{where} \quad J_{\text{hom}} := \frac{1}{a}\frac{dN}{dt} - a = 2\langle \hat{x} \rangle_c + \frac{dW}{dt} \tag{83}$$

is the homodyne photocurrent. Equation (82) nicely captures the effect of the measured current on the conditional state.

> Q19. From Eq. (80) show that the noise-averaged density matrix follows the Lindblad dynamics in Eq. (65).

> Q20. Check that normalizing $|\phi_c\rangle$ in Eq. (81) leads to Eq. (80). (Make sure to include all terms up to $O(\mathrm{d}t)$ in each differential, noting that $\mathrm{d}W^2 = \mathrm{d}t$ [54].)

If one measures both quadratures, $\hat{L} \pm \hat{L}^\dagger$, using a detuned local oscillator (called heterodyne detection [8, Chap. 4]), Eq. (80) changes to

$$\mathrm{d}|\psi_c\rangle = \left(\hat{L} - \langle\hat{L}\rangle_c\right)|\psi_c\rangle\mathrm{d}Z + \left(-\mathrm{i}\hat{H}_{\mathrm{eff}} + \hat{L}\langle\hat{L}\rangle_c^* - \frac{1}{2}\left|\langle\hat{L}\rangle_c\right|^2\right)|\psi_c\rangle\mathrm{d}t \,, \tag{84}$$

where $\mathrm{d}Z$ is a complex Wiener increment with independent real and imaginary parts, such that $\overline{\mathrm{d}Z} = \mathrm{d}Z^2 = 0$ and $\mathrm{d}Z^*\mathrm{d}Z = \mathrm{d}t$. This unraveling is known as the quantum state diffusion [56], which was realized in [57]. Equations (80) and (84) can be readily generalized for multiple dissipation channels, e.g., the unnormalized version of the latter reads

$$\mathrm{d}|\phi_c\rangle = -\mathrm{i}\hat{H}_{\mathrm{eff}}|\phi_c\rangle\mathrm{d}t + \sum_\mu \left(\hat{L}_\mu\langle\hat{L}_\mu^\dagger\rangle_c|\phi_c\rangle\mathrm{d}t + \hat{L}_\mu|\phi_c\rangle\mathrm{d}Z_\mu\right) \,, \tag{85}$$

where $\mathrm{d}Z_\mu$ are independent Wiener increments with $\mathrm{d}Z_\mu^*\mathrm{d}Z_\nu = \delta_{\mu,\nu}\mathrm{d}t$.

Figures 3(a) and 3(b) show a trajectory of a driven, damped harmonic oscillator undergoing quantum state diffusion, with $\hat{H} = \mathrm{i}(\hat{a}^\dagger - \hat{a})$ and $\hat{L} = \hat{a}$. Physically, this corresponds to a lossy cavity that is resonantly driven by a laser and continuously monitored (by heterodyne detection). Starting from a Fock state with 8 photons, the cavity evolves to the coherent steady state $|\alpha = 2\rangle$ (with $\hat{a}|\alpha\rangle = \alpha|\alpha\rangle$) where the fluctuations die out (see Q21). Here, the measurements localize the oscillator in phase space [56]. Note that each trajectory evolves differently before reaching the same, pure steady state. This is reflected in the initial loss and subsequent regain of purity of the ensemble-averaged density matrix [Fig. 3(c)]. Generically, the steady state is not pure and individual trajectories fluctuate forever (as in Fig. 2).

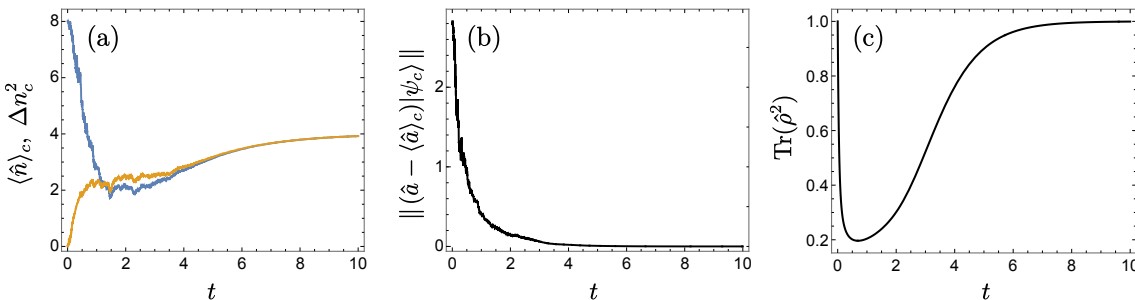

Figure 3: (a) Expectation (blue) and variance (orange) of the photon number in a given realization of quantum state diffusion for $\hat{H} = \mathrm{i}(\hat{a}^\dagger - \hat{a})$ and $\hat{L} = \hat{a}$. (b) Distance from a coherent state, where $\| \, |\psi\rangle \, \| := \sqrt{\langle\psi|\psi\rangle}$. (c) Purity of the ensemble-averaged density matrix.

Q21. Consider a driven, lossy cavity described by $\hat{H} = \frac{\Omega}{2}\hat{a}^\dagger + \frac{\Omega^*}{2}\hat{a}$ and $\hat{L} = \hat{a}$. Show that it reaches the coherent state $|\alpha = -i\Omega\rangle$ from the Lindblad equation. This is a consequence of the single-photon drive and loss. A two-photon drive leads to squeezed states [10,58].

Q22. Consider a purely dissipative dynamics of a qubit with $\hat{H} = 0$ and $\hat{L} = \hat{\sigma}^z$.

    1) Starting from $|+\rangle = \frac{1}{\sqrt{2}}(|\uparrow\rangle + |\downarrow\rangle)$, what is the Lindbladian steady state?

    2) Generate Monte Carlo wave functions and quantum state diffusion trajectories for the same initial state. How do the two types of trajectories differ? (You can simulate Eq. (84) with the Euler-Maruyama method [59]. Check that the norm is conserved.)

    3) From Eq. (84) show that any (pure-state) trajectory undergoing quantum state diffusion reaches an eigenstate of $\hat{\sigma}^z$.

## 4.3 Tracing out a reservoir

Let us now sketch the standard route to the master equation by starting from a Hamiltonian for the system plus a reservoir and then tracing out the reservoir. Several derivations exist in the literature; here we largely follow the steps in [1, Chap. 1]. Consider the Hamiltonian

$$\hat{H} = \hat{H}_S + \hat{H}_R + \hat{H}_{SR} \,, \tag{86}$$

where $\hat{H}_S$, $\hat{H}_R$, and $\hat{H}_{SR}$ describe the system, the reservoir, and their interactions, respectively. Let $\hat{\chi}$ denote the density matrix of the system + reservoir and $\hat{\rho}$ the reduced density matrix of the system, $\hat{\rho} = \text{Tr}_R(\hat{\chi})$. The equation of motion for $\hat{\chi}$ is simply

$$\frac{\mathrm{d}\hat{\chi}}{\mathrm{d}t} = -i[\hat{H}, \hat{\chi}] \,. \tag{87}$$

The derivation assumes that the system-reservoir coupling is weak, so it is natural to separate the rapid evolution generated by $\hat{H}_S + \hat{H}_R$ from the slow evolution generated by $\hat{H}_{SR}$. This is achieved by transforming all operators $\hat{O}$ as

$$\tilde{O}(t) := e^{i(H_S+H_R)t} O e^{-i(H_S+H_R)t}. \tag{88}$$

(We assume that $\hat{H}_S + \hat{H}_R$ is time independent and drop the hats over operators for notational clarity.) In this interaction picture, the equation of motion becomes

$$\frac{\mathrm{d}\tilde{\chi}}{\mathrm{d}t} = -i[\tilde{H}_{SR}(t), \tilde{\chi}] \tag{89}$$

and the system density matrix is given by $\tilde{\rho} = \text{Tr}_R(\tilde{\chi})$. We formally integrate Eq. (89),

$$\tilde{\chi}(t) = \chi(0) - i \int_0^t \mathrm{d}t' [\hat{H}_{SR}(t'), \tilde{\chi}(t')] \,, \tag{90}$$

and substitute this expression back into the right-hand side of Eq. (89) to obtain

$$\frac{\mathrm{d}\tilde{\chi}(t)}{\mathrm{d}t} = -i[\tilde{H}_{SR}(t), \chi(0)] - \int_0^t \mathrm{d}t' [\tilde{H}_{SR}(t), [\tilde{H}_{SR}(t'), \tilde{\chi}(t')]] \,. \tag{91}$$

Suppose the interaction is turned on at $t = 0$ when there is no correlation between the system and the reservoir, i.e., the state factorizes as $\chi(0) = \rho(0) \otimes R_0$. Using this form in Eq. (91) and tracing out the reservoir gives

$$\frac{d\tilde{\rho}(t)}{dt} = -i\big[\text{Tr}_R\big(\tilde{H}_{SR}(t)R_0\big), \rho(0)\big] - \int_0^t dt'\, \text{Tr}_R\big[\tilde{H}_{SR}(t), \big[\tilde{H}_{SR}(t'), \tilde{\chi}(t')\big]\big]. \qquad (92)$$

One typically considers interactions of the form

$$H_{SR} = \sum_\mu s_\mu r_\mu \qquad (93)$$

where $s_\mu$ acts on the system and $r_\mu$ acts on the reservoir. The first term in Eq. (92) vanishes if $\text{Tr}_R(\tilde{r}_\mu(t)R_0) = 0$ or $\text{Tr}_R\big(r_\mu e^{-iH_R t}R_0 e^{iH_R t}\big) = 0$. If the reservoir is steady under its own Hamiltonian, i.e., $[H_R, R_0] = 0$, this condition is guaranteed by redefining $H_{SR} \to \sum_\mu s_\mu\big(r_\mu - \langle r_\mu\rangle_{R_0}\big)$ and $H_S \to H_S + \sum_\mu s_\mu\langle r_\mu\rangle_{R_0}$. Thus, it is customary to write Eq. (92) as

$$\frac{d\tilde{\rho}(t)}{dt} = -\int_0^t dt'\, \text{Tr}_R\big[\tilde{H}_{SR}(t), \big[\tilde{H}_{SR}(t'), \tilde{\chi}(t')\big]\big]. \qquad (94)$$

Next comes the Born approximation: as the coupling is weak, we assume $\chi(t)$ is close to an uncorrelated (i.e., product) state at all times, deviating only up to $O(H_{SR})$. Furthermore, the reservoir is so large that it is virtually unaffected by its coupling to $S$, which gives

$$\tilde{\chi}(t) = \tilde{\rho}(t) \otimes R_0 + O(H_{SR}). \qquad (95)$$

Under this approximation, the equation of motion correct to second-order in $H_{SR}$ is

$$\frac{d\tilde{\rho}(t)}{dt} = -\int_0^t dt'\, \text{Tr}_R\big[\tilde{H}_{SR}(t), \big[\tilde{H}_{SR}(t'), \tilde{\rho}(t')R_0\big]\big]. \qquad (96)$$

This equation is still non-Markovian. Physically, this occurs because earlier states of the system affect the reservoir, which is fed back to the system later. However, if the reservoir is large and in equilibrium, minor changes due to interactions with the system do not persist for long. It relaxes back to equilibrium over a short timescale $\tau_R$, during which there is hardly any change in $\tilde{\rho}$. Thus, most of the contribution in Eq. (96) comes from $t' \gtrsim t - \tau_R$, when $\tilde{\rho}(t') \approx \tilde{\rho}(t)$. Under this Markov approximation, Eq. (96) becomes (for $t \gg \tau_R$),

$$\frac{d\tilde{\rho}(t)}{dt} = -\int_0^\infty d\tau\, \text{Tr}_R\big[\tilde{H}_{SR}(t), \big[\tilde{H}_{SR}(t - \tau), \tilde{\rho}(t)R_0\big]\big]. \qquad (97)$$

This is called a Redfield master equation, which is not necessarily of the Lindblad form and thus does not always preserve positivity, although it can be more accurate in other respects [60]. As we shall see in the example below Q23, one recovers a Lindblad form by dropping rapidly oscillating terms, which is called a rotating-wave or secular approximation.

First let us see how the Markov approximation works for the coupling in Eq. (93). Writing $\tilde{H}_{SR}(t) = \sum_\mu \tilde{s}_\mu(t)\tilde{r}_\mu(t)$ and expanding the commutators in Eq. (96), we find

$$\frac{d\tilde{\rho}(t)}{dt} = \sum_{\mu,\nu} \int_0^t dt'\Big\{\big[\tilde{s}_\nu(t')\tilde{\rho}(t')\tilde{s}_\mu(t) - \tilde{s}_\mu(t)\tilde{s}_\nu(t')\tilde{\rho}(t')\big]\langle\tilde{r}_\mu(t)\tilde{r}_\nu(t')\rangle$$

$$+ \big[\tilde{s}_\mu(t)\tilde{\rho}(t')\tilde{s}_\nu(t') - \tilde{\rho}(t')\tilde{s}_\nu(t')\tilde{s}_\mu(t)\big]\langle\tilde{r}_\nu(t')\tilde{r}_\mu(t)\rangle\Big\}. \qquad (98)$$

This explicitly depends on the two-time correlations $\langle \tilde{r}_\mu(t)\tilde{r}_\nu(t')\rangle$ of the reservoir (with respect to the state $R_0$), which decay over a short time $\tau_R$. Approximating $\langle \tilde{r}_\mu(t)\tilde{r}_\nu(t')\rangle \approx C_{\nu,\mu}\delta(t-t')$ gives the Markovian (Redfield) master equation

$$\frac{\mathrm{d}\tilde{\rho}}{\mathrm{d}t} = \sum_{\mu,\nu} C_{\mu,\nu}\left[\tilde{s}_\mu\tilde{\rho}\tilde{s}_\nu - \frac{1}{2}\left(\tilde{s}_\nu\tilde{s}_\mu\tilde{\rho} + \tilde{\rho}\tilde{s}_\nu\tilde{s}_\mu\right)\right], \tag{99}$$

which, upon unwinding the transformation in Eq. (88) becomes

$$\frac{\mathrm{d}\rho}{\mathrm{d}t} = -\mathrm{i}[H_S,\rho] + \sum_{\mu,\nu} C_{\mu,\nu}\left[s_\mu\rho s_\nu - \frac{1}{2}\left(s_\nu s_\mu\rho + \rho s_\nu s_\mu\right)\right]. \tag{100}$$

This looks like the nondiagnonal form of the Lindblad equation in Eq. (16). However, it is not necessarily completely positive as the coefficient matrix $C_{\mu,\nu}$ is not always positive semidefinite.

As an illustration of the above points, consider a two-level atom coupled to many photon modes, described by

$$H_S = \frac{1}{2}\omega_S\sigma^z, \quad H_R = \sum_k \omega_k b_k^\dagger b_k, \tag{101}$$

$$H_{SR} = \sigma^+ B + \sigma^- B^\dagger, \quad \text{with} \quad B = \sum_k \alpha_k b_k. \tag{102}$$

Comparing with Eq. (93), we identify $s_1 = \sigma^+$, $s_2 = \sigma^-$, $r_1 = B$, and $r_2 = B^\dagger$. Using Eq. (88), the interaction-picture operators are

$$\tilde{s}_1 = \sigma^+ e^{\mathrm{i}\omega_S t}, \quad \tilde{s}_2 = \sigma^- e^{-\mathrm{i}\omega_S t}, \quad \tilde{r}_1 = \sum_k \alpha_k b_k e^{-\mathrm{i}\omega_k t}, \quad \text{and} \quad \tilde{r}_2 = \tilde{r}_1^\dagger. \tag{103}$$

To evaluate the correlations in Eq. (98), let us take the reservoir at temperature $T$, for which $\langle b_k b_{k'}\rangle = 0$ and $\langle b_k^\dagger b_{k'}\rangle = n_k(T)\delta_{k,k'}$, where $n_k(T) = 1/(e^{\omega_k/T}-1)$. This gives

$$\langle \tilde{r}_1(t)\tilde{r}_1(t')\rangle = 0, \quad \langle \tilde{r}_2(t)\tilde{r}_2(t')\rangle = 0, \tag{104}$$

$$\langle \tilde{r}_1(t)\tilde{r}_2(t')\rangle = \sum_k |\alpha_k|^2 \left[n_k(T)+1\right]e^{-\mathrm{i}\omega_k(t-t')}, \tag{105}$$

$$\text{and} \quad \langle \tilde{r}_2(t)\tilde{r}_1(t')\rangle = \sum_k |\alpha_k|^2 n_k(T)\, e^{\mathrm{i}\omega_k(t-t')}. \tag{106}$$

For simplicity, consider $T=0$ for which $n_k = 0$ and Eq. (98) reduces to

$$\frac{\mathrm{d}\tilde{\rho}(t)}{\mathrm{d}t} = \int_0^t \mathrm{d}\tau \sum_k |\alpha_k|^2 e^{\mathrm{i}(\omega_S-\omega_k)\tau}\left[\sigma^-\tilde{\rho}(t-\tau)\sigma^+ - \sigma^+\sigma^-\tilde{\rho}(t-\tau)\right] + \text{h.c.}. \tag{107}$$

To simplify the right-hand side, we take the continuum limit for the photon modes, replacing $\sum_k$ by $\int \mathrm{d}\omega g(\omega)$, where $g(\omega)$ is the density of states. Then one has to evaluate the integral

$$I = \int_0^t \mathrm{d}\tau\left[\int_0^\infty \mathrm{d}\omega\, g(\omega)|\alpha(\omega)|^2\, e^{\mathrm{i}(\omega_S-\omega)\tau}\right]\tilde{\rho}(t-\tau). \tag{108}$$

The quantity inside the square brackets is related to the Fourier transform of $g(\omega)|\alpha(\omega)|^2$. If the reservoir has a large bandwidth $1/\tau_R$, the integral is significant only for $\tau \lesssim \tau_R$, over

which $\tilde{\rho}(t-\tau) \approx \tilde{\rho}(t)$. Thus, for $t \gg \tau_R$,

$$I \approx \tilde{\rho}(t) \int_0^\infty \mathrm{d}\omega \, g(\omega)|\alpha(\omega)|^2 \int_0^\infty \mathrm{d}\tau \, e^{\mathrm{i}(\omega_S - \omega)\tau} \tag{109}$$

$$= \tilde{\rho}(t) \int_0^\infty \mathrm{d}\omega \, g(\omega)|\alpha(\omega)|^2 \left[ \pi\delta(\omega_S - \omega) + \mathrm{i}\frac{\mathcal{P}}{\omega_S - \omega} \right], \tag{110}$$

where $\mathcal{P}$ is the principal value [61]. Evaluating the integral gives $I \approx (\gamma/2 + \mathrm{i}\Delta)\tilde{\rho}(t)$, with

$$\gamma = 2\pi g(\omega_S)|\alpha(\omega_S)|^2 \quad \text{and} \quad \Delta = \mathcal{P}\int_0^\infty \mathrm{d}\omega \, \frac{g(\omega)|\alpha(\omega)|^2}{\omega_S - \omega} \,. \tag{111}$$

Using this result in Eq. (107), we find

$$\frac{\mathrm{d}\tilde{\rho}}{\mathrm{d}t} = \left( \frac{\gamma}{2} + \mathrm{i}\Delta \right) \left[ \sigma^- \tilde{\rho}(t)\sigma^+ - \sigma^+\sigma^-\tilde{\rho}(t) \right] + \text{h.c.} \tag{112}$$

$$= \gamma\mathcal{D}[\sigma^-]\tilde{\rho} - \mathrm{i}\left[ \Delta\sigma^+\sigma^-, \tilde{\rho} \right]. \tag{113}$$

Finally, unwinding the transformation in Eq. (88) yields the Lindblad equation

$$\frac{\mathrm{d}\rho}{\mathrm{d}t} = -\mathrm{i}[H_S + \Delta\sigma^+\sigma^-, \rho] + \gamma\mathcal{D}[\sigma^-]\rho \,. \tag{114}$$

Here, $\Delta$ alters the gap between the two energy levels of $H_S$ and is called the Lamb shift.

---

Q23. Show that, for $T \neq 0$, the above steps would lead to the Lindblad equation

$$\frac{\mathrm{d}\rho}{\mathrm{d}t} = -\mathrm{i}[H_S + \Delta'\sigma^+\sigma^-, \rho] + \gamma\left[n(\omega_S, T) + 1\right]\mathcal{D}[\sigma^-]\rho + \gamma n(\omega_S, T)\mathcal{D}[\sigma^+]\rho \,, \tag{115}$$

where the Lamb shift $\Delta'$ depends on $T$ and $n(\omega, T) := 1/(e^{\omega/T} - 1)$. The terms proportional to $n(\omega_S, T)$ in Eq. (115) represent stimulated emission and absorption.

---

In the above example, the Born-Markov approximation led directly to a Lindblad equation. This is because $H_{SR}$ in Eq. (102) exchanges excitation between the atom and the electromagnetic field, so the total excitation is conserved. If one instead has a Dicke-type coupling,

$$H_{SR} = \sigma^x (B + B^\dagger) \,, \tag{116}$$

Eq. (107) would become, after the Markov and continuum approximations,

$$\frac{\mathrm{d}\tilde{\rho}(t)}{\mathrm{d}t} = \int_0^\infty \mathrm{d}\tau \int_0^\infty \mathrm{d}\omega \, g(\omega)|\alpha(\omega)|^2 e^{-\mathrm{i}\omega\tau} \left\{ e^{\mathrm{i}\omega_S(\tau-2t)}\sigma^-\tilde{\rho}(t)\sigma^- + e^{\mathrm{i}\omega_S(2t-\tau)}\sigma^+\tilde{\rho}(t)\sigma^+ \right.$$

$$\left. + e^{\mathrm{i}\omega_S\tau}\left[\sigma^-\tilde{\rho}(t)\sigma^+ - \sigma^+\sigma^-\tilde{\rho}(t)\right] + e^{-\mathrm{i}\omega_S\tau}\left[\sigma^+\tilde{\rho}(t)\sigma^- - \sigma^-\sigma^+\tilde{\rho}(t)\right] \right\} + \text{h.c.} \tag{117}$$

The first two terms inside the curly braces oscillate as $e^{\pm 2\mathrm{i}\omega_S t}$. For $\omega_S \gg H_{SR}$, these rapid oscillations average out and can be ignored. Only after this rotating-wave approximation does one recover a Lindblad form, which reduces to Eq. (114) in the limit $\omega_S \to \infty$ (note that the contribution of the last term in Eq. (117) falls as $1/\omega_S$).

# 5 Structure of steady states

So far we have discussed the mathematical framework and physical realizations of Markovian quantum master equations. In this section we will go over the various types of steady-state manifolds one can stabilize.

## 5.1 Uniqueness of steady state

If the Liouvillian has a unique steady state (and all other eigenvalues have negative real parts), any initial state evolves to the same steady state. This complete loss of memory stems from the Hilbert space being fully connected under the dynamics (a form of ergodicity). In particular, one can prove that:

Systems with a finite Hilbert-space dimension have a unique, full-rank steady state if and only if there is no invariant subspace, i.e., the only projectors $\hat{P}$ satisfying

$$\mathcal{L}(\hat{P}\hat{\rho}\hat{P}) = \hat{P}\mathcal{L}(\hat{P}\hat{\rho}\hat{P})\hat{P} \quad \forall \hat{\rho} \tag{118}$$

are $\hat{0}$ and $\hat{1}$. This condition is called Davies irreducibility [62]. Furthermore, as the evolution is generated by $\hat{L}_\mu$ and $\hat{H}_{\text{eff}}$ [Eq. (15)], one can derive the following algebraic condition [63]: For a finite Hilbert-space dimension, the dynamics is Davies irreducible if and only if $\hat{L}_\mu$ and $\hat{H}_{\text{eff}}$ generate, under addition and multiplication, the whole operator algebra. (This is not necessarily the same as the algebra generated by $\hat{L}_\mu$ and $\hat{H}$; see Q24.)

Although this condition may sound restrictive, it generically holds for interacting Hamiltonians with local Lindblad operators. For instance, consider a transverse-field Ising chain with incoherent spin flips at one end:

$$\hat{H} = \sum_{i=1}^{N} h_i \hat{\sigma}_i^z + \sum_{i=1}^{N-1} J_i \hat{\sigma}_i^x \hat{\sigma}_{i+1}^x, \quad \hat{L}_1 = \sqrt{\gamma_+}\, \hat{\sigma}_1^+, \quad \hat{L}_2 = \sqrt{\gamma_-}\, \hat{\sigma}_1^-, \tag{119}$$

where $\gamma_\pm > 0$ and $h_i, J_i \neq 0\ \forall i$. Let us denote the algebra generated by $\{\hat{L}_1, \hat{L}_2, \hat{H}_{\text{eff}}\}$ by $\mathcal{W}$. Then we have

$$\hat{\sigma}_1^\pm \in \mathcal{W} \quad \Longrightarrow \quad \hat{1}, \hat{\sigma}_1^x, \hat{\sigma}_1^y, \hat{\sigma}_1^z \in \mathcal{W} \quad \Longrightarrow \quad \hat{L}_1^\dagger, \hat{L}_2^\dagger \in \mathcal{W} \tag{120}$$

$$\Longrightarrow \quad \hat{H} = \hat{H}_{\text{eff}} + \frac{i}{2}\big(\hat{L}_1^\dagger \hat{L}_1 + \hat{L}_2^\dagger \hat{L}_2\big) \in \mathcal{W} \quad \Longrightarrow \quad \hat{H} - h_1 \hat{\sigma}_1^z \in \mathcal{W}. \tag{121}$$

Next, we can use commutation and multiplication to show that $\mathcal{W}$ contains the full algebra for site 2:

$$[\hat{\sigma}_1^y, \hat{H} - h_1\hat{\sigma}_1^z] = J_1[\hat{\sigma}_1^y, \hat{\sigma}_1^x \hat{\sigma}_2^x] = -2iJ_1 \hat{\sigma}_1^z \hat{\sigma}_2^x \in \mathcal{W} \quad \Longrightarrow \quad \hat{\sigma}_1^z \hat{\sigma}_1^z \hat{\sigma}_2^x = \hat{\sigma}_2^x \in \mathcal{W}, \tag{122}$$

$$[\hat{\sigma}_2^x, \hat{H}] = h_2[\hat{\sigma}_2^x, \hat{\sigma}_2^z] = -2ih_2 \sigma_2^y \in \mathcal{W} \quad \Longrightarrow \quad \hat{\sigma}_2^x, \hat{\sigma}_2^y, \hat{\sigma}_2^z \in \mathcal{W}. \tag{123}$$

Subtracting the first site from $\hat{H}$ and repeating the process on site 2 gives $\hat{\sigma}_3^x, \hat{\sigma}_3^y, \hat{\sigma}_3^z \in \mathcal{W}$, and so on. Therefore, the system is Davies irreducible and has a unique, full-rank steady state. For more examples see [63].

Note that the theorem concerns full-rank (also called faithful) steady states, i.e. density matrices that are positive definite. It is possible for the dynamics to be Davies reducible yet have a unique steady state as long as it is not full rank. A classic example is a pure steady state $|\psi\rangle$ (also called a dark state), for which Eq. (118) is satisfied with $\hat{P} = |\psi\rangle\langle\psi|$. For instance, if one has a qubit with $\hat{H} = \hat{\sigma}^z$ and $\hat{L} = \hat{\sigma}^+$, the steady state is $|\uparrow\rangle$. One can readily check that $\hat{L}$ and $\hat{H}_{\text{eff}}$ do not generate the full algebra as $\hat{L}\hat{H}_{\text{eff}} = -(1 + i/2)\hat{L}$ and $\hat{H}_{\text{eff}}\hat{L} = \hat{L}$.

---

Q24. Find the steady state(s) and check the algebraic condition for irreducibility for:
1) A single qubit with $\hat{H} = 0$ and $\hat{L} = 1 + \hat{\sigma}^+$.
2) Two qubits with $\hat{H} = 0$ and $\hat{L}_1 = \hat{P}_1^\uparrow \hat{\sigma}_2^+$, $\hat{L}_2 = \hat{P}_1^\downarrow \hat{\sigma}_2^-$, $\hat{L}_3 = \hat{\sigma}_1^+ \hat{P}_2^\uparrow$, $\hat{L}_4 = \hat{\sigma}_1^- \hat{P}_2^\downarrow$, $\hat{L}_5 = \hat{\sigma}_1^z + \hat{\sigma}_2^z$.

---

## 5.2 Dark states

Any pure steady state is called a dark state. When the system reaches such a state, no further quantum jumps or "photoemissions" can occur (as that would change the state), so the system becomes "dark." As the jumps are caused by $\hat{L}_\mu$ and the no-jump evolution is generated by $\hat{H}_{\text{eff}}$ [see Eq. (72)], a dark state should be a simultaneous eigenstate of $\hat{L}_\mu$ and $\hat{H}_{\text{eff}}$. In fact, this is both necessary and sufficient.

*Proof.* Suppose a normalized state $|\psi\rangle$ is a simultaneous eigenstate with $\hat{L}_\mu|\psi\rangle = \lambda_\mu|\psi\rangle$ and $\hat{H}_{\text{eff}}|\psi\rangle = \lambda_{\text{eff}}|\psi\rangle$. From Eq. (15) we find

$$\mathcal{L}(|\psi\rangle\langle\psi|) = \Big[ 2\,\text{Im}(\lambda_{\text{eff}}) + \sum_\mu |\lambda_\mu|^2 \Big]|\psi\rangle\langle\psi| \,. \tag{124}$$

However, using Eq. (14) we also have

$$\lambda_{\text{eff}} = \langle\psi|\hat{H}_{\text{eff}}|\psi\rangle = \langle\psi|\hat{H}|\psi\rangle - \frac{i}{2}\sum_\mu |\lambda_\mu|^2 \,, \tag{125}$$

$$\text{or} \quad 2\,\text{Im}(\lambda_{\text{eff}}) + \sum_\mu |\lambda_\mu|^2 = 0 \,, \tag{126}$$

since $\hat{H}$ is Hermitian. Therefore, $\mathcal{L}(|\psi\rangle\langle\psi|) = 0$, or $|\psi\rangle$ is a dark state. Next, let us check the converse. If $|\psi\rangle$ is a dark state, Eq. (13) gives

$$-i\hat{H}|\psi\rangle\langle\psi| + i|\psi\rangle\langle\psi|\hat{H} + \sum_\mu \left( \hat{L}_\mu|\psi\rangle\langle\psi|\hat{L}_\mu^\dagger - \frac{1}{2}\hat{L}_\mu^\dagger \hat{L}_\mu|\psi\rangle\langle\psi| - \frac{1}{2}|\psi\rangle\langle\psi|\hat{L}_\mu^\dagger \hat{L}_\mu \right) = 0 \,. \tag{127}$$

Sandwiching both sides between $\langle\psi|$ and $|\psi\rangle$, we get

$$\sum_\mu \Big[ \langle\psi|\hat{L}_\mu^\dagger \hat{L}_\mu|\psi\rangle - |\langle\psi|\hat{L}_\mu|\psi\rangle|^2 \Big] = 0 \,. \tag{128}$$

However, the expression inside brackets is simply $\langle\chi|\chi\rangle$ where $|\chi\rangle = \hat{L}_\mu|\psi\rangle - \langle\psi|\hat{L}_\mu|\psi\rangle|\psi\rangle$. Thus, Eq. (128) implies that $|\psi\rangle$ is an eigenstate of $\hat{L}_\mu$, i.e., $\hat{L}_\mu|\psi\rangle = \lambda_\mu|\psi\rangle$. Using this result in Eq. (127) and sandwiching between $\langle\phi|$ and $|\psi\rangle$, where $|\phi\rangle$ is any state orthogonal to $|\psi\rangle$, gives $\langle\phi|\hat{H}_{\text{eff}}|\psi\rangle = 0$, which means $|\psi\rangle$ is also an eigenstate of $\hat{H}_{\text{eff}}$. ∎

An iconic example of a dark state occurs in light-matter physics. Consider an atom with energy levels $|g\rangle$, $|e\rangle$, and $|r\rangle$, where the excited state $|e\rangle$ has a lifetime $1/\gamma$ beyond which it decays to $|g\rangle$, and $|g\rangle$, $|r\rangle$ represent either two hyperfine ground states or a ground state and a highly excited Rydberg state with a long lifetime. A "probe" laser resonantly couples $|g\rangle$ and $|e\rangle$ with Rabi frequency $\Omega_p$ and a "control" laser resonantly couples $|e\rangle$ and $|r\rangle$ with Rabi frequency $\Omega_c$. The system is described by $\hat{H} = \Omega_p |e\rangle\langle g| + \Omega_c |e\rangle\langle r| + \text{h.c.}$ and $\hat{L} = \sqrt{\gamma}|g\rangle\langle e|$. Under this drive and loss, the atom evolves to the dark state $|\psi\rangle \propto \Omega_c |g\rangle - \Omega_p |r\rangle$, which is a null eigenstate of both $\hat{L}$ and $\hat{H}$. As there is no amplitude in $|e\rangle$, the atom cannot absorb any photon from the probe laser, even though it is on resonance. In other words, the presence of the control beam turns the medium transparent for the probe beam, giving rise to electromagnetically induced transparency (EIT) [64].

In many-body systems, generic states, including eigenstates of an interacting Hamiltonian, have correlations between far-off sites. Such states are typically not eigenstates of local operators, so they are not stabilized as dark states under local dissipation. However, in certain cases it is possible to engineer local dissipation (one-site or two-site Lindblad operators) that stabilizes a specific entangled dark state [65, 66]. We will discuss such examples in Sec. 6.1.

> Q25.  The dark states we have encountered so far are null eigenstates of the Lindblad operators. Can you think of an example where this is not the case?

> Q26.  Consider two qubits with a spin-exchange coupling $\hat{H} = \hat{\sigma}_1^+ \hat{\sigma}_2^- + \hat{\sigma}_1^- \hat{\sigma}_2^+$ and a correlated dissipation $\hat{L} = \hat{\sigma}_1^- + \hat{\sigma}_2^+$. Show that they have an entangled dark state. Do all initial states evolve toward this dark state?

## 5.3  Symmetries and conservation laws

Conservation laws arising from symmetries play an important role in determining the steady states. In Hamiltonian systems there is a one-to-one correspondence between symmetry and conservation laws. If the Hamiltonian is invariant under a unitary operator, $\hat{U}^\dagger \hat{H} \hat{U} = \hat{H}$, or

$$\hat{H}\hat{U}|\psi\rangle = \hat{U}\hat{H}|\psi\rangle \quad \forall\, |\psi\rangle \,, \tag{129}$$

the different eigenspaces of $\hat{U}$ evolve independently, so the weights in these mutually orthogonal symmetry sectors are conserved. For a continuous symmetry $\hat{U} = e^{i\varphi\hat{J}}$, the generator $\hat{J}$ is a conserved observable. As we will see, in Lindblad dynamics this one-to-one correspondence is lost and symmetries come in two major varieties [67–69].

### 5.3.1  Weak symmetry

The natural generalization of Eq. (129) for density-matrix evolution is

$$\mathcal{L}\big(\hat{U}\hat{\rho}\hat{U}^\dagger\big) = \hat{U}\mathcal{L}(\hat{\rho})\hat{U}^\dagger \quad \forall\, \hat{\rho} \,, \tag{130}$$

which is equivalent to the condition $[\mathcal{L}, \mathcal{U}] = 0$, with $\mathcal{U}(\hat{\rho}) := \hat{U}\hat{\rho}\hat{U}^\dagger$. Then $\hat{U}$ is called a weak symmetry. Typically, this means the Hamiltonian is invariant and the Lindblad operators

unitarily transform among themselves under $\hat{U}$, although this is not necessary. The symmetry is called weak as it does not imply a conservation law despite a block diagonal structure of $\mathcal{L}$.

To see this, let $\lambda_\alpha = e^{i\theta_\alpha}$ be the eigenvalues of $\hat{U}$, and $\mathcal{H}_\alpha$ be the corresponding eigenspaces. The spectrum of $\mathcal{U}$ is formed of all the products $\lambda_\alpha \lambda_\beta^* = e^{i(\theta_\alpha - \theta_\beta)}$, as $\mathcal{U}(|\psi\rangle\langle\phi|) = \lambda_\alpha \lambda_\beta^* |\psi\rangle\langle\phi|$ for any $|\psi\rangle \in \mathcal{H}_\alpha$, $|\phi\rangle \in \mathcal{H}_\beta$. Let $\Lambda_\nu$ ($\nu = 1, 2, \dots$) denote the distinct eigenvalues of $\mathcal{U}$, with $\Lambda_1 = 1$ and $\Lambda_\nu \neq 1$ for $\nu > 1$. The corresponding operator spaces $\mathcal{B}_\nu$ evolve independently as $[\mathcal{L}, \mathcal{U}] = 0$. However, only $\mathcal{B}_1$ contains operators with non-vanishing trace and has to contain a steady state. This is because any physical state can be written as $\hat{\rho}(t) = \hat{\rho}_1(t) + (\text{other terms})$, where $\hat{\rho}_1(t) \in \mathcal{B}_1$ and has trace 1, while the other terms are in $\mathcal{B}_{\nu>1}$, and it is entirely possible for these other terms to vanish as $t \to \infty$. Thus, a weak symmetry does not imply steady-state degeneracy, although it can be used to block diagonalize $\mathcal{L}$.

The steady-state degeneracy also gives the number of independent conserved quantities. This is because a conserved operator $\hat{O}$ satisfies $\mathcal{L}^\dagger(\hat{O}) = 0$, since $d\langle\hat{O}\rangle/dt = \langle\mathcal{L}^\dagger(\hat{O})\rangle$ [Eq. (35)]. However, as $\mathcal{L}$ and $\mathcal{L}^\dagger$ share the same spectrum, the number of such operators must equal the number of steady states of $\mathcal{L}$. For a unique steady state, only $\text{Tr}(\hat{\rho})$ is conserved, corresponding to $\mathcal{L}^\dagger(\hat{1}) = 0$.

Additionally, from Eq. (130), if $\mathcal{L}(\hat{\rho}) = 0$ then $\mathcal{L}(\hat{U}\hat{\rho}\hat{U}^\dagger) = 0$, i.e., the steady-state manifold is invariant under $\hat{U}$. Thus, even though a weak symmetry does not lead to multiple steady states, it constrains the form of the steady states.

A simple instance of weak symmetry occurs in a damped harmonic oscillator or lossy cavity, described by $\hat{H} = \omega \hat{a}^\dagger \hat{a}$ and $\hat{L} = \sqrt{\gamma}\hat{a}$. The steady state is just the vacuum and only the trace is conserved. However, the operator $\hat{U} = e^{i\theta\hat{a}^\dagger\hat{a}}$ constitutes a weak symmetry as $\hat{U}\hat{H}\hat{U}^\dagger = \hat{H}$ and $\hat{U}\hat{L}\hat{U}^\dagger = e^{-i\theta}\hat{L}$, which implies Eq. (130). Since $\mathcal{U}(|m\rangle\langle n|) = e^{i\theta(m-n)}|m\rangle\langle n|$, where $|m\rangle$, $|n\rangle$ are Fock states, the eigenspaces $\mathcal{B}_\nu$ are composed of operators $|m\rangle\langle n|$ where $m - n$ is a given integer. In other words, the different diagonals of the density matrix in the Fock basis evolve independently, which greatly simplifies the computation. Note the same holds for a nonlinear cavity or anharmonic oscillator, e.g., $\hat{H} = \omega \hat{a}^\dagger \hat{a} + \kappa \hat{a}^\dagger \hat{a}^\dagger \hat{a}\hat{a}$.

For a many-body example of a weak symmetry, consider a boundary-driven *XXZ* spin-1/2 chain described by

$$\hat{H} = \sum_{i=1}^{N-1} \left( \hat{\sigma}_i^x \hat{\sigma}_{i+1}^x + \hat{\sigma}_i^y \hat{\sigma}_{i+1}^y + \Delta \hat{\sigma}_i^z \hat{\sigma}_{i+1}^z \right) \quad \text{and} \quad \hat{L}_1 = \sqrt{\gamma_+}\,\hat{\sigma}_1^+, \; \hat{L}_2 = \sqrt{\gamma_-}\,\hat{\sigma}_N^-. \quad (131)$$

Clearly, $\hat{H}$ has a reflection symmetry $\hat{R}$ that exchanges site $i$ with site $N-i+1$. If one combines this reflection with a spin flip on all sites, $\hat{U} = \hat{R}\prod_i \hat{\sigma}_i^x$, then $\hat{H}$ is unaltered and $\hat{L}_1$, $\hat{L}_2$ get interchanged provided $\gamma_+ = \gamma_-$. Thus, for equal pump and loss rates, $\hat{U}$ is a weak symmetry. This system is known to have a unique, full-rank steady state [63,70]. As $\hat{U}$ has eigenvalues $\pm 1$, the Liouvillian reduces to two blocks $\mathcal{B}_1 = \{+1, +1\}\bigoplus\{-1, -1\}$ and $\mathcal{B}_1 = \{+1, -1\}\bigoplus\{-1, +1\}$, of which the former contains the steady state. In addition, the steady state is invariant under $\hat{U}$, which means $\langle\hat{\sigma}_i^z\rangle = -\langle\hat{\sigma}_{N-i+1}^z\rangle$, $\langle\hat{\sigma}_i^x \hat{\sigma}_j^x\rangle = \langle\hat{\sigma}_{N-i+1}^x \hat{\sigma}_{N-j+1}^x\rangle$, and so on.

---

Q27. Find a weak symmetry for a spin-$S$ with $\hat{H} = \omega\hat{S}^z$ and $\hat{L} = \sqrt{\gamma}\hat{S}^+$.

---

> Q28.  Consider a fermion chain with periodic boundary conditions and loss at each site, described by
>
> $$\hat{H} = -J \sum_i \left( \hat{c}_i^\dagger \hat{c}_{i+1} + \hat{c}_{i+1}^\dagger \hat{c}_i \right) + \Delta \sum_i \left( \hat{c}_i \hat{c}_{i+1} + \hat{c}_{i+1}^\dagger \hat{c}_i^\dagger \right) + g \sum_i \hat{n}_i \hat{n}_{i+1} \qquad (132)$$
>
> and $\hat{L}_i = \sqrt{\gamma}\,\hat{c}_i$, with $\hat{n}_i := \hat{c}_i^\dagger \hat{c}_i$.
>
> 1) Show that the fermion-number parity $\hat{U} = (-1)^{\hat{N}}$, where $\hat{N} = \sum_i \hat{n}_i$, is a weak symmetry. How does it constrain the steady state (assuming it is unique)?
>
> 2) The setup has another weak symmetry. Can you find it?
>
> 3) How many disjoint blocks does $\mathcal{L}$ have based on these two symmetries?

### 5.3.2  Strong symmetry

A strong symmetry commutes with each of generators, $\hat{H}$ and $\hat{L}_\mu$, not just with the full Liouvillian, i.e.,

$$[\hat{U}, \hat{H}] = 0 \quad \text{and} \quad [\hat{U}, \hat{L}_\mu] = 0 \ \forall \mu \,. \qquad (133)$$

As $\hat{U}$ is unitary, it also commutes with $\hat{L}_\mu^\dagger$. Thus, Eq. (133) and Eq. (29) imply $\mathcal{L}\big(\hat{U}\hat{\rho}\big) = \hat{U}\mathcal{L}(\hat{\rho})$ and $\mathcal{L}\big(\hat{\rho}\hat{U}^\dagger\big) = \mathcal{L}(\hat{\rho})\hat{U}^\dagger \ \forall \hat{\rho}$. Defining the superoperators $\mathcal{U}_l(\hat{\rho}) := \hat{U}\hat{\rho}$ and $\mathcal{U}_r(\hat{\rho}) := \hat{\rho}\hat{U}^\dagger$, we have $[\mathcal{L}, \mathcal{U}_l] = [\mathcal{L}, \mathcal{U}_r] = [\mathcal{U}_l, \mathcal{U}_r] = 0$. So, $\mathcal{L}$ can be block diagonalized in the joint eigenspaces $\mathcal{B}_{\alpha,\beta}$ of $\mathcal{U}_l$ and $\mathcal{U}_r$, given by $\mathcal{U}_l(|\psi\rangle\langle\phi|) = e^{i\theta_\alpha}|\psi\rangle\langle\phi|$ and $\mathcal{U}_r(|\psi\rangle\langle\phi|) = e^{-i\theta_\beta}|\psi\rangle\langle\phi|$ for any $|\psi\rangle \in \mathcal{H}_\alpha$ and $|\phi\rangle \in \mathcal{H}_\beta$. Of these, only the diagonal blocks ($\beta = \alpha$) contain operators with non-vanishing trace and must contain a steady state. Thus, a strong symmetry leads to at least as many steady states as the number of symmetry sectors. Furthermore, as the dynamics is trace preserving, the weight in each sector is conserved. For a continuous symmetry $\hat{U} = e^{i\varphi\hat{J}}$, the generator $\hat{J}$ is a conserved observable.

We see that like a Hamiltonian symmetry, a strong symmetry leads to a conservation law and decoupled symmetry sectors. However, here this relation is one way: Conserved quantities or degenerate steady states can also appear without a strong symmetry as long as the dynamics is Davies reducible. In such cases the steady states do not have full rank [63]. We have already seen an example of this in the two-qubit problem of Q24. The existence of a strong symmetry is a stronger condition, called Evans reducibility [71].

As an example, consider a two-site Bose-Hubbard model (also called Bose-Hubbard dimer) with incoherent hopping, described by [72]

$$\hat{H} = -J\big(\hat{a}^\dagger \hat{b} + \hat{b}^\dagger \hat{a}\big) + \frac{U}{2} \sum_{i=a,b} \hat{n}_i(\hat{n}_i - 1) \quad \text{and} \quad \hat{L}_1 = \sqrt{\gamma_l}\,\hat{a}^\dagger \hat{b}, \ \hat{L}_2 = \sqrt{\gamma_r}\,\hat{b}^\dagger \hat{a}\,. \quad (134)$$

Clearly, $\hat{H}$, $\hat{L}_1$, and $\hat{L}_2$ all conserve the particle number $\hat{N} = \hat{n}_a + \hat{n}_b$, so $\hat{U} = e^{i\varphi\hat{N}}$ is a strong symmetry. As a result, the different sectors of $\hat{N}$ are uncoupled, each having at least one steady state. Furthermore, in a given sector, the problem maps to a spin-$\frac{N}{2}$ with $\hat{S}_z := \frac{1}{2}(\hat{n}_a - \hat{n}_b)$ and $\hat{S}_+ := \hat{a}^\dagger \hat{b}$, for which $\hat{H} = -2J\hat{S}_x + U\hat{S}_z^2$ (up to a constant) and $\hat{L}_1 = \sqrt{\gamma_l}\,\hat{S}_+$, $\hat{L}_2 = \sqrt{\gamma_r}\,\hat{S}_-$. For $J = 0$ this setup has a weak symmetry $e^{i\varphi\hat{S}_z}$, which decouples the diagonals in the $\hat{S}_z$ basis. For $\gamma_l = \gamma_r$, exchanging the two sites (equivalently, exchanging $\pm\hat{S}_z$) is another weak symmetry. This example shows a single setup can have multiple strong and weak symmetries.

Take a different example of two interacting spins coupled to a lossy cavity, modeled by

$$\hat{H} = B\big(\hat{S}_1^z + \hat{S}_2^z\big) + J\hat{\mathbf{S}}_1 \cdot \hat{\mathbf{S}}_2 + g\big(\hat{S}_1^x + \hat{S}_2^x\big)\big(\hat{a} + \hat{a}^\dagger\big), \tag{135}$$

and $\hat{L} = \sqrt{\gamma}\,\hat{a}$. Both $\hat{H}$ and $\hat{L}$ remain invariant under exchanging the two spins, hence this is a strong symmetry. In addition, $\hat{H}$ depends only on the total-spin components, so the model has a strong $SU(2)$ symmetry, which conserves the total spin $\hat{\mathbf{S}}^2$. The exchange parity $\hat{P}$ commutes with $\hat{\mathbf{S}}^2$; in fact, each sector of $\hat{\mathbf{S}}^2$ has a definite parity. Thus, the dynamics simply splits into the different total-spin sectors, giving rise to at aleast $2S + 1$ steady states. In particular, the singlet (total spin zero) does not interact with the cavity and is a dark state. Notice that, even if the $SU(2)$ symmetry is lost, e.g., if one only has a $z$-$z$ coupling $J\hat{S}_1^z\hat{S}_2^z$, the singlet would remain dark for qubits ($S = 1/2$) as it is the only anti-symmetric ($P = -1$) state.

It is possible to have multiple strong symmetries that do not commute (called non-Abelian symmetries). For example, if one has three spins in the above example with

$$\hat{H} = B\sum_{i=1}^3 \hat{S}_i^z + J\big(\hat{S}_1^z\hat{S}_2^z + \hat{S}_2^z\hat{S}_3^z + \hat{S}_3^z\hat{S}_1^z\big) + g\big(\hat{a} + \hat{a}^\dagger\big)\sum_{i=1}^3 \hat{S}_i^x, \tag{136}$$

then exchanging any two spins is a strong symmetry. However, these pairwise exchanges do not commute but form a permutation group. Here the maximal decomposition of the dynamics occurs in an irreducible representation of the group. The resulting lower bound on the steady-state degeneracy is derived in [73]. Strong symmetries can be used to stabilize long-range entanglement [33] and control quantum transport [74].

---

Q29. Consider the setup in Q26 where two qubits have a spin-exchange interaction and a correlated dissipation $\hat{L} = \hat{\sigma}_1^- + \hat{\sigma}_2^+$.

    1) Identify a strong symmetry.

    2) What are the uncoupled blocks of the Lindblad dynamics?

    3) What are the steady states?

    4) What is the final state if both qubits are initially pointing along $+z$?

    5) Can you modify $\hat{L}$ such that every initial state reaches an entangled dark state?

    6) Can this dark state be maximally entangled?

---

Q30. Consider a spin-1 chain with exchange interactions (Lai-Sutherland model) subjected to incoherent boundary drives, described by [75]

$$\hat{H} = \sum_{i=1}^{N-1} \sum_{m,m'} |m\rangle\langle m'|_i \otimes |m'\rangle\langle m|_{i+1} \quad \text{and} \quad \hat{L}_1 = \big(\hat{S}_1^+\big)^2, \ \hat{L}_2 = \big(\hat{S}_N^-\big)^2, \tag{137}$$

where $m \in \{1, 0, -1\}$ labels the three eigenstates of $\hat{S}^z$.

    1) Find a strong symmetry. What is conserved?

    2) What is the minimum number of steady states?

    3) Find a weak symmetry.

    4) Is each steady state invariant under the weak symmetry?

### 5.3.3 Dynamical symmetries

There is a different type of symmetry that gives rise to purely imaginary eigenvalues, leading to persistent oscillations as opposed to multiple steady states. These are called strong dynamical symmetries [76]. They arise when there exists an operator $\hat{A}$ such that

$$[\hat{A}, \hat{L}_\mu] = [\hat{A}, \hat{L}_\mu^\dagger] = 0 \ \forall \mu \quad \text{and} \quad [\hat{A}, \hat{H}] = \omega \hat{A}, \tag{138}$$

where $\omega$ is nonzero and real valued. One can use such an operator to generate eigenstates of $\mathcal{L}$ whose eigenvalues differ by multiples of $i\omega$. To see this, suppose $\hat{\rho}$ is a right eigenstate with eigenvalue $\lambda$. Using the expression of $\mathcal{L}$ [Eq. (13)] and Eq. (138), we find

$$\mathcal{L}(\hat{A}\hat{\rho}) = \hat{A}\mathcal{L}(\hat{\rho}) + i\hat{A}[\hat{H}, \hat{\rho}] - i[\hat{H}, \hat{A}\hat{\rho}] = (\lambda + i\omega)\hat{A}\hat{\rho}. \tag{139}$$

Thus, $\hat{A}\hat{\rho}$ is a right eigenstate with eigenvalue $\lambda + i\omega$. Similarly, one finds $\mathcal{L}(\hat{\rho}\hat{A}^\dagger) = (\lambda - i\omega)\hat{\rho}\hat{A}^\dagger$. Iterating this process generates a family of right eigenstates, $\hat{A}^m \hat{\rho}(\hat{A}^\dagger)^n$ with $m, n \in \mathbb{Z}_{\geq 0}$ and eigenvalues $\lambda + i(m-n)\omega$. In particular, acting on a steady state $\hat{\rho}_{ss}$ generates purely imaginary eigenvalues $i(m-n)\omega$ corresponding to the right eigenstates $\hat{\rho}_{m,n} = \hat{A}^m \hat{\rho}_{ss}(\hat{A}^\dagger)^n$.

The left eigenstates are found similarly by noting that $\mathcal{L}^\dagger(\hat{O}) = \lambda\hat{O}$ implies [see Eq. (34)] $\mathcal{L}^\dagger(\hat{A}\hat{O}) = (\lambda - i\omega)\hat{A}\hat{O}$ and $\mathcal{L}^\dagger(\hat{O}\hat{A}^\dagger) = (\lambda + i\omega)\hat{O}\hat{A}^\dagger$. In conjunction with $\mathcal{L}^\dagger(\hat{1}) = 0$, this gives $\mathcal{L}^\dagger(\hat{A}^m(\hat{A}^\dagger)^n) = i(n-m)\omega\hat{A}^m(\hat{A}^\dagger)^n$. Thus, $\langle \hat{A}^m(\hat{A}^\dagger)^n \rangle$ oscillates as $e^{-i(m-n)\omega t}$ [see Eq. (35)]. If all imaginary eigenvalues are of this form, the long-time density matrix oscillates periodically at frequency $\omega$ for generic initial states. Note, however, that one does not necessarily have all harmonics of $\omega$ as the iteration may terminate, i.e., $\hat{A}^m = 0$ for $m > m_c$. Indeed, this must be the case if the Hilbert-space dimension is finite.

While we have shown that a strong dynamical symmetry is sufficient to have long-time oscillations, one can also show that it is necessary provided $\mathcal{L}$ has a full-rank steady state [77]. Additionally, a system with a finite Hilbert-space dimension and a unique steady state cannot have purely imaginary eigenvalues [63].

As an example of a strong dynamical symmetry, consider a Hubbard model with on-site dephasing [76],

$$\hat{H} = -\sum_{\langle i,j \rangle, \sigma = \uparrow, \downarrow} \hat{c}_{i,\sigma}^\dagger \hat{c}_{j,\sigma} + \sum_j \left[ U\hat{n}_{j,\uparrow}\hat{n}_{j,\downarrow} + \varepsilon_j \hat{n}_j - \frac{B}{2}(\hat{n}_{j,\uparrow} - \hat{n}_{j,\downarrow}) \right] \quad \text{and} \quad \hat{L}_j = \sqrt{\gamma}\,\hat{n}_j, \tag{140}$$

where $\hat{c}_{j,\sigma}$ annihilates a fermion of spin $\sigma$ at site $j$, $\hat{n}_{j,\sigma} := \hat{c}_{j,\sigma}^\dagger \hat{c}_{j,\sigma}$ are the site occupations, $\hat{n}_j := \hat{n}_{j,\uparrow} + \hat{n}_{j,\downarrow}$, $\varepsilon_j$ are on-site disorder potentials, $B$ is a uniform magnetic field, and $\gamma$ is a uniform dephasing rate. Both $\hat{H}$ and $\hat{L}_j$ conserve the total particle number $\hat{N} = \sum_j \hat{n}_j$, the total $z$-magnetization $\hat{S}^z = \sum_j(\hat{n}_{j,\uparrow} - \hat{n}_{j,\downarrow})/2$, as well as the total spin $\hat{\mathbf{S}}^2 = (\hat{S}^z)^2 + \frac{1}{2}(\hat{S}^+\hat{S}^- + \hat{S}^-\hat{S}^+)$, where $\hat{S}^+ = \sum_j \hat{c}_{j,\uparrow}^\dagger \hat{c}_{j,\downarrow}$ and $\hat{S}^- = \sum_j \hat{c}_{j,\downarrow}^\dagger \hat{c}_{j,\uparrow}$. (Check this yourself using the anticommutation $\{\hat{c}_{i,\sigma}, \hat{c}_{j,\sigma'}\} = \delta_{i,j}\delta_{\sigma,\sigma'}$.) Within a common eigenspace of $\hat{N}$, $\hat{S}^z$, and $\hat{\mathbf{S}}^2$ (which commute with each other), the Hermitian Lindblad operators cause heating to a completely mixed steady state. All sites appear on an equal footing in such a projector, so the steady states are spatially uniform, even though $\hat{H}$ has disorder. Crucially, $\hat{S}^+$ is a strong dynamical symmetry satisfying $[\hat{S}^+, \hat{H}] = B\hat{S}^+$, which gives rise to purely imaginary eigenvalues at $i(m-n)B$, corresponding

to the right eigenstates $\hat{\rho}_{m,n} = (\hat{S}^+)^m \hat{\rho}_{ss} (\hat{S}^-)^n$. As $\hat{S}^\pm$ changes $\hat{S}^z$ by $\pm 1$ without affecting $\hat{N}$ or $\hat{\mathbf{S}}^2$, $\hat{\rho}_{m,n}$ describes coherence between two eigenspaces that differ by $\Delta S^z = m - n$. For a given value of $N$ and total-spin quantum number $S \leq N/2$, $S^z$ varies from $-S$ to $S$ in steps of 1, so we get $2S + 1$ steady states and imaginary eigenvalues up to $\pm i(2S+1)B$, as shown in Fig. 4(a) for $N = 4$ and $S = 1$. Since $\hat{S}^\pm$ and $\hat{\rho}_{ss}$ are spatially uniform, so are all these eigenstates.

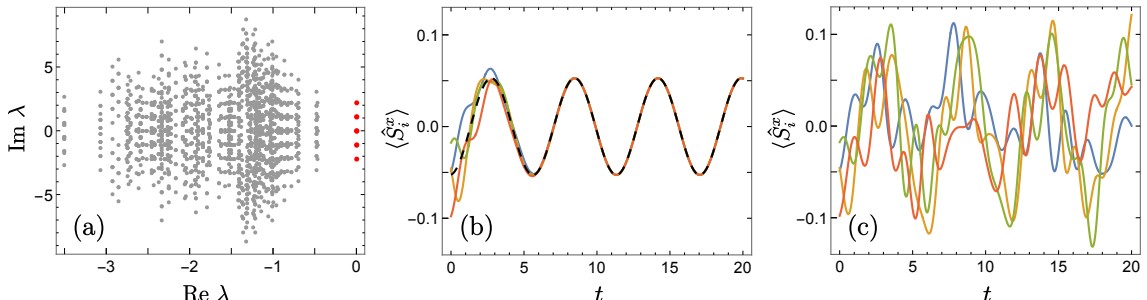

Figure 4: (a) Lindblad spectrum for Eq. (140) with $U = 1.4$, $B = 1.1$, $\gamma = 1$, and $\varepsilon_j \in [-0.3, 0.3]$ for $N = 4$, $S = 1$, and 4 sites. (b) Evolution of $\langle \hat{S}_i^x \rangle = \text{Re} \langle \hat{c}_{i,\uparrow}^\dagger \hat{c}_{i,\downarrow} \rangle$, $i = 1, 2, 3, 4$, starting from a random state. Dashed curve shows the (uniform) oscillation resulting from the imaginary eigenvalues (red dots). (c) Evolution for $\gamma = 0$.

The dynamical symmetry leads to persistent oscillations, with $\langle (\hat{S}^+)^m (\hat{S}^-)^n \rangle \sim e^{-i(m-n)Bt}$. In particular, with $\hat{S}^\pm := \hat{S}^x \pm i\hat{S}^y$, $\langle \hat{S}^x \rangle(t) = \langle \hat{S}^x \rangle(0) \cos(Bt) + \langle \hat{S}^y \rangle(0) \sin(Bt)$. As the non-decaying eigenstates are translationally invariant, the spins at different sites synchronize at long times, each oscillating as $\langle \hat{S}_i^x \rangle = \frac{1}{L} \langle \hat{S}^x \rangle$, where $L$ is the number of sites. This is shown in Fig. 4(b). The synchronization is absent for a purely Hamiltonian dynamics [Fig. 4(c)], since the latter has many incommensurate frequencies (energy gaps). The dissipation damps out all but one of these modes—selected by the dynamical symmetry—leading to synchronization. For more connections to synchronization see [77,78]. Note that synchronization can also arise without such dynamical symmetries [79,80].

---

Q31. Consider two interacting qutrits (spin-1) with on-site quadratic dephasing:

$$\hat{H} = \hat{S}_1^+ \hat{S}_2^- + \hat{S}_1^- \hat{S}_2^+ + \Delta \hat{S}_1^z \hat{S}_2^z + \omega(\hat{S}_1^z + \hat{S}_2^z) \quad \text{and} \quad \hat{L}_1 = \sqrt{\gamma}\,(\hat{S}_1^z)^2, \;\; \hat{L}_2 = \sqrt{\gamma}\,(\hat{S}_2^z)^2 \,. \quad (141)$$

     1) Show that $|\phi\rangle := |\uparrow\downarrow\rangle - |\downarrow\uparrow\rangle$ is a dark state.

     2) Find a strong symmetry. What steady states does it give rise to?

     3) Show that the following are strong dynamical symmetries:

$$\hat{A}_1 = |\uparrow 0\rangle\langle\downarrow 0| + |0\uparrow\rangle\langle 0\downarrow|, \quad \hat{A}_2 = |\uparrow\uparrow\rangle\langle\downarrow\downarrow|, \quad \hat{A}_3 = |\phi\rangle\langle\uparrow\uparrow|, \quad \hat{A}_4 = |\phi\rangle\langle\downarrow\downarrow|. \quad (142)$$

     4) Find the Lindblad spectrum for $\Delta = 0.7$ and $\omega = \gamma = 1$. Can you explain all the non-decaying eigenstates?

     5) Starting from the pure, product state $|+\rangle \otimes |+\rangle$, where $|+\rangle := \frac{1}{\sqrt{3}}(|\uparrow\rangle + |0\rangle + |\downarrow\rangle)$, visualize the long-time dynamics of the density matrix (in the $z$ basis). Can you explain the oscillating off-diagonal terms?

---

Note that the condition $[\hat{A}, \hat{H}] = \omega \hat{A}$ with $\omega \neq 0$ implies that $\hat{A}$ is traceless. It also implies

that $\hat{A}$ is not invertible, as otherwise one can write $\hat{H} - \hat{A}^{-1}\hat{H}\hat{A} = \omega\hat{1}$ which, upon taking the trace, gives $\omega = 0$. Therefore, $\hat{A}$ cannot be unitary, unlike a strong symmetry.

There are also anti-unitary symmetries, which preserve the norm but involve complex conjugation. A well-known example of such a symmetry in Hamiltonian systems is time reversal, which has generalizations to Lindblad dynamics. In particular, in systems with balanced gain and loss one can define a parity-time reversal symmetry as $\mathcal{PT}(\hat{O}) := \hat{P}\hat{O}^\dagger\hat{P}$, where $\hat{P}$ involves a reflection that exchanges the gain and loss sites, such that $\mathcal{L}\big[\mathcal{PT}(\hat{H}), \{\mathcal{PT}(\hat{L}_\mu)\}\big] = \mathcal{L}\big[\hat{H}, \{\hat{L}_\mu\}\big]$ [81]. The steady state of such systems exhibit a symmetry-breaking transition, as we will see in Sec. 6.3. There are also alternative definitions of such symmetries [82].

## 5.4   Decoherence-free subspaces and subsystems

Non-decaying eigenstates can arise if a subspace of the Hilbert space is invariant under both $\hat{H}$ and $\hat{L}_\mu$, even if there is no strong symmetry. Specifically, if a subspace is unaffected by dissipation and evolves unitarily under $\hat{H}$, it is called a decoherence-free subspace (DFS) [9]. For this, the subspace must be closed under $\hat{H}$, i.e., spanned by a set of energy eigenstates, $\hat{H}|\psi_j\rangle = E_j|\psi_j\rangle$, and any superposition of these states must be unaffected by the dissipators, i.e., $\sum_\mu \mathcal{D}[\hat{L}_\mu]\big(|\psi_j\rangle\langle\psi_k|\big) = 0 \ \forall j, k$. Typically, one requires this immunity irrespective of the loss rates, which means the dissipators should vanish individually (for each $\mu$). This is satisfied if $\hat{L}_\mu|\psi_j\rangle = 0 \ \forall \mu, j$ or, more generally, $\hat{L}_\mu|\psi_j\rangle = c_\mu|\psi_j\rangle$ and $\hat{L}_\mu^\dagger|\psi_j\rangle = c_\mu^*|\psi_j\rangle \ \forall \mu, j$ [9].

The dynamics within a DFS leads to the eigenstates $\mathcal{L}(|E_j\rangle\langle E_k|) = -\mathrm{i}(E_j - E_k)|E_j\rangle\langle E_k|$. For a $d$-dimensional subspace, this gives at least $d$ dark states: $|E_j\rangle$. The coherence between these states oscillates as $e^{-\mathrm{i}(E_j - E_k)t}$. Unlike the non-decaying states arising from a strong dynamical symmetry, here the manifold is spanned by pure states. Also, the eigenvalues are generally not equally spaced on the imaginary axis. If the system does not have other non-decaying eigenstates, it is driven to the DFS at long time, which is particularly useful for encoding quantum information—any "error" that takes one out of this subspace is automatically corrected by the dissipation. Furthermore, one can manipulate this information by applying unitary gates that rotate the state within the subspace [83, 84].

A simple instance of a DFS occurs for a uniform tight-binding model with open boundary conditions and odd number of sites, with particle loss at the center site.

$$\hat{H} = -\sum_{i=-l}^{l-1}\big(\hat{c}_i^\dagger\hat{c}_{i+1} + \hat{c}_{i+1}^\dagger\hat{c}_i\big) \quad\text{and}\quad \hat{L} = \sqrt{\gamma}\,\hat{c}_0\,. \tag{143}$$

All of the odd single-particle eigenfunctions of $\hat{H}$ vanish at the center site and do not experience the loss. Thus, any many-body state where the particles occupy only these odd wave functions is unaffected by loss, forming an exponentially large DFS. If the initial state has occupations in both even and odd wave functions, the occupations of the even modes die out, but those of the odd modes remain. For example, if one starts with a fermion on each site, one is finally left with $l$ fermions in the $l$ odd modes, i.e., half the particles survive indefinitely even though there is a "hole" at the center!

It is also possible to have multiple disjoint DFSs. For instance, consider two qubits with

$$\hat{H} = \hat{\sigma}_1^+\hat{\sigma}_2^- + \hat{\sigma}_1^-\hat{\sigma}_2^+ \quad\text{and}\quad \hat{L} = \hat{\sigma}_1^z + \hat{\sigma}_2^z\,. \tag{144}$$

The energy eigenstates are $\hat{H}|\uparrow\uparrow\rangle = \hat{H}|\downarrow\downarrow\rangle = 0$ and $\hat{H}|\pm\rangle = \pm|\pm\rangle$, where $|\pm\rangle := \frac{1}{\sqrt{2}}(|\uparrow\downarrow\rangle \pm |\downarrow\uparrow\rangle)$. Of these, $|\pm\rangle$ are null states of $\hat{L}$ and form a two-dimensional DFS, whereas $|\uparrow\uparrow\rangle$ and $|\downarrow\downarrow\rangle$ are also eigenstates with different eigenvalues, so they form separate one-dimensional DFSs.

---

Q32.  Consider 4 interacting qubits subject to a collective loss on the first two:

$$\hat{H} = \sum_{i<j} J_{i,j}\hat{\mathbf{S}}_i \cdot \hat{\mathbf{S}}_j - B\sum_i \hat{S}_i^z \quad \text{and} \quad \hat{L} = \sqrt{\gamma}\,(\hat{S}_1^- + \hat{S}_2^-)\,, \qquad (145)$$

where $J_{i,j}$ are unequal real numbers.  Find the disjoint decoherence-free subspaces and their dimensions.

---

A DFS is not the most general way to encode quantum information that is protected from dissipation. One can also use a decoherence-free or noiseless subsystem [85], which occurs if the Hamiltonian has eigenstates with a tensor-product structure, $|E_{i,j}\rangle = |\psi_i\rangle \otimes |\phi_j\rangle$, where the two parts describe subsystems $A$ and $B$, respectively, and the loss operators act only on $B$, i.e., $\hat{L}_\mu|E_{i,j}\rangle = |\psi_i\rangle \otimes (\hat{l}_\mu|\phi_j\rangle)$ and $\hat{L}_\mu^\dagger|E_{i,j}\rangle = |\psi_i\rangle \otimes (\hat{l}_\mu^\dagger|\phi_j\rangle)$. Then any information encoded in the states $\{|\psi_i\rangle\}$ is preserved, leading to non-decaying eigenstates of the form $|\psi_j\rangle\langle\psi_k| \otimes \hat{\rho}_{\rm ss}$.

For example, consider a center-of-mass preserving hopping model subject to pump and loss at the ends [86]

$$\hat{H} = \sum_{i=2}^{N-2} \hat{c}_{i-1}^\dagger \hat{c}_i \hat{c}_{i+1} \hat{c}_{i+2}^\dagger + \hat{U}(\{\hat{n}_i\}) \quad \text{and} \quad \hat{L}_1 = \hat{c}_1^\dagger\,, \ \hat{L}_2 = \hat{c}_N\,, \ \hat{L}_3 = \sqrt{\gamma}\hat{c}_1\,, \ \hat{L}_3 = \sqrt{\gamma}\hat{c}_N^\dagger\,, \ (146)$$

where $\hat{c}_i$ annihilates a fermion at site $i$ and $\hat{U}$ is an arbitrary function of the site occupations $\hat{n}_j = \hat{c}_j^\dagger \hat{c}_j$. Because of the constrained hopping, the bulk is shielded from the boundary dissipation for any state of the form

$$\boxed{0/1}\,\boxed{111}\,\boxed{1 \qquad\qquad\qquad 0}\,\boxed{000}\,\boxed{0/1}\,. \qquad (147)$$

Thus, information in the bulk is preserved, which constitutes a noiseless subsystem, while the end sites continue to flip between 0 and 1, taking the system to a combination of mixed states

$$\hat{\rho} \propto (\gamma|0\rangle\langle 0| + |1\rangle\langle 1|) \otimes |\psi_j^{\rm bulk}\rangle\langle\psi_k^{\rm bulk}| \otimes (|0\rangle\langle 0| + \gamma|1\rangle\langle 1|)\,, \qquad (148)$$

where $|\psi_j^{\rm bulk}\rangle$ describe the bulk. Note that a noiseless subsystem does not have to be associated with a spatial partitioning. It can also represent a set of quantum numbers, e.g., the total-spin quantum number may be immune to dissipation, although the overall state is affected [83].

It is possible for the same system to exhibit all of the structures we have discussed, including strong symmetries, decoherence-free subspaces, and noiseless subsystems. For a concrete example based on a variant of the above center-of-mass preserving model, see [86].

# 6  Novel Physical Phenomena

Let us conclude by highlighting some intriguing dynamical phenomena that can arise out of dissipative dynamics and have led to fruitful research directions.

## 6.1   Reservoir engineering

Usually, environmental coupling or dissipation is thought of as a menace, which destroys quantum correlations. The idea behind reservoir engineering or dissipative state preparation is that this does not have to be the case. Instead, one can engineer the environment such that it drives the system to a desired quantum state. See [2, 6, 7] for recent reviews.

An experimental demonstration of this idea is found in [87], where a combination of drive and loss allowed the preparation of a Mott insulator of photons. In a traditional "band" insulator, electrons cannot hop as there is no partially filled energy band. By contrast, a Mott insulator stems from strong interactions: particles cannot hop due to a large energy cost for putting two of them at the same site. This is exemplified by the Bose-Hubbard model

$$\hat{H} = -J \sum_{\langle i,j \rangle} \hat{a}_i^\dagger \hat{a}_j + U \sum_j \frac{1}{2} \hat{n}_j (\hat{n}_j - 1) \, , \tag{149}$$

where $\hat{n}_j := \hat{a}_j^\dagger \hat{a}_j$. The Mott insulator corresponds to the ground state for $U/J \to \infty$, where each site has one particle (or some integer filling) and they cannot hop. This model can be realized in a superconducting circuit as an array of microwave resonators that can exchange photons. The interaction $U$ arises from a nonlinearity in the spectrum of each resonator, such that the energy of the $n$-th level ($n$ photons) is $\varepsilon_n = n\omega_c + \frac{U}{2} n(n-1)$, where $\omega_c$ is the resonator frequency. For $|U| \gg J$, the Mott insulator can be realized by irreversibly injecting photons of frequency $\omega_c$ at an end site and letting them hop until all resonators are filled. The irreversible (i.e., dissipative) photon pump requires some reservoir engineering. In [87] this was realized by driving the end site at two-photon resonance, i.e., with a laser of frequency $\omega = \omega_c + U/2$, which can inject or remove two photons. In addition, this driven site is coupled to a lossy resonator of frequency $\omega_c + U$, such that one of the two injected photons can resonantly hop to the lossy site and leak out (Fig. 5). When the drive and loss are sufficiently fast compared to $J$, this mechanism ensures that the end site is stabilized at one photon, which can hop to a neighboring site if it is empty, eventually filling up each site with one photon.

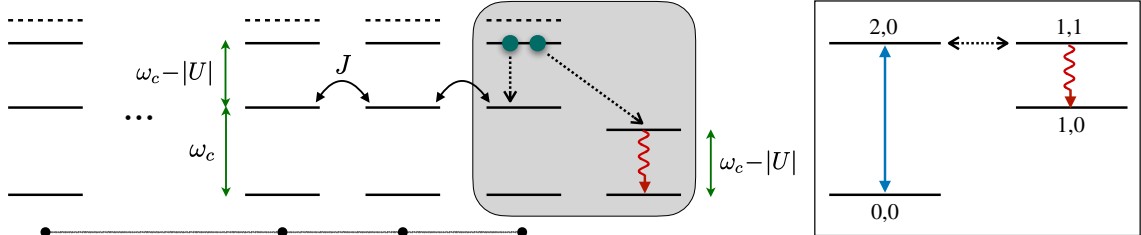

Figure 5: Using drive and loss to create a photonic Mott insulator [87]. Each site is a resonator with frequency $\omega_c$ and nonlinearity $U < 0$. The end site is driven at two-photon resonance and coupled to a lossy resonator of frequency $\omega_c - |U|$. The box shows energy levels of the last two resonators labeled by their photon numbers.

One can also use dissipation to generate states with long-range entanglement. As an example, consider a variant of the setup in Q26 and Q29: Two qubits with exchange interaction

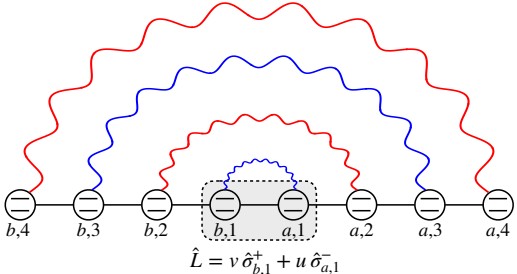

$$\hat{L} = v\,\hat{\sigma}^+_{b,1} + u\,\hat{\sigma}^-_{a,1}$$

Figure 6: Spin-1/2 XY chain driven to a rainbow state by a central two-site dissipation. Red and blue lines denote the Bell states $v|\uparrow_a\uparrow_b\rangle \pm u|\downarrow_a\downarrow_b\rangle$. Adapted from [66].

and a correlated dissipation [66]:

$$\hat{H} = \hat{\sigma}^+_a\hat{\sigma}^-_b + \hat{\sigma}^-_a\hat{\sigma}^+_b \quad \text{and} \quad \hat{L} = u\,\hat{\sigma}^-_a + v\,\hat{\sigma}^+_b\,, \tag{150}$$

where $u$ and $v$ are real and positive. Clearly, the state $|\psi\rangle = v|\uparrow_a\uparrow_b\rangle - u|\downarrow_a\downarrow_b\rangle$ is a dark state. For $u = v$ the system has a strong symmetry $\hat{\Pi}$ that exchanges $a$ with $b$ and flips both spins. Then the dark state is the only state with $\Pi = -1$ and decoupled from the rest of the manifold, which has a separate steady state. However, for $u \neq v$ the symmetry is lost and all states flow to the entangled dark state. Remarkably, this scheme generalizes to a chain of qubits where the dissipation acts only on the central two sites, as shown in [66],

$$\hat{H} = \hat{\sigma}^+_{a,1}\hat{\sigma}^-_{b,1} + \hat{\sigma}^-_{a,1}\hat{\sigma}^+_{b,1} + \sum_{i=1}^{l-1} J_i\big(\hat{\sigma}^+_{a,i}\hat{\sigma}^-_{a,i+1} + \hat{\sigma}^+_{b,i}\hat{\sigma}^-_{b,i+1} + \text{h.c.}\big) \quad \text{and} \quad \hat{L} = u\,\hat{\sigma}^-_{a,1} + v\,\hat{\sigma}^+_{b,1}\,. \tag{151}$$

The setup is sketched in Fig. 6. Here the steady state becomes a product of entangled pairs:

$$|\psi\rangle = \bigotimes_{i=1}^{l}\big[v\,|\uparrow_{a,i}\uparrow_{b,i}\rangle + (-1)^i u\,|\downarrow_{a,i}\downarrow_{b,i}\rangle\big]\,. \tag{152}$$

This "rainbow" state is annihilated by each of the terms in the summation in Eq. (151) (check this yourself!). The correlated dissipation may be realized by coupling those two sites to a lossy cavity as $\hat{H}' = (g_a\hat{\sigma}^x_a + g_b\hat{\sigma}^x_b)(\hat{c} + \hat{c}^\dagger)$ and applying local fields such that only $\hat{\sigma}^-_a\hat{c}^\dagger$ and $\hat{\sigma}^+_b\hat{c}^\dagger$ are resonant [66]. For more examples of dissipative state preparation see [2–7].

---

Q33. Consider identical bosons hopping on a uniform 1D lattice with periodic boundary conditions and subject to correlated dissipation on neighboring sites,

$$\hat{H} = -J\sum_i\big(\hat{b}^\dagger_i\hat{b}_{i+1} + \hat{b}^\dagger_{i+1}\hat{b}_i\big) \quad \text{and} \quad \hat{L}_i = \sqrt{\gamma}\,\big(\hat{b}^\dagger_i + \hat{b}^\dagger_{i+1}\big)\big(\hat{b}_i - \hat{b}_{i+1}\big) \;\forall i\,. \tag{153}$$

1) Does the dynamics conserve the total number of bosons?
2) What is the ground state $|\Psi\rangle$ of $N$ bosons?
3) Show that this is a dark state.
4) Show that $|\Psi\rangle$ is the only null state of all the Lindblad operators (for $N$ bosons).
5) $\hat{L}_i$ creates a symmetric superposition of sites $i$ and $i+1$. Do you see how that

---

leads to $|\Psi\rangle$? To learn about how such dissipation processes may be realized, see [65].

## 6.2 Quantum Zeno effect

Zeno's classic paradox says that motion (or evolution of a state) should be impossible because, in order to get from point A to point B, one has to reach the halfway mark, then the halfway of the remainder, and so on, requiring an infinite number of steps. Of course, we understand this is not really a paradox, as it is possible to divide a finite interval into infinitely many parts. On the other hand, in quantum mechanics, it is genuinely possible to freeze a system's dynamics over a finite duration by measuring it infinitely frequently. This effect has been demonstrated experimentally and is known as the quantum Zeno effect [88].

The simplest scenario is that of a spin-1/2 oscillating between its two levels under a magnetic field $\hat{H} = \frac{\Omega}{2}\hat{\sigma}^x$. Starting from $\uparrow$, the spin rotates in the $y$-$z$ plane at frequency $\Omega$, becoming $\downarrow$ after a period $T = \pi/\Omega$. Let us see how this changes if we measure $\hat{\sigma}^z$ at equal intervals $\tau = T/N$. The density matrix in the $z$ basis can be written as,

$$\hat{\rho} = \frac{1}{2}\begin{bmatrix} 1 + \langle\hat{\sigma}^z\rangle & \langle\hat{\sigma}^x\rangle - i\langle\hat{\sigma}^y\rangle \\ \langle\hat{\sigma}^x\rangle + i\langle\hat{\sigma}^y\rangle & 1 - \langle\hat{\sigma}^z\rangle \end{bmatrix}, \tag{154}$$

where the unitary dynamics follows

$$\frac{d}{dt}\langle\hat{\sigma}^x\rangle = 0, \quad \frac{d}{dt}\langle\hat{\sigma}^y\rangle = -\Omega\langle\hat{\sigma}^z\rangle, \quad \frac{d}{dt}\langle\hat{\sigma}^z\rangle = \Omega\langle\hat{\sigma}^y\rangle. \tag{155}$$

Starting from $\uparrow$, these equations give $\langle\hat{\sigma}^x\rangle = 0$, $\langle\hat{\sigma}^y\rangle = -\sin\Omega t$, $\langle\hat{\sigma}^z\rangle = \cos\Omega t$. A measurement of $\hat{\sigma}^z$ projects the spin to either $\uparrow$ or $\downarrow$, setting $\langle\hat{\sigma}^x\rangle = \langle\hat{\sigma}^y\rangle = 0$. Hence, the ensemble-averaged density matrix after the first measurement is diagonal with $\langle\hat{\sigma}^z\rangle = \cos\Omega\tau$. Then we evolve under Eq. (155) again for interval $\tau$ and measure $\hat{\sigma}^z$, which gives $\langle\hat{\sigma}^z\rangle = \cos^2\Omega\tau$. Iterating this process for $N$ measurements, one finds

$$\langle\hat{\sigma}^z\rangle(T) = \cos^N\Omega\tau = \cos^N(\pi/N) = \exp\left[-\frac{\pi^2}{2N} + O\left(\frac{1}{N^3}\right)\right] \xrightarrow[N\to\infty]{} 1. \tag{156}$$

Therefore, the state is frozen for sufficiently frequent measurements. This prediction [89] was confirmed experimentally in [90], where the measurements were performed by shining a pulse that transfers population from one of the levels (say $\uparrow$) to a third level that quickly decays back to $\uparrow$ by emitting a photon.

More generally, starting from an initial state $|\psi_0\rangle$, the survival probability under a Hamiltonian evolution is

$$P(t) = \left|\langle\psi_0|e^{-i\hat{H}t}|\psi_0\rangle\right|^2 = \left|1 - i\langle\hat{H}\rangle t - \frac{1}{2}\langle\hat{H}^2\rangle t^2 + O(t^3)\right|^2 = 1 - (\Delta E)^2 t^2 + O(t^4), \tag{157}$$

where $\Delta E$ is the energy uncertainty in $|\psi_0\rangle$, $(\Delta E)^2 = \langle\hat{H}^2\rangle - \langle\hat{H}\rangle^2$. On the other hand, if one measures the state at equal intervals $\tau \ll t$, the survival probability is

$$P_\tau(t) = [P(\tau)]^{t/\tau} = \exp\left[-\tau(\Delta E)^2 t + O(\tau^3)\right], \tag{158}$$

which decays at a rate $\tau(\Delta E)^2$, becoming steady for $\tau \to 0$.

The Zeno effect can also be used to arbitrarily steer the state of a system. To see this, let us go back to the spin-1/2 example but view it from a frame that rotates about the $x$ axis at frequency $\Omega$. In this frame, there is no magnetic field, but the measurements project the spin along the direction $\mathbf{e}_n = \sin(n\Omega\tau)\,\mathbf{y} + \cos(n\Omega\tau)\,\mathbf{z}$, $n = 1, 2, \dots, N$. The spin follows this moving axis, going from $\uparrow$ to $\downarrow$ in a time $T$. This evolution is caused solely by the measurements. This result is more general: if one continuously checks if a quantum system is evolving along a preselected trajectory in Hilbert space, it in fact does so [91]!

---

Q34.   Consider the spin-1/2 problem with $\hat{H} = \frac{\Omega}{2}\hat{\sigma}^x$, but where the measurement axis rotates arbitrarily in the $y$-$z$ plane. The $n$-th measurement is performed along the vector $\mathbf{e}_n = \cos\theta(n\tau)\,\mathbf{z} - \sin\theta(n\tau)\,\mathbf{y}$, where $\theta(t)$ is a smooth function of time and $\theta(0) = 0$. The spin points along $+z$ at $t = 0$. Show that the probability of finding the spin along the measurement vector at time $t$ is

$$P_{\mathrm{meas}}(t) = \exp\left[-\frac{\tau}{2}\int_0^t \mathrm{d}t'\left[\Omega - \omega(t')\right]^2 + O(\tau^3)\right], \quad \text{where} \quad \omega(t) := \frac{\mathrm{d}\theta}{\mathrm{d}t}. \quad (159)$$

---

Even if one always measures the same observable $\hat{O}$, the dynamics is not necessarily frozen. The measurements project the system onto the same eigenspace of $\hat{O}$, which is selected by the first measurement outcome. Within this Zeno subspace, however, the system can evolve unitarily under its Hamiltonian [92]. For example, consider spinless fermions hopping on a 1D lattice. If we continuously measure the occupation of a particular site, that site is frozen at either 0 or 1, but particles can still hop between the other sites. The freezing of the measured site splits the lattice into two halves, effectively turning off hopping to and from the measured site, as observed in [93]. More generally, the residual dynamics occurs under the block-diagonal Hamiltonian $\hat{H}_Z = \sum_n \hat{P}_n \hat{H} \hat{P}_n$, where $\hat{P}_n$ is the projector onto the $n$-th eigenspace of $\hat{O}$. Thus, the Zeno subspaces evolve independently even though $[\hat{H}, \hat{O}] \neq 0$. This superselection rule [88] emerges for infinitely frequent measurements.

Note that the same rule arises for a unitary dynamics under the Hamiltonian $\hat{H}' = \hat{H} + K\hat{O}$ in the limit $K \to \infty$. The only difference here is that, if the initial state is in a superposition of two eigenspaces of $\hat{O}$, the time-evolved state remains a superposition with the same relative weights. The similarity is not accidental. The continuously measured dynamics can also be modeled by a Hamiltonian $\hat{H}' = \hat{H} + K\hat{H}_{\mathrm{meas}}$, where $\hat{H}_{\mathrm{meas}}$ describes the interaction between the system and the measurement device [94]. Every measurement outcome or projection corresponds to a distinct (macroscopic) state of the device.

---

Q35.   Consider a qutrit with $\hat{H} = |0\rangle\langle 1| + |1\rangle\langle 0| + K(|1\rangle\langle 2| + |2\rangle\langle 1|)$, initialized in $|0\rangle$.

1) How does it evolve for $K = 0$? Plot the occupation of the three levels vs time.

2) Plot the evolution for $K = 10$. Does it make sense?

3) Show that the maximum occupation of $|1\rangle$ is given by $1/(K^2 + 1)$.

4) For $K = 10$, suppose we measure whether the system is in $|2\rangle$ at equal intervals $\tau = 0.1/K$, and find a negative answer every time. Plot the resulting evolution. Explain what you observe.

---

The Zeno effect also applies to Lindblad dynamics as it is closely related to measurements. In particular, as we saw in Sec. 4.1, a Hermitian Lindblad operator $\hat{O}$ is equivalent to the noise-averaged dynamics under a Hamiltonian $\hat{H}' = \hat{H} + \sqrt{\gamma}\,\eta(t)\hat{O}$, where $\gamma$ is the dissipation rate and $\eta(t)$ is a white noise satisfying $\overline{\eta(t)\eta(t')} = \delta(t-t')$. For $\gamma \to \infty$, the different eigenspaces of $\hat{O}$ acquire large random phases, which causes the coherence between these sectors to decay rapidly to zero upon noise averaging, after which the subspaces are decoupled. An example is a spin-1/2 with $\hat{H} = \hat{S}^x$ and $\hat{L} = \sqrt{\gamma}\hat{S}^z$. As shown in Fig. 7(a) for $\gamma = 20$, starting from a pure state in the $y$-$z$ plane, $\langle \hat{S}^y \rangle$ quickly becomes small, and thus $\langle \hat{S}^z \rangle$ relaxes very slowly [see Eq. (155)]. The residual coupling between $\uparrow$ and $\downarrow$ is due to the finite value of $\gamma$. One can find the relaxation rate from the Liouvillian gap $\Delta$ [Fig. 1(a)], which vanishes as $2/\gamma$ [Fig. 7(b)].

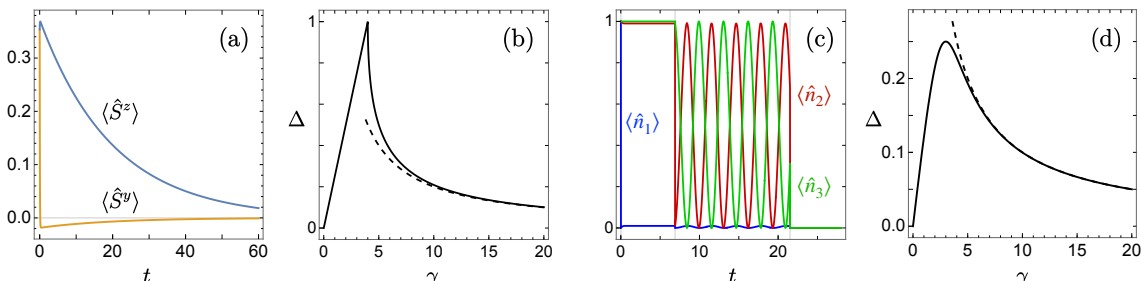

Figure 7: (a) Evolution of a spin-1/2 with $\hat{H} = \hat{S}^x$ and $\hat{L} = \sqrt{\gamma}\hat{S}^z$ for $\gamma = 20$. (b) Relaxation rate given by the Liouvillian gap $\Delta$. Dashed curve shows $2/\gamma$. (c) Evolution of a 3-site tight-binding chain with open boundaries (see text), where the first site has a loss rate $\gamma = 20$. (d) Relaxation rate. Dashed curve shows $1/\gamma$.

For a loss-type, non-Hermitian Lindblad operator $\hat{L} = \sqrt{\gamma}\,c$, it is useful to think of the dynamics as an average over quantum-jump trajectories, as discussed in Sec. 4.2.3. Along each trajectory, the unnormalized state $|\phi\rangle$ evolves under the effective Hamiltonian $\hat{H}_{\text{eff}} = \hat{H} - \frac{i}{2}\gamma\hat{c}^\dagger\hat{c}$, undergoing loss events whenever the norm $\langle\phi|\phi\rangle$ crosses below a preselected random number in $[0, 1]$. Clearly, for $\gamma \to \infty$, the norm will decay infinitely fast for any initial state with $\langle\hat{c}^\dagger\hat{c}\rangle > 0$. Thus, the system quickly reaches a state where $\hat{c}^\dagger\hat{c}$ vanishes. Thereafter, the dynamics is confined to the nullspace of $\hat{c}$ with projector $\hat{P}_0$. If $\hat{c}$ acts on a subsystem, the remaining part evolves unitarily under the projected Hamiltonian $\hat{H}_Z = \hat{P}_0\hat{H}\hat{P}_0$. For large but finite $\gamma$, one can show the dynamics within this Zeno subspace acquires a weak loss rate of $O(1/\gamma)$ [95], i.e., the remaining part eventually reaches steady state on a timescale of $O(\gamma)$. For example, consider a 3-site hopping model of spinless fermions where the first site is lossy, described by $\hat{H} = \hat{c}_1^\dagger\hat{c}_2 + \hat{c}_2^\dagger\hat{c}_3 + \text{h.c.}$ and $\hat{L} = \sqrt{\gamma}\,\hat{c}_1$ with $\gamma \gg 1$. Starting from a particle on each site, Fig. 7(c) shows that the first site quickly becomes empty ($t \sim 1/\gamma$). However, it takes a time $t \sim \gamma$ for the other sites to become empty. One can also see that this happens in stages: first the particle on site 2 escapes, then the last particle hops between 2 and 3 for a while until it is lost. The dissipation on the Zeno subspace acts on all neighbors of the lossy site (here, site 2), which can stabilize interesting long-range correlations if the lossy site is in the bulk [96].

## 6.3   Dissipative phase transitions

A dissipative phase transition corresponds to a sharp (non-analytic) change in the steady-state density matrix as a function of some parameter. The sudden change results from a closing of the Liouvillian gap in a thermodynamic or classical limit. Like equilibrium phase transitions, a dissipative transition can be discontinuous (first-order) or continuous (second or higher order). There is already a significant literature on the classification of different types of dissipative phase transitions [2, 97]. Here we mention a few instructive examples.

First, let us discuss a discontinuous transition in a driven nonlinear cavity, given by [98]

$$\hat{H} = -\Delta \hat{a}^{\dagger}\hat{a} + \frac{U}{2}\hat{a}^{\dagger}\hat{a}^{\dagger}\hat{a}\hat{a} + F(\hat{a}^{\dagger} + \hat{a}) \quad \text{and} \quad \hat{L} = \sqrt{\gamma}\,\hat{a}\,, \tag{160}$$

where $F$ is the drive amplitude, $\Delta$ is the detuning of the drive from the cavity frequency, $U$ is a nonlinearity in the spectrum (called Kerr nonlinearity), and $\gamma$ is the loss rate. To understand the phase transition, it is useful to analyze the classical equation of motion. Using Eq. (24),

$$\mathrm{i}\frac{\mathrm{d}}{\mathrm{d}t}\langle\hat{a}\rangle = -\left(\Delta + \mathrm{i}\frac{\gamma}{2}\right)\langle\hat{a}\rangle + U\langle\hat{a}^{\dagger}\hat{a}\hat{a}\rangle + F\,. \tag{161}$$

In the classical limit of large photon numbers, the state can be represented as a localized wave packet in phase space [25], so that $\langle\hat{a}^{\dagger}\hat{a}\hat{a}\rangle \approx \langle\hat{a}^{\dagger}\rangle\langle\hat{a}\rangle\langle\hat{a}\rangle$. This gives a closed equation of motion for the field amplitude $\alpha = \langle\hat{a}\rangle$. In particular, the rescaled amplitude $\tilde{\alpha} := \alpha/\sqrt{N}$ satisfies

$$\mathrm{i}\frac{\mathrm{d}}{\mathrm{d}t}\tilde{\alpha} = \left(-\Delta - \mathrm{i}\frac{\gamma}{2} + \tilde{U}|\tilde{\alpha}|^2\right)\tilde{\alpha} + \tilde{F}\,, \tag{162}$$

where $\tilde{U} := UN$ and $\tilde{F} := F/\sqrt{N}$, with the photon number scaling as $\langle\hat{a}^{\dagger}\hat{a}\rangle \approx N|\tilde{\alpha}|^2$. Thus, we can reach the classical limit by taking $N \to \infty$ at fixed $\tilde{U}$ and $\tilde{F}$. For $\Delta > (\sqrt{3}/2)\gamma$, Eq. (162) predicts two stable fixed points with different photon numbers for a range of drive amplitudes [dashed curve in Fig. 8(a)]. However, the quantum steady state is unique and interpolates between the two classical branches (solid curves). The interpolation becomes increasingly sharp as $N$ is increased, yielding a discontinuous transition in the limit $N \to \infty$. In the crossover region, the steady-state distribution is bimodal [Fig. 8(c)] and the system stochastically switches between the classical solutions along a given trajectory [99, 100], producing large fluctuations [Fig. 8(b)]. As $N$ is increased the crossover region shrinks to a point and the steady state picks one classical branch. The other branch is contained in the slowest-decaying mode, which is separated by an exponentially small gap throughout the bistable region [2] [Fig. 8(c)].

---

Q36.   Consider the same model with $\gamma = 1$, $\Delta = 3$, and $\tilde{U} = 1$.

1) Using $\tilde{\alpha} = x + \mathrm{i}p$ in Eq. (162), make a phase portrait ("stream plot") in the $x$-$p$ plane for $\tilde{F} = 1.5$. Identify two stable fixed points and a saddle.

2) By varying $\tilde{F}$ from 0.5 to 2.5, reproduce the dashed curve in Fig. 8(a).

3) Plot the Liouvillian gap on a log scale as a function of $\tilde{F}$ for $N = 1, 3, 10, 20$. What do you observe?

---

Let us now discuss a type of continuous transition that arises in systems with a parity-time reversal (PT) symmetry [81]. These are systems with balanced gain and loss acting on two

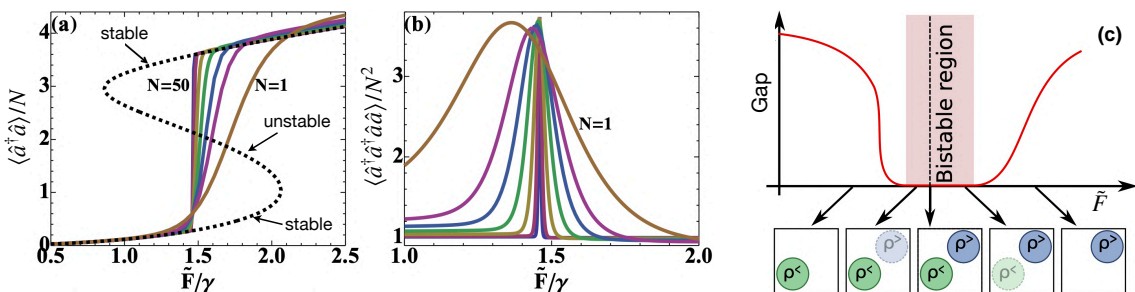

Figure 8: (a) Average photon number and (b) number fluctuation in the steady state of Eq. (160) for $\Delta = 3\gamma$ and $\tilde{U} = \gamma$. Dotted curve shows classical bistability. Reproduced with permission from [98] [Copyright (2017) by the American Physical Society]. (c) Sketch of the exponentially small Liouvillian gap and bimodal steady state across a bistability. Reproduced from [2].

symmetric parts. For example,

$$\hat{H} = g\big(\hat{S}_A^+\hat{S}_B^- + \hat{S}_A^-\hat{S}_B^+\big) \quad \text{and} \quad \hat{L}_A = \sqrt{2\Gamma}\,\hat{S}_A^+\,, \ \ \hat{L}_B = \sqrt{2\Gamma}\,\hat{S}_B^-\,, \tag{163}$$

where $A$ and $B$ are two spin-$S$. Such systems are invariant under a joint application of parity $\hat{P}$, which swaps $A$ and $B$, and time reversal, which exchanges gain with loss. In other words,

$$\mathcal{L}\big[\mathcal{PT}(\hat{H}), \{\mathcal{PT}(\hat{L}_A), \mathcal{PT}(\hat{L}_B)\}\big] = \mathcal{L}\big[\hat{H}, \{\hat{L}_A, \hat{L}_B\}\big] \quad \text{where} \quad \mathcal{PT}(\hat{O}) := \hat{P}\hat{O}^\dagger\hat{P}\,. \tag{164}$$

One can ask whether the steady state, which is unique for any finite $S$, has the parity symmetry. For the spin model, this can be measured by the order parameter [81]

$$\Delta := \frac{\big|\langle \hat{S}_A^-\hat{S}_A^+\rangle - \langle \hat{S}_B^-\hat{S}_B^+\rangle\big|}{\langle \hat{S}_A^-\hat{S}_A^+\rangle + \langle \hat{S}_B^-\hat{S}_B^+\rangle}\,, \tag{165}$$

which lies between 0 and 1. A nonzero value of $\Delta$ means the parity symmetry is broken. Figure 9(a) shows that, as $\Gamma/g$ is increased, the system undergoes a symmetry-breaking phase transition that becomes sharper for larger values of $S$. Furthermore, the steady state is completely mixed for $\Gamma/g \to 0$ and pure for $\Gamma/g \to \infty$. For $\Gamma < g$, the exchange between $A$ and $B$ occurs much faster than the gain and loss, which randomizes the system over a long time. On the other hand, for $\Gamma \gg g$, the gain and loss drive the system toward the polarized pure state where $A$ points along $z$ and $B$ points along $-z$. In addition, the dynamics changes from oscillatory to overdamped across the transition [Figs. 9(b) and 9(c)].

One can formulate a broader class of continuous phase transitions in terms of a spontaneous weak-symmetry breaking, where the symmetry-broken phase is gapless [2,30,97,101]. However, this is not a prerequisite [102]. One can also find both first-order and second-order transitions in the same system, as in [103–106].

Q37. Consider $N$ qubits that are resonantly driven and undergo collective spontaneous emission, described by $\hat{H} = \hat{S}^x$ and $\hat{L} = \sqrt{\gamma/S}\,\hat{S}^-$, where $S = N/2$. This is a model for cooperative resonance flourescence [107] and also for a boundary time crystal [29].

1) Show that the Liouvillian $\mathcal{L}$ commutes with the superoperator $\mathcal{T}$ defined as $\mathcal{T}(O) := e^{i\pi\hat{S}^z} O^* e^{-i\pi\hat{S}^z}$, where $O$ is any operator written in the $z$ basis.

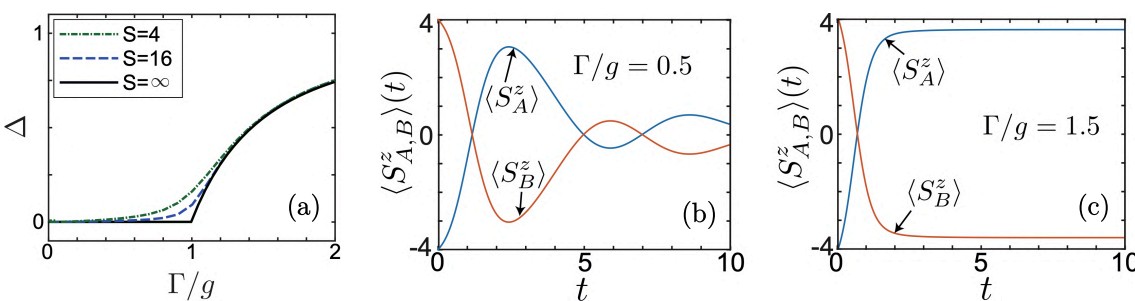

Figure 9: (a) Steady-state imbalance, (b,c) dynamics with $S = 4$ for the model in Eq. (163). Reproduced from [81].

2) How does $\mathcal{T}$ change the matrix elements of an arbitrary operator?

3) Given the steady state is unique, show it is invariant under $\mathcal{T}$ and has $\langle \hat{S}^x \rangle = 0$.

4) Plot $\langle \hat{S}^y \rangle$ and $\langle \hat{S}^z \rangle$ in steady state as a function of $\gamma$ for $S = 5, 10, 25, 100$. What kind of transition do you observe?

5) Find the equations of motion for $s_\alpha := \langle \hat{S}^\alpha \rangle / S$, $\alpha = x, y, z$ in the classical limit $S \to \infty$, when the noncommutativity of the spin operators can be ignored. Show that the equations conserve $s_x^2 + s_y^2 + s_z^2$.

6) Make phase portraits on the Bloch sphere, $s_x^2 + s_y^2 + s_z^2 = 1$, for $\gamma = 0.8$ and $1.2$. What is the long-time behavior of a generic initial state in the two cases?

7) Check that the quantum model shows the same behavior as $S$ is increased.

In many-body systems, continuous dissipative phase transitions provide a mechanism for the emergence of long-range order in a nonequilibrium steady state. An example is a boundary-driven XXZ spin-1/2 chain [108],

$$\hat{H} = \sum_{i=1}^{n-1} \left( \hat{\sigma}_i^x \hat{\sigma}_{i+1}^x + \hat{\sigma}_i^y \hat{\sigma}_{i+1}^y + \Delta \hat{\sigma}_i^z \hat{\sigma}_{i+1}^z \right) \quad \text{and} \quad \hat{L}_{1,\pm} = e^{\pm\mu} \hat{\sigma}_1^\pm, \quad \hat{L}_{n,\pm} = e^{\mp\mu} \hat{\sigma}_n^\pm, \qquad (166)$$

where $\Delta$ is the anisotropy and $2\mu$ is the chemical potential difference between two reservoirs connected to the opposite ends—on their own, the dissipators would drive the first and last sites to the state $\hat{\rho} \propto e^{\mu(\hat{\sigma}_1^z - \hat{\sigma}_n^z)}$. For the spin chain, one finds that the spin correlations $C(i,j) := \langle \hat{\sigma}_i^z \hat{\sigma}_j^z \rangle - \langle \hat{\sigma}_i^z \rangle \langle \hat{\sigma}_j^z \rangle$ are short-ranged for $\Delta < \Delta_c \approx 0.91$ and long-ranged for $\Delta > \Delta_c$. In particular, for $\Delta < \Delta_c$ they decay exponentially with the separation $r \equiv |i-j|$, $C(r) \sim e^{-r/\xi}$, where $\xi$ is independent of the system size $n$. On the other hand, for $\Delta > \Delta_c$ they have the form $C(r) \sim 4\mu^2(1 - r/n)^2$, becoming a constant for $n \to \infty$ [108]. In comparison, a thermal state has only short-ranged correlations. Even in the ground state, the correlations are short-ranged for $|\Delta| < 1$ and decay as a power law for $|\Delta| < 1$. Hence, the steady states throughout the phase transition are far from equilibrium.

## 6.4 Measurement-induced phase transition

So far we have discussed phase transitions in the density matrix, obtained by averaging over pure-state realizations. Phase transitions can also occur at the trajectory level, without necessarily showing up in the average density matrix. Such is the case for measurement-induced

phase transitions, which occur in many-body systems whose unitary evolution is punctuated by repeated projective measurements [109].

Interacting quantum systems typically evolve toward a state with volume-law entanglement. This is when the entanglement entropy between a (small) subsystem and its complement scales as the volume (number of sites) of the subsystem. These are also the most typical states of a many-body system [110]. In contrast, a local measurement can reduce entanglement by collapsing local degrees of freedom. For instance, a projective measurement of a single site in a definite state disentangles it from the rest of the system. Thus, one may ask: if each site is measured at a rate $p$, does the many-body state collapse to something close to a product state with area-law entanglement, or does it retain volume law? Surprisingly, for short-range interactions one finds a sharp, continuous transition from volume law to area law as $p$ is increased beyond a critical value $p_c$ [109]. Furthermore, starting from a product state in 1D, the entanglement $S$ saturates for $p > p_c$, grows linearly with time for $p < p_c$, and logarithmically with time $p = p_c$, as sketched in Fig. 10(a).

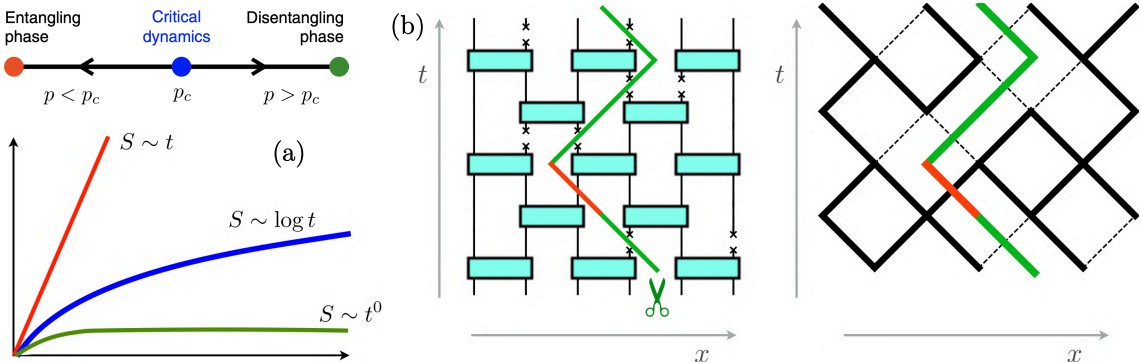

Figure 10: (a) Growth of entanglement entropy across a measurement-induced phase transition. (b) Evolution by unitary gates (boxes) punctuated by single-site measurements (crosses). Solid line shows a "minimal cut" separating the left and right halves (see text). (c) Analogy to bond percolation. Figures reproduced from [109].

To see how this comes about, it is instructive to consider a discrete unitary evolution of a spin-1/2 chain, where each step involves the application of two-site unitary gates to neighboring spins, represented by the boxes in Fig. 10(b). The gates can be identical as in a Floquet dynamics or they can be chosen at random. Each application of the gates entangles sites that are farther apart. Thus, in $t$ steps, $O(t)$ qubits in a subsystem get entangled with the rest of the system, leading to volume law. In fact, one can show that a measure of bipartite entanglement (the zeroth Rényi entropy) is given by the number of bonds one must cut in the spacetime circuit to isolate the subsystem [109]. Clearly, this number scales with $t$.

Now suppose after every time step, each spin is measured with probability $p$. Such a measurement destroys all entanglement the measured spin had developed with other spins. One can think of this as "breaking a bond" in the spacetime circuit. Then the entanglement is given by the minimum number of unbroken bonds one has to cut to separate the subsystem. Figure 10(b) shows such a minimal cut. This way the problem reduces to that of bond percolation on a 2D square lattice. For $p > p_c$ the broken bonds percolate, producing a continuous

path from top to bottom with $S \sim O(1)$, whereas for $p < p_c$, broken bonds exist only in small clusters, leading to $S \sim O(t)$. At $p = p_c$, the clusters have a scale-invariant fractal structure, which can be shown to yield $S \sim \log t$ [111]. One also finds that, close to the transition, both correlation lengths and times diverge as $|p - p_c|^{-2}$ [109].

Note, however, the transition does not show up in the ensemble-averaged steady state, which is typically maximally mixed for all values of $p$.

# 7 Conclusion

We have tried to present an overview of some of the key ideas in the field of open quantum systems, highlighting features that are absent in a Hamiltonian setting. For the sake of brevity, however, we have had to omit a number of exciting research topics, including non-Hermitian topology [112,113] and the physics of exceptional points [114–118], quantum thermodynamics [119–122], thermalization [123–127] and Mpemba effects [128,129], dissipative quantum chaos [43,130–135], emergence of classical nonlinear phenomena [30,136–139], dissipative phenomena on networks [140–142], universality [143], quantum resetting [144–148], as well as non-Markovian physics [149–153]. We hope these notes provide enough background and inspiration to the reader to further explore this rich and fertile research field.

# Acknowledgements

I thank Nigel Cooper and Masud Haque for countless discussions on open quantum systems.

**Funding information** This work was supported by intramural funds of the Raman Research Institute.

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
