# Peer review of "An introduction to Markovian open quantum systems"

_SciPost Physics Lecture Notes_

## Round 2 · Referee Report · Anonymous (Referee 1) · 2025-11-13

Strengths

1- a very useful and pedagogical introduction to Markovian master equations 2 - acceptance criteria match those of SciPost Physics Lecture Notes 3 - covers the stochastic unraveling, weak, strong, and dynamical symmetries in just 46 pages 4- There are also exercises

Report

The paper meets all the criteria of SciPost Physics Lecture Notes and therefore I recommend publishing the paper.

Recommendation

Publish (surpasses expectations and criteria for this Journal; among top 10%)

---

## Round 2 · Referee Report · Anonymous (Referee 2) · 2025-11-25

# Referee Report

*An Introduction to Markovian Open Quantum Systems*

Shovon Dutta

November 23, 2025

The referees wish to congratulate the author on such a beautifully crafted and concise introduction to QOS. We appreciate the effort and vision of the author in keeping it brief and to the point. We found the manuscript almost error-free. What follows is a list of suggestions and possible corrections. The author is encouraged to consider them, while feeling free to follow them or not.

**Comments**

- p.2 top line, perhaps one could mention other applications such as quantum thermodynamics

- p.2 second paragraph, while citing [2,7,13-15] perhaps one can cite [28] too.

- **Sentence:** "A density matrix describes our incomplete knowledge of a quantum state." (Sec. 2, p. 3)

  **Comment:** Perhaps not true entirely, since a density matrix can also be associated to a pure state.

- p.4 Consider citing the work on entanglement negativity by Plenio [Phys. Rev. Lett. 95, 090503 (2005)] along with that on negativity [17].

- **Sentence:** "An extreme example is $\rho \propto 1$, for which $\rho_A \propto 1_A$ and $S$ is maximum even though $A$ and $B$ are not entangled" (Sec. 2, p. 4)
  **Comment:** Maybe a couple more lines explaining this would be useful.

- p.5 Different symbols are used for the identity in this and other pages, including $I$ and $\hat{1}$. It is unclear to the reader whether the choice carries any connotation. If not, it may help to use a single symbol throughout the text.

- p.5 The choice of notation for the matrix elements of rho, $\rho_{(i,j),(i',j')}$ is somewhat cumbersome, though clear.

- **Sentence:** "This is achieved by setting all but one Kraus operators proportional to $\sqrt{dt}$." (Sec. 3, p. 6)
  **Comment:** It would be useful to write a sentence about why any other choice of Kraus operators (in terms of proportionality to $\sqrt{dt}$) gives an equivalent / incompatible description. This could even be a simple exercise for the reader.

- p.7, in-text eq above (14) replace 1 by $\hat{1}$ or $I$ in the definition of $\hat{K}_0$.

- **Sentence:** "$\hat{H}$ is an arbitrary Hermitian operator." (Sec. 3, p. 7)
  **Comment:** It is not clear why $\hat{H}$ has to be Hermitian.

- p.8, after (16), how a bout a citation when introducing the Kossakowski matrix?

- **Sentence:** Eqns. 23, 24 (Sec. 3, p. 9)
  **Comment:** Useful to explicitly remind the reader that $\langle \hat{O} \rangle = \text{Tr}(\rho \hat{O})$.

- p.9 Does (15) motivate the introduction of the adjoint of the dissipator?

- **Sentence:** Eqn. 26 (Sec. 3, p. 9)
  **Comment:** Please mention the meaning of the term $\langle \sigma_{x,y,z} \rangle$ that occurs in the density matrix.

- p.10 Q6 calls to mind a theorem by Lidar Shabani and Alicki on the conditions for purity decreasing dynamics, for which the map being unital is a sufficient condition

  https://doi.org/10.1016/j.chemphys.2005.06.038

- p.10, is there a typo in (32)? Using (31) it appears that the first H in the RHS of (32) should be $I \otimes H_{\text{eff}}^T$ instead of $I \otimes H_{\text{eff}}^*$ Note also that the definition of $H_{\text{eff}}$ only appears after (33), and could have provided earlier on to ease readability.

- p.11 In 3.3.7, we believe the author is trying to simplify the flow of the exposition in an accessible way. Perhaps the first sentence is not that accurate, as in general the diagonalization of the Liouvillian brings a Jordan block form. We have in mind the exposition at the level of the review by Advances in Physics 69, 3 (2020)

- **Sentence:** "but $(L\hat{\rho})^\dagger \neq \hat{\rho} L^\dagger$ or $\hat{\rho}^\dagger L^\dagger$." (Sec. 3, p. 11)
  **Comment:** Useful to specify that this is because $L$ is a superoperator and not an operator.

- p.13 Fluctuating Hamiltonians have a long history. One could cite, e.g., the book by van Kampen at the beginning of Sec 4.1

- p. 15, Eq. (53). It may be worth emphasizing whether this equation is exact or involves a perturbative expansion in the coupling. Compare this with the perturbative equation in Budini PHYSICAL REVIEW A, VOLUME 64, 052110 (2001)

- **Sentence:** "This is called a stochastic Schrödinger equation (SSE), and the resulting pure-state trajectories are called Monte-Carlo wave functions [47]." (Sec. 4, p. 19)
  **Comment:** The classic review by Zoller may be mentioned here: `https://arxiv.org/pdf/quant-ph/9702030`.

- **Sentence:** Eqn. 90. (Sec. 4, p. 13)
  **Comment:** Should be $\tilde{H}_{SR}$.

- p.28, well, a colon after $\langle \chi | \chi \rangle$?

- **Sentence:** "odd single-particle eigenfunctions". (Sec. 5.4, p. 35)
  **Comment:** Please specify what is meant by odd or even in this context.

- p.46, perhaps the citations in Sec 7 are somewhat idiosyncratic, but that is hard to avoid given the terseness of the closing paragraph.

---

## Round 2 · Referee Report · Anonymous (Referee 3) · 2025-12-15

Strengths

  1. Concise introduction to open systems
  2. Good practice problems
  3. Pedagogical and accessible

Report

I think this is a very nice and useful introduction to the field of open systems for the beginning graduate student. I particularly like the inclusion of a good number of practice problems. I have a few small points where I think the clarity could be improved:

1. On page 12, there is a discussion of the eigenvalues of the Lindbladian having nonpositive real part: “If the dynamics is well behaved at long times—as is true in most physical situations—all nonzero eigenvalues in the summation must have nonpositive real parts.” It could perhaps be useful to point out that this is always true for finite dimensional system (i.e. from the contraction-mapping theorem). What constitutes well-behaved dynamics is also a bit vague – one could reasonably define this as the Lindbladian being negative semi-definite, in which case this is a tautology. It could also be useful to point out there are very simple, physically motivated models in infinite dimensional Hilbert spaces (bosonic modes) where this fails, such as a simple parametric amplifier.

2. Also on page 12, when discussing the spectrum of a spin model vs a damped Harmonic oscillator, it is written: “The wedge shape in Fig. 1(b) is characteristic of a classical fixed point, whereas the parabolic shapes in Fig. 1(c) indicate damped oscillations”. I believe either of these could reasonably be called decaying oscillations. The wedge shape in 1(b) is characteristic of a non-interacting system, where there is a mode structure.

3. In section 4.1, and especially at the top of page 14 when discussing the delta function at the boundary of the integral, the text plays a bit fast and loose with the stochastic calculus. I appreciate the point of these lecture notes are not suitable for a rigorous derivation, but perhaps a reference to a proper derivation using the Ito calculus would help point the curious student in the right direction for how to put this on a more rigorous footing.

4. On page 14, colored noise is introduced without defining it. Should mention that this is any noise source with a non-flat spectral density (and perhaps some experimentally motivated examples, like 1/f noise).

5. On page 29, the text reads: “In many-body systems, generic states, including eigenstates of an interacting Hamiltonian, have correlations between far-off sites.” It is important to qualify that this is only eigenstates in the bulk of the spectrum. The reader might infer this to mean many-body ground states, which of course can only have long range correlations if the Hamiltonian is gapless.

6. On page 43, the text reads: “One can formulate a broader class of continuous phase transitions in terms of a spontaneous weak-symmetry breaking, where the symmetry-broken phase is gapless”. It is important to be careful with language here. From a quantum optics perspective, having two degenerate zero modes might be defined as a gapless spectrum, but from a more traditional condensed matter perspective, a spectrum is only gapless if there is a continuum of modes around the ground state energy, whereas just having multiple (but a finite number of) steady states separated by a finite gap would just be a ground state degeneracy. This is especially important when discussing phases of matter, which are typically only well defined for gapped Liouvillians/Hamiltonians (in the condensed matter sense), and the phase transition point is gapless. Some clarifying remarks might be useful to avoid confusion.

7. In the discussion of XXZ subject to boundary dissipation, one should also add the reference: Prosen, PRL 107, 137201 (2011). This gives exact, analytic results in agreement with Ref. [108] in the text.

Recommendation

Publish (easily meets expectations and criteria for this Journal; among top 50%)

---

## Editorial Decision

in_refereeing